# Unravelling the origins and P-T-t evolution of the allochthonous Sobrado unit (Órdenes complex, NW Iberia) using combined U-Pb titanite, monazite and zircon geochronology and REE geochemistry

José Manuel Benítez-Pérez[1,3], Pedro Castiñeiras[2], Juan Gómez-Barreiro[3][(*)], José R. Martínez Catalán[3], Andrew Kylander-Clark[4], Robert Holdsworth[5]

[1]Centro de Ciências e Tecnologias Nucleares. Instituto Superior Técnico. Universidad de Lisboa. Estrada Nacional 10 (km 139,7), 2695-066, Bobadela LRS, Portugal
[2]Departamento de Mineralogía y Petrología. Facultad de Ciencias Geológicas, Universidad Complutense de Madrid. C/ José Antonio Novais, 12. Ciudad Universitaria, 28040, Madrid, Spain
[3]Departamento de Geología, Universidad de Salamanca, Pza. de los Caídos s/n, 37008, Salamanca, Spain
[4]Department of Earth Science, University of California, Santa Barbara, CA 93106, United States
[5]Earth Science Department, Durham University, Science Labs, Durham DH1 3LE, United Kingdom

Correspondence to: J. Gómez Barreiro (jugb@usal.es)

jose.benitez@ctn.tecnico.ulisboa.pt
castigar@ucm.es
jrmc@usal.es
kylander@geol.ucsb.edu
r.e.holdsworth@durham.ac.uk

## Abstract

The Sobrado unit, within the upper part of the Órdenes complex (NW Iberia) represents an allochthonous tectonic slice of exhumed high grade metamorphic rocks formed during a complex sequence of orogenic processes in the middle to lower crust. In order to constrain those processes, U-Pb geochronology and REE analyses of accessory minerals in migmatitic paragneisses (monazite, zircon), and mylonitic amphibolites (titanite) were conducted using LASS-ICP-MS. The youngest metamorphic zircon age obtained coincides with a Middle Devonian concordia monazite age (~380 Ma) and is interpreted to represent the minimum age of the Sobrado high-P granulite-facies metamorphism that occurred during the early stages of the Variscan Orogeny. Metamorphic titanites from the mylonitic amphibolites yield a Late Devonian age (~365 Ma) and track the progressive exhumation of the Sobrado unit. In zircon, cathodoluminescence images and REE analyses allow two aliquots with different origins in the paragneiss to be distinguished. An Early Ordovician age (~490 Ma) was obtained for metamorphic zircons, although with a large dispersion, related to the evolution of the rock. This age is considered to mark the onset of granulite-facies metamorphism in the Sobrado unit under intermediate-P conditions, and related to intrusive magmatism and coeval burial in a magmatic arc setting. A maximum depositional age for the Sobrado unit is established in the late Cambrian (~511 Ma). The zircon dataset also record several inherited populations. The youngest cogenetic set of zircons yields crystallization ages of 546 and 526 Ma and are thought to be related to the peri-Gondwana magmatic arc. The additional presence of inherited zircons older than 1000 Ma is interpreted as suggesting a West African Craton provenance.

**Keywords**: U-Pb geochronology, LASS-ICP, zircon, titanite, monazite, REE, Sobrado Unit

# 1. Introduction

Zircon, monazite and titanite are accessory mineral phases found in rocks with a very wide range of compositions. These minerals can resist numerous sedimentary, igneous and metamorphic events across a wide range of temperatures, pressures and strains, even when fluids are present. Frequently, compositional domains can be defined in these minerals that record changes in different parameters (Storey et al., 2007; Castiñeiras et al., 2010; Stübner et al., 2014; Hacker et al., 2015; Stearns et al., 2016; Stipska et al., 2016). These minerals additionally provide several closed decay chains or disintegration systems ($^{238}U \rightarrow {}^{206}Pb$, $^{235}U \rightarrow {}^{207}Pb$ y $^{232}Th \rightarrow {}^{208}Pb$), because they hold variable concentrations of uranium (U) and/or thorium (Th) in their crystal lattices. Such variations in concentration allow accurate dating using microscopic scale analysis (tens of microns size).

Titanite is stable in metabasites across a wide range of metamorphic conditions (Spear, 1981; Frost et al., 2000) and is able to record metamorphic and deformational events (Franz and Spear, 1985; Verts and Frost, 1996; Rubatto and Hermann, 2001; Spencer et al., 2013; Stearns et al., 2015, 2016). The titanite U/Pb system is a widely used geochronometer for deformation events in granulite-amphibolite facies rocks (Spear, 1981; Cherniak, 2006; Harlov et al., 2006). Monazite is common in amphibolite and higher-grade facies. Zoning in this mineral can have an igneous or metamorphic origin (DeWolf et al., 1993; Hawkins and Bowring, 1997; Zhu et al., 1997; Spear and Pyle, 2002). The crystallization stages seen in zoned monazites, with changes in Y, Ca, Si, Sr, Ba, REE, U and Th, has been linked to certain metamorphic reactions (Kohn and Malloy, 2004; Corrie and Kohn, 2008) or deformation process (Terry and Hamilton, 2000). Zircon survives the majority of magmatic, metamorphic and erosive terrestrial processes. Cathodoluminescence analysis of zircon zoning patterns allows a large variety of reactions to be distinguished and can clarify the petrogenetic evolution (Corfu et al., 2003). Th/U ratios can also be used to separate zircon based on their igneous or metamorphic origin (Hoskin and Ireland, 2000; Möller et al., 2002; Hokada and Harley, 2004; Hoskin, 2005). Rare-earth element (REE) abundances can also be used as a qualitative petrological indicator. Heavy rare-earth elements (HREE) are preferentially incorporated into zircon compared to light rare-earth elements (LREE). Hence, the normalized HREE slope can be used to interpret whether a zircon crystallized or recrystallized when garnet and xenotime ($YPO_4$) were present, because these minerals also preferentially assimilate HREE in the lattice (Hoskin and Ireland, 2000; Rubatto, 2002; Hermann and Rubatto, 2003; Rubatto et al., 2009).

The events recorded in individual grains can be radiometrically dated employing combined laser ablation analyses and cathodoluminescence (CL) images in zircons (Corfu et al., 2003) and compositional maps obtained using electron microprobe (EMPA) in monazite (Gonçalves et al., 2005; Williams et al., 2007) to recognize different growth zones. The chemical analysis, especially REE, links the development of growth zones to specific metamorphic or deformative events (Frost et al., 2000; Rubatto, 2002; Whitehouse and Platt, 2003; Zheng et al., 2007; Chen et al., 2010; Gagnevin and Daly, 2010; Holder et al., 2015;). Simultaneous geochronology and REE data are a powerful tool in the interpretation of ages - this is known as REE-assisted geochronology (Castiñeiras et al., 2010).

This methodology has been applied to selected samples of the Sobrado unit (Fig. 1), which forms part of the allochthonous complexes of NW Iberia and one where the structural and metamorphic evolution is rather well known (Pablo Maciá et al., 1984; Díaz García et al., 1999; Arenas and Martínez Catalán, 2002; Benítez-Pérez, 2017). This unit occurs in the Upper Allochthon of the Órdenes complex, and represents a tectonic slice of exhumed ultramafics, eclogites, high-P granulites, amphibolites and migmatitic paragneisses derived from a peri-Gondwanan terrane. It evolved along a complex sequence of geological processes involving Cambro-Ordovician rifting with voluminous bimodal plutonism, nearly contemporaneous granulite facies metamorphism of intermediate-P, early Variscan subduction and subsequent exhumation by ductile thrusting during Variscan collision with the external margin of northern Gondwana.

Previous geochronological data on the Sobrado unit include U-Pb ages from four samples
(Fernández-Suárez et al., 2002, 2007). A middle Cambrian protolith age of a gabbro and a Middle
Devonian metamorphic age of a high-P basic granulite supposedly derived from the same gabbro were
obtained by zircon dating. Zircon in a migmatitic, mylonitized paragneisses yielded discordant ages with
an Early Ordovician lower intercept, while monazite dating provided Cambro-Ordovician ages in another
migmatitic, mylonitized paragneiss. This new study aims at constraining the metamorphic evolution of
the unit including dating a migmatitic paragneiss, as previous data missed the early Variscan ages found
in intercalated high-P granulites, and also an amphibolite, which could date advanced stages of
exhumation. Monazite and zircon ages of paragneisses and titanite ages of amphibolites taken from
separate, but presently adjacent tectonic slices of the high-P/high-T of Sobrado unit are compared and
interpreted using REE-assisted geochronology. This sheds new light upon the possible origin, ages and
relationships between the regional foliation development and the partial melting processes that have
occurred in this and equivalent units of the NW Iberia Upper Allochthon.

**2.     Geological background**

The Allochthonous complexes in NW Iberia are remmants of a huge nappe stack preserved as
klippen in the core of late Variscan synforms. They consist of units mostly of peri-Gondwanan derivation,
which can be classified in three groups based on their structural position in the tectonic pile and origin:
The Upper, Middle and Lower allochthons (Fig. 1).

The Upper Allochthon is a piece of the northern margin of Gondwana detached and drifted away
during the Cambro-Ordovician opening of the Rheic Ocean. The Middle Allochthon is formed by
lithospheric pieces of oceanic affinity, or oceanic supracrustal sequences that formed part of the Rheic
oceanic realm and are often referred to as ophiolitic units. The Lower Allochthon derives from distal parts
of the Gondwanan continental margin.

The Allochthon units are separated from the Iberian Autochthon by a series of kilometer-scale
imbricated sheets, known as the Parautochthon (Ribeiro et al., 1990), or Schistose Domain in Galicia
(NW Spain), consisting of a set of Paleozoic metasedimentary and volcanic rocks. The Parautochthon has
stratigraphic and igneous affinities with the Iberian autochthon of the Central Iberian Zone, and no
ophiolites occur between them. For these reasons it is interpreted as a distal part of the Gondwanan
continental margin (Farias et al., 1987; Dias da Silva et al., 2014).

The allochthonous units are regarded as a stack of Varican thrust sheets with associated tectonic
fabrics and metamorphic events. Due to the "piggy-back" nature of the sequence, the structurally higher
units are thought to represent the furthest travelled paleogeographic domains. Recumbent folds, thrusts
and extensional detachments formed during the Variscan collision are found in all three allochthonous
units (Martínez Catalán et al., 1999; Gómez-Barreiro et al., 2007).

Maximum sedimentation ages obtained from the study of detrital zircon carried out in
metasediments from the Upper allochthon can be estimated between 530 and 510 Ma (e.g., Fuenlabrada
et al., 2010). Intrusive rocks in the Upper allochthon have been dated between 520 and 490 Ma and are
associated with the development of a magmatic arc which evolves into an extensional scenario that ended
with the opening of the Rheic ocean (Peucat et al., 1990; Ordóñez Casado, 1998; Abati et al., 1999, 2007;
Fernández-Suárez et al., 2007; Castiñeiras et al., 2010). Two high-P/high-T metamorphic events have
been recognized in this unit. The oldest one has yielded 490-480 Ma (Kuijper 1979; Peucat et al. 1990;
Abati et al., 1999, 2007; Fernández-Suárez et al., 2002) and the youngest one has been dated
approximately between 405-390 Ma (Santos Zalduegui et al., 1996; Ordóñez Casado et al., 2001;
Fernández-Suárez et al., 2007). In the Middle allochthon, crystallization ages vary between 390 and 375
46  Ma (Peucat et al., 1990; Dallmeyer et al., 1991, 1997) and ages from 375 to 365 Ma have been related to
continental subduction (Santos Zalduegui et al., 1995; Abati et al., 2010). Thrust wedge collapse, in the

middle and lower allochthonous units, is thought to have happened between 390 and 365 Ma, followed by a collision in the internal zones around 365-330 Ma, causing further folding and thrusts (Dallmeyer et al., 1997; Martínez Catalán et al., 2009). Afterwards, there was another extensional collapse phase until 315 Ma, followed by a final phase of shortening and folding up until approximately 305 Ma related to the regional oroclinal bending in Iberia (Aerden, 2004; Martínez Catalán, 2011, 2012; Álvarez-Valero et al., 2014).

The Upper Allochthon is further subdivided into high-P/high-T and intermediate-P units (Gómez Barreiro et al., 2007). The present study focuses on two of the high-P/high T upper units (Figs. 1 and 2). The origin of the high-P event recorded in these units is controversial, but might reflect the accretion of the units into the continental part of northern Gondwana, during the Early-Middle Devonian (400-390 Ma; Ballèvre et al., 2014).

The Sobrado antiform consists of three tectonic slices bounded by two extensional detachments (Figs. 1 and 2). The lower tectonic slice comprises highly serpentinized ultramafic rocks with interlayered metabasite units. The metabasites include eclogites (Omp + Grt + Qtz + Rt ± Ky and Zo, mineral abbreviations according to Whitney and Evans, 2010) and related clinopyroxene-garnet rocks without primary plagioclase (Cpx + Grt + Qtz + Rt ± Zo), as well as other types of rocks derived from the retrogression and mylonitization of the early high-P stages. The intermediate tectonic slice is made up of migmatitic felsic gneisses (mainly paragneisses), with frequent inclusions of high-P granulites (Cpx + Grt + Pl + Qtz + Rt ± Ky). Remnants of igneous protoliths are not preserved either in the lower or intermediate tectonic slices. The upper tectonic slice, however, contains migmatitic felsic gneisses and mafic layers derived from deformed and recrystallized gabbros with locally preserved relict igneous textures, reaching high-P granulite facies conditions. The progressive transformation from gabbros to high-P granulites (Na-Di + Grt + Pl + Qtz + Rt) has occurred in a series of different stages with a metamorphic peak at 13-17 kbar and 660-770ºC (Arenas and Martínez Catalán, 2002).

The metamorphic evolution of the Sobrado Unit as described in the literature indicates that felsic gneisses underwent differing degrees of partial melting after the metabasites reached their peak pressure. Consequently, the felsic gneisses are thought to have developed a regional foliation under amphibolite facies conditions, as did the amphibolic gneisses, "flaser" amphibolites, and fine-grained amphibolites. This metamorphic evolution is described by Arenas and Martínez Catalán (2002) as a clockwise P-T path, with a metamorphic peak of, at least, 15 kbar and >800ºC, followed by an isothermal decompression. This trajectory is interpreted to result from gravitational collapse of an overthickened orogenic wedge (Gómez-Barreiro et al., 2007; Ballèvre et al., 2014). Although some regional structures, such as the Fornás detachment (FD, Fig. 1; Gómez-Barreiro et al., 2007; Álvarez-Valero et al., 2014) or the Corredoiras detachment (CD, Fig. 1; Díaz García et al., 1999), have been related to this gravitational readjustment, no study has dealt with the development of the extensional fabrics in any detail. Overall, it is thought that the extensional flow has generated a pervasive thinning of the orogenic pile and that the preserved sequence of tectonic slices is strongly condensed.

## 3. Sample description and Methodology
### 3.1. Selected samples

Two representative samples (*JBP-71-15A* and *JBP-71-21*) were selected from two structurally separate but currently adjacent parts of the high-P/high-T Sobrado unit, within the Órdenes complex, for laser ablation (LASS-Laser Ablation Split Stream) analyses, including U–Pb geochronology and REE determinations. The samples locations are presented in Figure 2.

Sample *JBP-71-21* is a mylonitic fine-grained amphibolite, without any preserved igneous relics. Fine-grained amphibolites represent an advanced stage in the mylonitization of the metabasites. They appear as relatively thin layers inside the maffic rocks and dominate in a thick mylonitic layer (300

- 700m; Fig. 2; Arenas and Martínez Catalán, 2002; Benítez Pérez, 2017) located at the base of the upper tectonic slice. The sample comprises Hbl + Pl + Grt ± Cpx ± Bt ± Rt ± Ttn ± Ilm ± Zo ± Qtz (Fig. 3). Mylonitic foliation and lineation are defined by amphibole and plagioclase (Fig. 3). Garnet appears as subrounded porphyroclasts partially resorbed (Fig. 3). Titanite grains are parallel to the foliation and in textural equilibrium with plagioclase and amphibole. Rutile and ilmentite are always hosted as inclusions both in titanite and/or garnet (Fig. 3 c, d).

Sample *JBP-71-15A* is a granulite facies migmatitic paragneiss from the underlying intermediate tectonic slice. It comprises Qtz + Pl + Grt + Kfs + Ky + Bt + Ilm + Rt and shows microscopic scale textural evidence of partial melting. Temperature and pressure estimation ranges between 750-850ºC and 11-16 kbar for the anatectic fabric (Benítez-Pérez, 2017). Leucocratic domains with Qtz, Kfs and Pl, with evidence of plastic deformation define the foliation and Bt, Ky, Rt and Grt define linear aggregates resulting in a pervasive lineation (Fig. 4). Along the strained leucosomes, garnets show evidence of plastic deformation (Benítez Pérez, 2017) like (Fig. 4 a, f, g): sigmoidal, dumb-bell-shaped grains and pinch-and-swell microstructures (Ji and Martignole, 1994; Kleinschrot and Duyster, 2002; Passchier and Trouw, 2005). Zircon and monazite are found in different microtextural settings. Small-elongated prims of zircon are always found shielded whitin garnets (Fig. 4 g), while relatively larger zircon grains with irregular, elongated, sub-rounded shapes, appear across the fabric in leucosomes, biotite aggregates and even within kyanite crystals (Fig. 4 a,b,c,d,f). In few cases, elongated/sub-rounded zircons have been found as inclusions in garnets (Fig. 4 e). Monazite grains show elongated to sub-rounded grains located in Qtz-Kfs-Pl-Bt domains, which define the main foliation of the rock (Fig. 4 d,f,h).

### 3.2. Sample preparation

Sample preparation was carried out at the laboratories of the Universidad Complutense (Madrid). The rocks were crushed, pulverized and sieved to achieve a 0.1-0.5 mm grain size. Heavy minerals were concentrated using a Wilfley™ table. The non-magnetic minerals from this heavy fraction are then separated using a Frantz isodynamic magnetic separator. A final concentrated fraction is obtained using heavy liquids (methylene iodide, $CH_2I_2$).

Zircon (translucent, colourless or light brown), monazite (yellow) and titanite (colorless) grains are selected by handpicking, according to their external morphology viewed under a binocular microscope. All the grains collected were arranged separately in parallel rows, mounted on glass slide with a double-sided adhesive and set in epoxy resin. After the resin was cured, the surface was eroded using a wet abrasive silicon carbide abrasive paper (4000 grit) and polished with 0.3 $\mu$m aluminium oxide. The surface was then coated with gold, to avoid charging problems under the scanning electron microscope (SEM). Prior to isotopic analysis, cathodoluminescence images (CL) of zircon grains were taken on a JEOL JSM-820 SEM, and compositional maps of monazite grains were created on a JEOL Superprobe JXA-8900M microprobe (National Center for Electron Microscopy, Madrid). Secondary electron images (SE) were also taken to determine the exact location of the spots, identify the internal structure, and presence of inclusions and defects in zircon, monazite and titanite grains.

### 3.3. Mineral description

Titanite grains are generally rounded, with an average grain size of 100 $\mu$m, and irregular morphologies. Their secondary electron images reveal homogeneous compositions and the presence of solid inclusions. This grain size permits large spatial resolution analyses (50 $\mu$m) to be carried out. Monazite grains have a more variable grain size distribution, with an average of 60-70 $\mu$m. Their habit is irregular and they usually show rounded morphologies or broken grains. We carried out La, Th, Y, U and Nd compositional maps for every monazite grain in order to discover a compositional zonation that could be attributed to different growth events. Thorium zoning is the one that developed better and was taken into account to select the spots for isotopic analysis (Fig. 5), yet it never exceeds 30% of the grain.

Several spots were analyzed in monazite crystals with the greatest compositional contrasts to determine if they really represented different growth stages in the monazite grains.

Zircon grains from the paragneisses usually have scarce mineral inclusions and they can display a wide variety of morphologies, including irregular and sub-rounded shapes typical of metamorphic zircon, pristine elongated dipyramidal prisms interpreted as igneous in origin, and equigranular grains with abrasion signs with a probable detrital origin. Their length-to-width ratios vary between 3:1 and 2:1. Cathodoluminescence images (Fig. 6) are useful to relate the crystallization of parts of zircon crystals to specific igneous, metamorphic or deformational events (Corfu et al., 2003, Nasdala et al., 2003, Zeck et al., 2004). It is common to image a homogeneous xenocrystic core in zircon grains and even a less luminescent mantle in some grains (grain numbers 5, 6). The core aspect is mainly rounded, with irregular or angular shapes. In most of the zircon grains, the internal parts of the grains display an oscillatory zoning (grain numbers 33, 71), with different thickness, although in some cases, this zoning is faint (grain number 26). There are several grains with sector zones (grain numbers 26, 27) parallel to the zircon *c*-axis (Watson and Yan Liang, 1995) and even one case of soccerball zoning (grain number 82). The zoning usually appears to be partially truncated and surrounded by a discontinuous poorly luminescent rim (grain numbers 20, 79).

### 3.4. Analytical techniques

U/Th-Pb, REE and Hf analyses of zircon, titanite and monazite were carried out using the laser ablation split stream (LASS) at the University of California at Santa Barbara (UCSB). The samples were ablated using a Photon Machines 193 nm ArF excimer ultraviolet laser with a HelEx ablation cell coupled to a Nu Instruments Plasma high-resolution multi-collector inductively coupled plasma mass spectrometer (MC-ICP-MS) and either a Nu Instruments AttoM high-resolution single-collector ICP or an Agilent 7700S quadrupole ICP-MS (Kylander-Clark et al., 2013). This installation allows the simultaneous isotopic and compositional (REE) analysis to be carried out. The laser spot diameter was 20 $\mu$m for zircon, 7-10 $\mu$m for monazite (Košler et al., 2001) and 50 $\mu$m for titanite (Stearns et al., 2016), resulting in pit depths between 6 $\mu$m for monazite and 30 $\mu$m for titanite. The laser has a fluence of ~1 J/cm$^2$ and was fired twice to remove common Pb from the sample surface. This material was allowed to wash out for 15 s, prior to the material being ablated at 3 Hz for 20 s for analysis. On the ICP-MS, the masses $^{204}$Pb+Hg, $^{206}$Pb, $^{207}$Pb, and $^{208}$Pb were measured using ion counters, and the masses $^{232}$Th and $^{238}$U were measured using Faraday detectors.

The U/Th-Pb standardization for monazite was carried out using sample 44069 (Aleinikoff et al., 2006) as the primary reference material (RM), whereas the Bananeira sample was employed as primary RM for trace element corrections (Kylander-Clark et al., 2013; Palin et al., 2013). Additionally, FC-1 (Horstwood et al., 2003), Trebilcock (Tomascak et al., 1996) and Bananeira were also used as secondary monazite RM, allowing $^{206}$Pb/$^{238}$U ages to be within 2% of their accepted values. U-Pb proportions in titanite were corrected using Bear Laken (Aleinikoff et al., 2007) and Y1710C5 (Spencer et al., 2013) as RM. 91500 (Wieldenbeck et al., 1995) and GJI (Jackson et al., 2004) were used as RM for zircon, both for isotopic composition and trace element calibrations. Radiogenic lead versus common lead ($^{207}$Pb/$^{206}$Pb) measurements require up to 2% additional external error attributable either to variation count statistics, or ablation signal stability (Spencer et al., 2013; Hacker et al., 2015b). These external errors were incorporated into the data in the experiments.

The Iolite plug-in v. 2.5 (Paton et al., 2011) from the Wavemetrics Igor Pro software was used to improve and reduce the analyses (Hacker et al., 2015). The isotopic ratios $^{238}$U/$^{206}$Pb and $^{207}$Pb/$^{206}$Pb for each analysis were plotted on Tera-Wasserburg diagrams using Isoplot and Topsoil programs (Ludwig, 2012; Zeringue et al., 2014). All date uncertainties are reported at the 95% confidence interval, assuming

a Gaussian distribution of measurement errors. Zircon, titanite, and monazite REE analyses were
normalized against the McDonough and Sun (1995) chondrite values.

**4.  Results**
### 4.1.  Titanite (amphibolite, upper tectonic slice)

Fifty-one titanite analyses were projected onto a Tera-Wasserburg concordia plot (Fig. 7 a).
After a preliminary evaluation, twelve analyses were rejected due to either high common Pb or high
discordance (>10%) and were considered no further (Table 1). The remaining analyses define a Pb/U
semi-total isochron between the common or initial Pb ($^{207}$Pb) and radiogenic Pb ($^{206}$Pb) (Ludwig, 1998).
The good fit of the isochron confirms the chemical homogeneity of the data (Stearns et al., 2016) and it
intercepts the concordia at 364.8 $\pm$ 4.5 Ma (2σ). Titanite chondrite-normalized REE analyses are detailed
in Table 1 and are shown in Fig. 7 b. Titanite REE patterns are convex upwards, relatively flat, with slight
LREE depletions versus HREE with respect to chondrite. They generally lack a europium anomaly (Eu*),
but some analyses show a non-distinctive positive or negative anomaly.

### 4.2.  Monazite (paragneiss, middle tectonic slice)

For monazite U/Th-Pb geochronology, we used the thorium zoning in monazite grains to select
the analytic spots. As shown in Figure 5 there are no significant age differences between spots or zones
with different Th chemical concentrations in a single grain. The obtained REE patterns are also very
similar, and the mismatch at the HREE is probably due to either uncertainties in measurement because of
the lower counts or interference effects of intermediate rare-earth oxides (Holder et al., 2015).

Data from seventy-six U/Th-Pb monazite analyses are shown in Table 2 and displayed using a
Tera-Wasserburg concordia plot (Fig. 8 a). Four of these analyses, not related to chemical zoning, were
discarded due to common Pb loss and were considered no further. The remaining analyses form a single
population (mean square of weighted deviation; MSWD = 0.48) centered on a concordia age of 382.5 $\pm$
1.0 Ma (2σ). Monazite geochemistry is shown in Figure 8b. REE patterns analyzed show an LREE
enrichment, HREE depletion and negative Y anomalies with respect to chondrite with little variation
within or between samples.

### 4.3.  U-Pb zircon (paragneiss, middle tectonic slice)

Eighty-three analyses were performed on eighty zircon grains from the Sobrado paragneiss
(Table 3). In a preliminary assessment of the data two analyses were rejected due to analytical errors
(grain numbers 8, 36). Additionally, 23 analyses yielded a discordance higher than 10% and will not be
further considered. The remaining 58 zircon analyses are shown on a Wetherill concordia plot (Fig. 9 a).
The $^{207}$Pb/$^{206}$Pb ages older than 1000 Ma are presented in a probability density plot (Fig. 9b). Most of the
old ages are distributed between 1880 and 2200 Ma, with two peaks at 2030 and 2100 Ma. Three ages are
older, around 2600 Ma, and there are also two analyses around 1300 Ma. The $^{206}$Pb/$^{238}$U ages younger
than 1000 Ma are ploted in a Tera-Wasserburg concordia plot (Fig. 10 a) and a probability density plot
(Fig. 10 b). The data are continuously distributed between 589 and 380 Ma and attending to their CL
texture, ages from 589 to 510 Ma are obtained mainly from internal areas with oscillatory or sector
zoning (e.g., 33, 27 and 62, see Fig. 6); whereas, ages from 510 to 380 Ma correspond to discordant rims
that have homogeneous CL signal, except for some grains (e.g., 10, 11 and 26, see Fig. 6) that are cores
with oscillatory or sector zoning.

### 4.4.  REE zircon (paragneiss, middle tectonic slice)

The chondrite-normalized REE patterns of the zircons with ages older than 600 Ma are shown in
Figure 11. In general, this group has REE patterns with a pronounced fractionation from light to heavy
REE and two anomalies in Ce (positive) and in Eu (negative). There are only three analyses that diverge
from this trend and show a flat HREE pattern.

For the younger analyses, the chondrite-normalized REE patterns have different features

depending on the age (Fig. 12). The oldest ages (589-510 Ma) show a tight HREE fractionation, variable Eu anomaly, and a pronounced Ce anomaly. The youngest ages (510-380 Ma) has a variable HREE slope, whereas the Eu anomaly is more regular, and the Ce anomaly is less marked.

When we plot age versus Hf, Yb/Gd and Eu/Ey* (Figs. 13 a, b, c), the group of oldest ages (589-510 Ma) does not define any apparent trend. In sharp contrast, in the 500-380 Ma group it is remarkable not only the good correlation between age and composition, but also the divergent evolution depending of the age (grey arrows). Hafnium, Yb/Gd and Eu/Eu* increase from 500 to ~430 Ma while there is a striking decrease from ~420 to 380 Ma. Finally, we have used the U/Ce-Th graph proposed by Bacon et al. (2012) to discriminate between metamorphic and igneous zircon.

## 5. Discussion
### 5.1. Zircon geochronology: inheritance and age of magmatism

The presence of zircon inclusions shielded in garnets (e.g. Fig. 4 e and g) is compatible with the preservation of magmatic, metamorphic or detrital inheritances during the synkinematic partial melting recorded in these rocks (Benítez Pérez, 2017). As shown in Fig 12, the majority of analyzed zircon grains older than 600 Ma have fractionated REE patterns that are characteristic of igneous zircon (Hoskin and Schaltegger, 2003, Whitehouse and Platt, 2003; Hanchar and Westrenen, 2007; Grimes et al., 2015). Only three analyses show a flat HREE pattern that can be related to the presence of garnet and interpreted as metamorphic zircon. The provenance of these zircon grains older than 600 Ma in the Sobrado migmatitic paragneisses (Fig. 9 b), is probably similar to those reported in the intermediate-P units of NW Iberia upper allochthons like the Betanzos unit (Fuenlabrada et al., 2010) and Cariño gneisses (Albert et al., 2015). We obtained two Mesoproterozoic ages, between 1.2 and 1.4 Ga. Similar ages are also found in the Parautochthonous (Díez Fernández et al., 2012), in the basal allochthonous units (Díez Fernández et al., 2010) and, to a lesser extent, in the intermediate-P units of NW Iberia (Albert et al., 2015). These inherited zircons, although scarce (Fernández-Suárez et al., 2003), likely have their origin in rocks derived from Saharan and Arabian-Nubian cratons, and presumably transported during the Cadomian orogeny (e.g., Martínez Catalán et al., 2004). Paleoproterozoic populations range from 1.8 to 2.2 Ga, clustered at 2.1 Ga, are also common in the Allochthonous complexes (Fernández-Suárez et al., 2003) and their origin likely involves materials generated or reworked during the Eburnian orogeny (e.g., Egal et al., 2002; Ennih and Liegeois, 2008) from the West African craton (Peucat et al., 2005). Finally, the Archean population in the Sobrado paragneisses ranges from 2.5 to 2.8 Ga, and it is likely related to intrusive events in the Western Reguibat Shield (Schofield et al., 2012) and the northern part of the West African craton (Albert et al., 2015), with some reworking processes of juvenile rocks formed at ca. 3.0 Ga (Potrel et al., 1998).

Based on CL images, ages and zircon composition, we can determine that the youngest zircon with magmatic origin is grain number 34 (Fig. 6). The number obtained in this grain is comparable to other maximum depositional ages obtained from similar units in the NW Iberia allochthonous complexes, such as the intermediate-P Betanzos uppermost unit (530-510 Ma, Fuenlabrada et al., 2010), and the intermediate-P Cariño uppermost unit (~510 Ma, Albert et al., 2015).

In order to get more insight into the meaning of the data younger than 600 Ma, we have plotted these ages versus the Th/U ratios (Fig. 14). Analyses that yielded ages between 589 and 510 Ma cluster together with Th/U values higher than 0.3. The REE patterns displayed in this cluster are consistent with a magmatic origin. In the age versus Hf, Yb/Gd and Eu/Eu* plots for this group (Fig. 13 a, b, c), the absence of a trend suggests that the different zircon grains are not connected by a fractional crystallization process (e.g., Barth and Wooden, 2010). In the U/Ce-Th graph proposed by Bacon et al. (2012) to discriminate between metamorphic and igneous zircon. In this diagram, zircon with ages between 589 and 510 Ma plot in the igneous zircon field (Fig. 13 d).

From the first group of data between 589 and 510 Ma, we can extract two ages, 546 ± 5 Ma out of six analyses, and 526 ± 3 Ma from 14 analyses (Fig. 15). These magmatic ages are also recognized in

other well-characterized high-P/high-T units of the allochthonous complexes (Peucat et al., 1990; Santos Zalduegui et al., 2002; Castiñeiras et al., 2010), and they are related to a magmatic arc creation around the periphery of Gondwana (Abati et al., 1999, 2007).

## 5.2. Evolution of metamorphism of the Sobrado migmatitic paragneiss based on zircon composition

Extracting ages from a dataset where the data are evenly distributed from 500 to 380 Ma is a challenging task. When such smear in the age sorting happens, if we can rule out analytical error, we have three possibilities (e.g., Castiñeiras et al., 2010), namely, (i) the correct age is the youngest and the dispersion is related to zircon inheritance, (ii) the real age is the oldest and the spread is caused by lead loss, or, (iii) the age range is recording some kind of protracted geological event. Taking into account the CL texture of the youngest spots, we can remove the first possibility as the majority of analyses were performed in zircon rims. Furthermore, maximum depositional ages obtained from similar units from the Allochthonous complexes preclude their interpretation as inheritance. The second possibility is plausible, considering the complex metamorphic evolution and the high grade attained by these rocks. However, a young lead loss episode should have affected also the inherited zircon ages, and the presence of various old age peaks (e.g., ~2000, 546 and 536 Ma) suggests that they did not experienced lead loss. However, we can argue that limited lead loss occurred in some grains that have similar composition to the 589-510 Ma group (Th/U> 0.15), but younger ages (between 502 and 468 Ma). This group of outlier analyses (#10, 11, 26 and 63) could have experienced a decoupling between their actual age and their composition (e.g., Flowers et al., 2012). Finally, we favor the last option when we take into account the strong correlation observed between age and zircon composition (Figs. 13 a, b, c and 14).

This group of young analyses defines a trend with negative correlation from 500 to 380 Ma and Th/U values from 0.01 to 0.13. This correlation between the age and the composition of zircon suggests that the age dispersion is related to an actual geological process and is not caused by lead loss. Furthermore, even though the Th/U ratio shows a consistent evolution from 510 to 380 Ma, the rest of the proxies considered (Hf, Yb/Gd and Eu/Eu*, Figs. 13 a, b, c) point to a two-stage evolution, and the characteristics of the REE patterns are compatible with a metamorphic origin (Chen et al., 2009, 2010; Rubatto et al., 2009; Peters et al., 2013; Stipska et al., 2016).

This scenario is congruent with the presence of two metamorphic events in the HP-HT units, one at ~490-480 Ma and other at ~390-380 Ma (e.g., Fernández-Suárez et al., 2002, 2007). The increase in Yb/Gd values (Fig. 13 b), related to the slope of the HREE, in zircons aged from 502 to 430 Ma is congruent with a higher availability of HREE in the rock. As garnet is the most important HREE reservoir in metamorphic rocks, we argue that this behaviour is the record of the progressive destabilization of garnet in a decompressive path from HP-HT conditions. The increase observed in the Eu/Eu* ratio is consistent with a progressive destabilisation of plagioclase, which is the main europium reservoir in rocks (Barth and Wooden, 2010; Castiñeiras et al., 2011).

The sharp decrease observed in the Yb/Gd ratio from 420 to 380 Ma, is probably related to a new event of garnet growth (Rubatto et al., 2006; Stipska et al., 2016), i.e., the second HP-HT event. The evolution in the Eu/Eu* ratio suggests that this event took place under granulite facies conditions, as plagioclase was present to pump out all the available europium.

## 5.3. Monazite and titanite geochronology: Age of the youngest metamorphism in the Sobrado antiform

The youngest zircon data recorded (380 ± 4 Ma) is coherent with the monazite concordia age (382 ± 1 Ma) in the migmatitic paragneises of the Middle tectonic slice. Besides, both monazite and irregular/sub-round zircons share their microtextural setting along the migmatitic fabric, pointing to a coeval character with partial melting at relatively high-P. Furthermore, the chemical profiles observed in

the monazite suggest simultaneous crystallization of this mineral with garnet (Rubatto, 2002; Rubatto et al., 2006; Mottram et al., 2014; Holder et al., 2015). Negative Eu anomalies indicate a preferential incorporation of europium to feldspars, in particular to K-feldspar, during melt crystallization (Buick et al., 2010; Rubatto et al., 2013). These characteristics are compatible with monazite crystallization in a single pulse (MSWD <1) in the presence of garnet, supporting its metamorphic origin. This Middle Devonian age can be interpreted to represent the *minimum* age of the youngest metamorphic event in Sobrado unit, which reached high-P granulite facies (Fernández-Suárez et al., 2007; Ordóñez Casado et al., 2001). It is suggested that monazite captures the onset of the exhumation process in the migmatitic paragneisses (Holder et al., 2015).

Titanite crystallization is synkinematic with the mylonitic fabric of the fine-grained amphibolites located at the base of the Upper tectonic slice. The growth of titanite in amphibolitic conditions is supportted by the umbrella-shaped REE patterns shown in Fig. 7b, typical of titanite coexisting with amphibole (Mulrooney and Rivers, 2005; Chambefort et al., 2013; Lesnov, 2013). A Late Devonian age (~365 Ma) could be related to the onset of retrograde metamorphic conditions during the development of the shear zone, and suggests a prolongated exhumation process, reaching amphibolite facies. The Late Devonian age lies close to the Ar-Ar exhumation ages of the uppermost units in the Órdenes complex as recorded in the Corredoiras detachment (376 ± 2.0 Ma in hornblende, Dallmeyer et al., 1997), or in the Ponte Carreira Detachment (371 ± 4.0 Ma in muscovite, Gómez Barreiro et al. 2006).

The U-Pb zircon age for the onset of the oldest metamorphic event was estimated using the TuffZirc method, developed by Ludwig and Mundil (2002), which calculates the median by choosing the largest set of concordant analyses that are statistically coherent. The best estimate obtained for this event is 489.58 (+ 12.15 - 6.76) Ma, obtained by pooling together only six of sixteen analyses (Fig. 16). The 510-380 Ma zircon aliquot shows a clear correlation between its cathodoluminescence texture and its geochemistry and could be also related to shielded population of "metamorphic" zircon grains within garnets (Fig. 4 e). The age recorded in the migmatitic paragneisses is thought to correspond to a metamorphic event, dated in the Early Ordovician (~490 Ma), and is in very good agreement with upper high-P/high-T dates of equivalent units carried out during previous studies (Kuijper, 1979; Peucat et al., 1990; Fernández-Suárez et al., 2002, 2007). This age also coincides with those obtained from intermediate-P units, where large plutons were emplaced and there is a lack of later high-P/high-T metamorphism during the Devonian. The westernmost upper intermediate-P units of the Órdenes Complex underwent a granulite-facies metamorphism dated between ca. 500 and 485 Ma, contemporaneous with the intrusion of massive gabbros and granodiorites related to Cambrian magmatic arc activity (Abati et al. al., 1999, 2003, 2007, 1999; Andonaegui et al., 2002, 2012, 2016; Castiñeiras et al., 2002, 2010). The granulite-facies metamorphism is associated with heating produced by the intrusions, accompanied by a quick burial, almost coeval with igneous emplacement (Abati et al., 2003; Castiñeiras, 2005; Fernández-Suárez et al., 2007).

Clearly, the metamorphic event recorded in some zircons is pre-Variscan and it is therefore independent of the high-P/high-T granulite-facies metamorphism that occurred during the Early-Middle Devonian that has been identified in the underlying upper units, such as in Sobrado with 660-770°C and 13-17 kbar (Arenas and Martínez Catalán, 2002), or 750-850°C and 12-15 kbar (Benítez-Pérez, 2017). The pre-Variscan metamorphism was probably followed by a decompression stage, associated with partial melting (Fernández-Suárez et al., 2002). Later on, HP-HT Devonian metamorphism occurred, during which exhumation through an isothermal decompression lead to partial melting in paragneisses and basic granulites (Fernández-Suárez et al., 2007). As the zircon composition and microstructural setting clearly suggests, the notable slope observed in the TuffZirc plot from 489 to 380 Ma (Fig. 16) is the result of these exhumation, burial and new exhumation processes accompanied by partial melting, in which the shielding role of garnet has played an important role.

## 6. Conclusions

This study provides new age constraints on the processes that have affected the Sobrado unit, part of the Órdenes Complex, and allows some correlation with events recognized in other parts of the allochthonous high-P/high-T complexes of NW Iberia. Titanite, monazite and zircon dating, together with REE analyses have been combined together in these rocks for the first time in order to carry out a geochronological investigation of the amphibolites and paragneisses.

According to the analyses, the youngest ages recorded by the metamorphic zircons are coherent with the concordia monazite age obtained from seventy-six analyses in the paragneisses. The microtextural setting of both metamorphic zircon and monazite along the HP-HT main foliation support that interpretation. Therefore, the Middle Devonian age (~380 Ma) represents the minimum age of the last Sobrado metamorphic event under high-P granulite-facies conditions and represents the first stages of the Variscan orogeny in this part of Iberia. Dating of metamorphic titanite in the amphibolite yields a Late Devonian age (~365 Ma) and is associated with very homogeneous REE patterns suggesting the prolongation of the exhumation process in the Sobrado unit, reaching amphibolite-facies metamorphic conditions. In zircon, there is a strong relationship between their textures, as seen in cathodoluminescence images (CL), REE patterns and $^{206}Pb/^{238}U$ ages. Metamorphic zircon defines an Early Ordovician age (~490 Ma) although showing a large dispersion. This date is linked to the first pre-Variscan granulite-facies metamorphism seen in in Sobrado unit under intermediate-P conditions, and it is interpreted to be related to the intrusion of basic and intermediate composition rocks, and coeval with burial in a magmatic arc context. Microstructural analysis of zircon, monazite and titanite provide complementary microtextural context to understand the origin of this population mixture. In situ dating should be conducted to confirm some textural relationships in the future.

The maximum depositional age of the Sobrado unit is suggested to be late Cambrian based on the age of the youngest inherited zircon (511 Ma). From the youngest set of inherited zircon, two ages can be obtained (546 and 526 Ma) , pointing to the formation of a peri-Gondwana magmatic arc. The protoliths of inherited zircon older than 1000 Ma from the Sobrado unit are found in other Iberian complexes and are thought to be related to sources mainly in the West African craton.

*Data availability*

The data are not publicly accessible

*Supplement*

There is no supplement related to this article.

*Author contributions*

JMBP, PC, JGB and JRMC contributed equally to the field, experimental and elaboration of the manuscript. AKC contributes to U-Pb-REE acquisition and data reduction and RH participated in the writing of the text and the geological interpretation.

*Competing interests*

The authors declare that they have no conflict of interest.

**Acknowledgements**

This paper has been funded by the research projects CGL2011-22728 and CGL2016-78560-P of the Spanish Ministry of Economy, Industry and Competitiveness, as part of the National Program of Projects in Fundamental Research. JMBP appreciate financial support by the Spanish Ministry of Economy, Industry and Competitiveness though the Formación de Profesional Investigador grant FPI 2013-2016 (BES-2012-059893). JGB appreciates financial support by the Spanish Ministry of Science

and Innovation through the IEDI-2016-00691 fellowship. We kindly appreciate excelent reviews from M. Zucali, F. Rossetti, P. Ayarza and an anonymous referee which contributed to improve the manuscript.

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

FIGURE CAPTIONS

Figure 1. (a) Simplified map of the Variscan orogen in Europe with the location of the Órdenes Complex.
(b) Map of the Órdenes Complex with indication of the main units and tectonic contacts. Extensional
detachments are labeled: BPSD, Bembibre-Pico Sacro system; CD: Corredoiras; FD, Fornás; PCD, Ponte
Carreira. Fig. 2 location indicated.

Figure 2. Geological map of the study area and location of the samples, modified from Arenas and
Martínez Catalán (2002): (a) Location of the Órdenes complex within the Iberian massif. (b) Sobrado
Unit map, indicating units and tectonic slices. (c) Cross-section in WNW-ESE direction and (d) SW-NE
and SSW-NNE direction of the Sobrado antiform. Sample location are indicated.

Figure 3. Fine grained amphibolites (sample JBP-71-21) from the basal mylonitic band in the Sobrado
Upper tectonic slice. (a) Hornblende-plagioclase mixtures and titanite with preferred orientation, defining
the mylonitic fabric. (b) Main fabric surrounding a garnet porfiroclast. Note titanite is the Ti phase in the
mylonitic fabric but ilmenite grains have been preserved inside the garnet. The position of (c) is indicated
by a white dashed square. (c) Garnet inclusions from (b) where ilmenite inclusions show a progressive
breakdown into titanite (Ilm>Ttn). Note ilmenite inclusions inside pristine garnet areas show no evidence
of transformation. (d) Inclusions of rutile inside titanite and partial transformation of plagioclase into
zoisite. All micrographs in plane-polarized light (PPL). (Mineral abbreviations after Whitney and Evans,
2010; Hbl: hornblende, Ttn: titanite, Ilm: ilmenite, Rt: rutile, Pl: plagioclase, Zo: zoisite, Grt: garnet).

Figure 4. Granulite facies migmatitic paragneiss from the Sobrado Middle tectonic slice (Fig. 2). (a)
General microstructure where leucosome bands and preferred orientation of elongated garnets, biotite,
rutile and kyanite define the foliation. (b) Zircon elongated grain in recrystallized leucosome domain.
Main inclusions in garnet include rutile, ilmenite and zircon (PPL) and (c) Cross-polarized ligth (CPL).
(d) Monazite and zircon subrounded grain in the leucosome domain. (e) Subrounded elongated grain of
zircon included in a garnet. (f) Plastically deformed garnet (sigmoidal grain) in a leucosome. Zircon and
monazite are found in the quartz-rich area around the garnet. (g) Sigmoidal garnet in a leucosome.
Inclusions of prismatic and bipiramidal zircons and rutile are observed. (h) Monazite grain in a garnet
pressure shadow with biotite. Ky: kyanite, Bt: biotite, Qtz: quartz, Kfs: K-felspard, Pl: plagioclase, Rt:
rutile, Zrn: zircon, Mnz: monazite, Grt: garnet.

Figure 5. (a) Monazite grain compositional maps in paragneiss with a 30% thorium variation. Location
and spot numbers (46 and 47) are indicated, as well as the $^{206}Pb/^{238}U$ age and error ($\pm 2\sigma$). (b)
Chondrite-normalized rare earth element (REE) patterns for the same monazites in (a).

Figure 6. Cathodoluminescence (CL) images with the location of the analyzed spots for selected zircon
grains.

Figure 7. (a) Tera-Wasserburg diagram showing distribution of analysed titanites (n = 51) from Sobrado
amphibolite (JBP-71-21). The rejected analyses are represented by gray ellipses. The ellipses represent
the $^{207}Pb/^{206}Pb$ and $^{238}U/^{206}Pb$ errors ($\pm 2\sigma$). (b) Chondrite-normalized rare earth element (REE) patterns
for the same titanites.

Figure 8. (a) Tera-Wasserburg diagram showing distribution of analysed monazites (n = 76) from
Sobrado paragneiss (JBP-71-15). The rejected analyses are represented by gray ellipses. The ellipses
represent the $^{207}Pb/^{206}Pb$ and $^{238}U/^{206}Pb$ errors ($\pm 2\sigma$). (b) Chondrite-normalized rare earth element
(REE) patterns for the same monazites.

Figure 9. Concordia plot (a) including all zircon with concordance >90% from sample JBP-71-15 (Sobrado migmatitic paragneiss), and (b) age histogram and probability density plot for ages older than 1000 Ma.

Figure 10. Tera-Wasserburg diagram (a) for the analyses between 589 and 380 Ma, and, (b) age histogram and probability density plot for the same ages.

Figure 11. Chondrite-normalized plots for inherited zircon older than 1000 Ma.

Figure 12. Chondrite-normalized plots for zircon between 589 and 380 Ma.

Figure 13. (a) Hafnium versus age, (b) Yb/Gd versus age, (c) Eu/Eu* versus age, and (d) U/Ce versus Th for zircon analyses between 589 and 380 Ma.

Figure 14. Th/U ratio versus $^{206}Pb/^{238}U$ ages for the zircon analyses from 589 to 380 Ma. Analysis 63 (510 Ma) is not represented as it has an anomalous value (6.59).

Figure 15. Weigthed average obtained from magmatic ages distributed between 589 and 510 Ma.

Figure 16. Age of the onset of the oldest HP-HT metamorphic event obtained using the TuffZirc algorythm.

Table 1: U-Th_Pb+REE Titanite_McD_S

Table 2A: U-Th_Pb+REE Monazite_McD_S

Table 3: U-Th_Pb Zircon sorted by age

Table 4A:  REE Zircon_McD_S sorted by age

Table 4B: REE Zircon_McD_S sorted by age

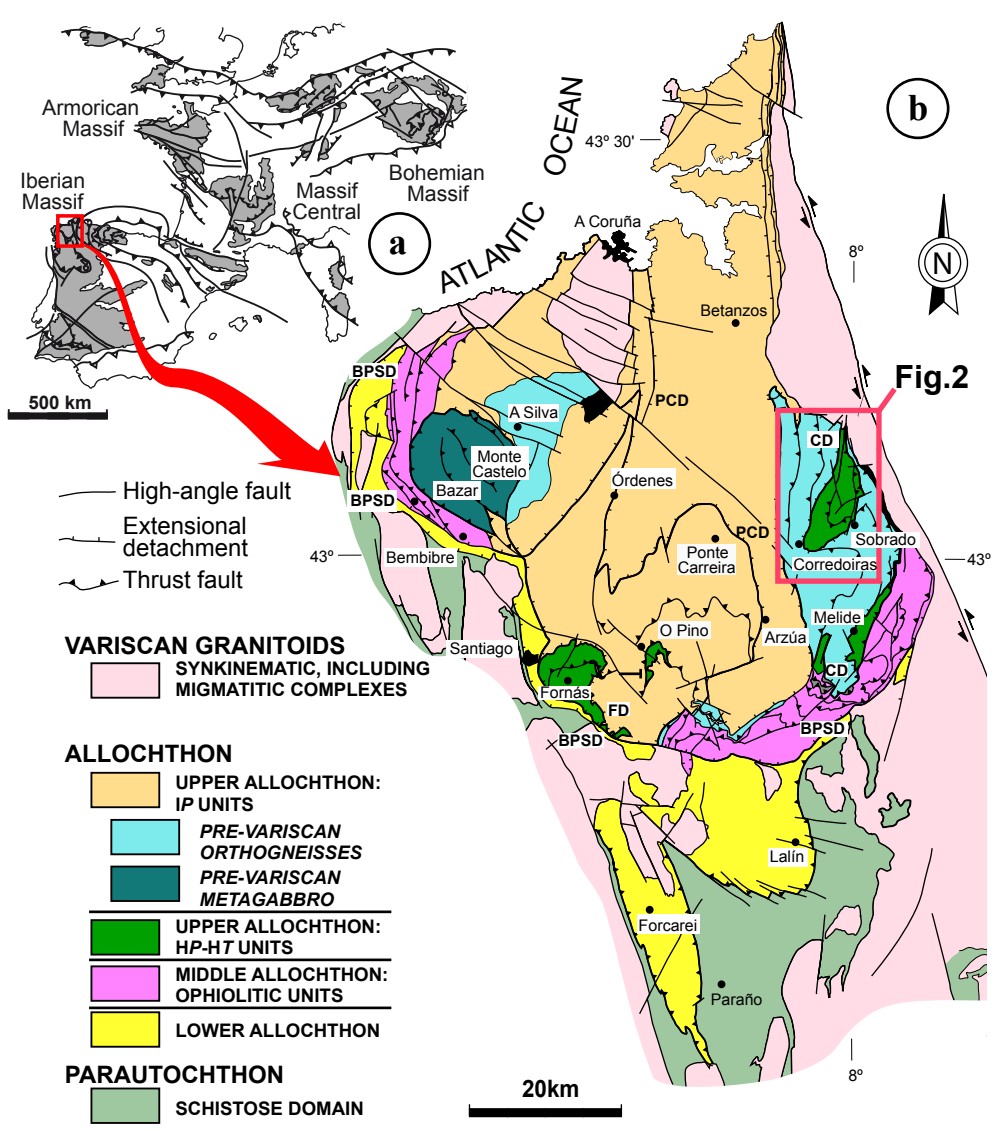

Figure 1: a) Simplified map of the Variscan orogen in Europe with the location of the Órdenes Complex. b) Map of the Órdenes Complex with indication of the main units and tectonic contacts. Extensional detachments are labeled: BPSD, Bembibre-Pico Sacro system; CD: Corredoiras; FD, Fornás; PCD, Ponte Carreira. Fig. 2 location indicated.

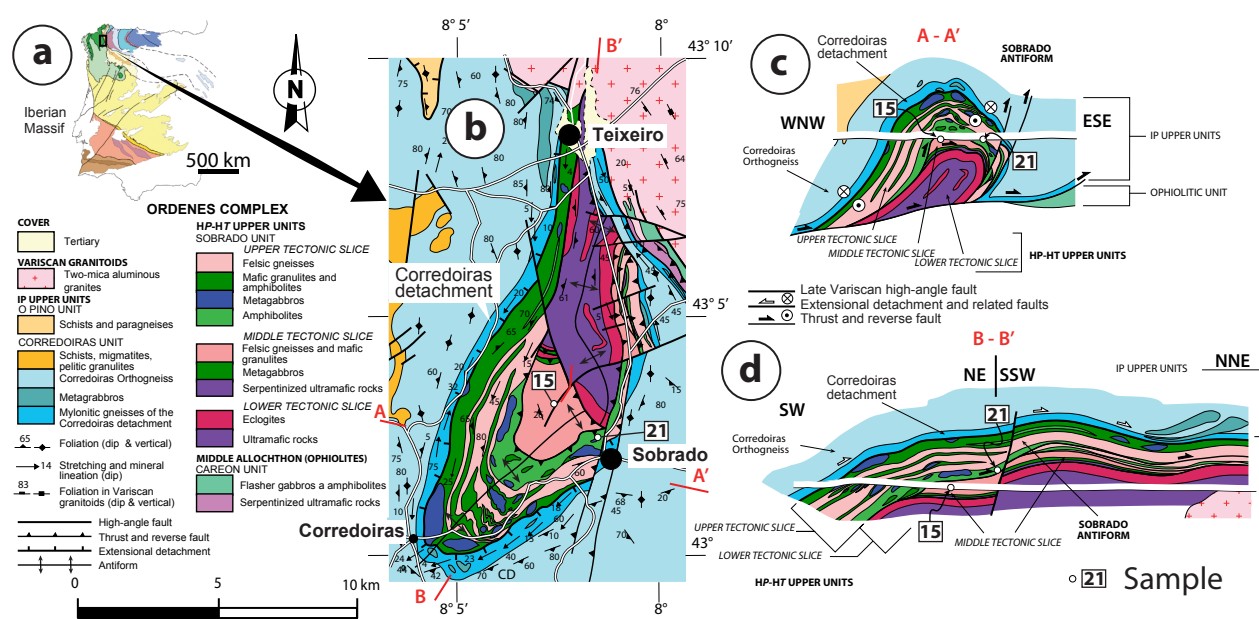

Figure 2. Geological map of the study area and location of the samples, modified from Arenas and Martínez Catalán (2002): (a) Location of the Órdenes complex within the Iberian massif. (b) Sobrado Unit map, indicating units and tectonic slices. (c) Cross-section in WNW-ESE direction and (d) SW-NE and SSW-NNE direction of the Sobrado antiform. Sample location are indicated.

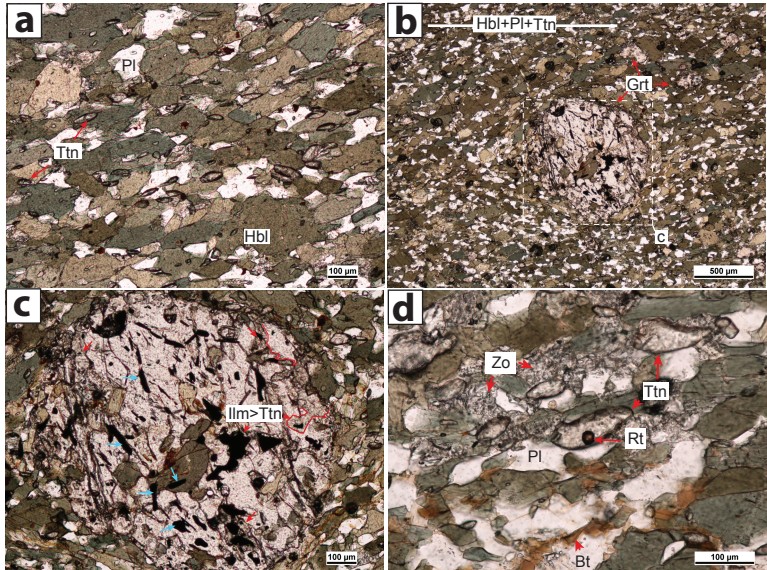

Figure 3. Fine grained amphibolites (sample JBP-71-21) from the basal mylonitic band in the Sobrado Upper tectonic slice. (a) Hornblende-plagioclase mixtures and titanite with preferred orientation, defining the mylonitic fabric. (b) Main fabric surrounding a garnet porfiroclast. Note titanite is the Ti phase in the mylonitic fabric but ilmenite grains have been preserved inside the garnet. The position of (c) is indicated by a white dashed square. (c) Garnet inclusions from (b) where ilmenite inclusions show a progressive breakdown into titanite (Ilm>Ttn). Note ilmenite inclusions inside pristinte garnet areas show no evidence of transformation. (d) Inclusions of rutile inside titanite and partial transformation of plagioclase into zoisite. All micrographs in plane-polarized light (PPL). (Mineral abbreviations after Whitney and Evans, 2010; Hbl: hornblende, Ttn: titanite, Ilm: ilmenite, Rt: rutile, Pl: plagioclase, Zo: zoisite, Grt: garnet).

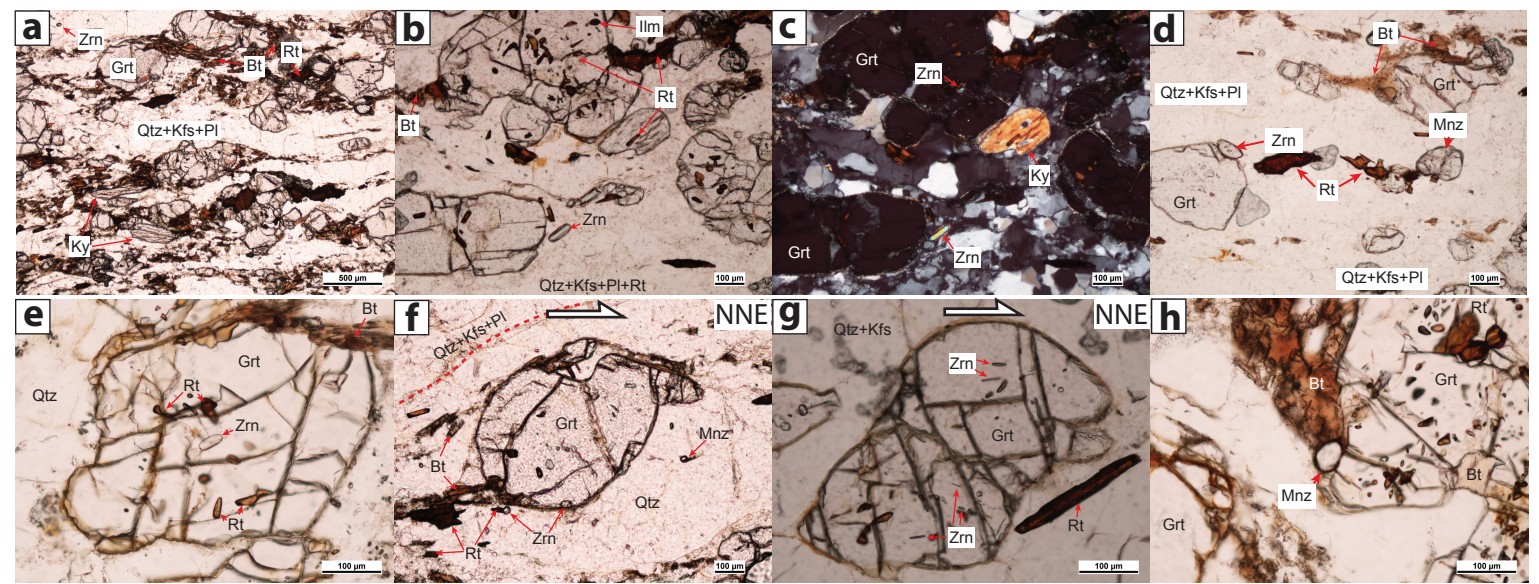

Figure 4. Granulite facies migmatitic paragneiss from the Sobrado Middle horse (Fig. 2). (a) General microstructure where leucosome bands and preferred orientation of elongated garnets, biotite, rutile and kyanite define the foliation. (b) Zircon elongated grain in recrystallized leucosome domain. Main inclusions in garnet include rutile, ilmenite and zircon (PPL) and (c) Cross-polarized ligth (CPL). (d) Monazite and zircon subrounded grain in the leucosome domain. (e) Subrounded elongated grain of zircon included in a garnet. (f) Plastically deformed garnet (sigmoidal grain) in a leucosome. Zircon and monazite are found in the quartz-rich area around the garnet. (g) Sigmoidal garnet in a leucosome. Inclusions of prismatic and bipiramidal zircons and rutile are observed. (h) Monazite grain in a garnet pressure shadow with biotite. Ky: kyanite, Bt: biotite, Qtz: quartz, Kfs: K-felspard, Pl: plagioclase, Rt: rutile, Zrn: zircon, Mnz: monazite, Grt: garnet.

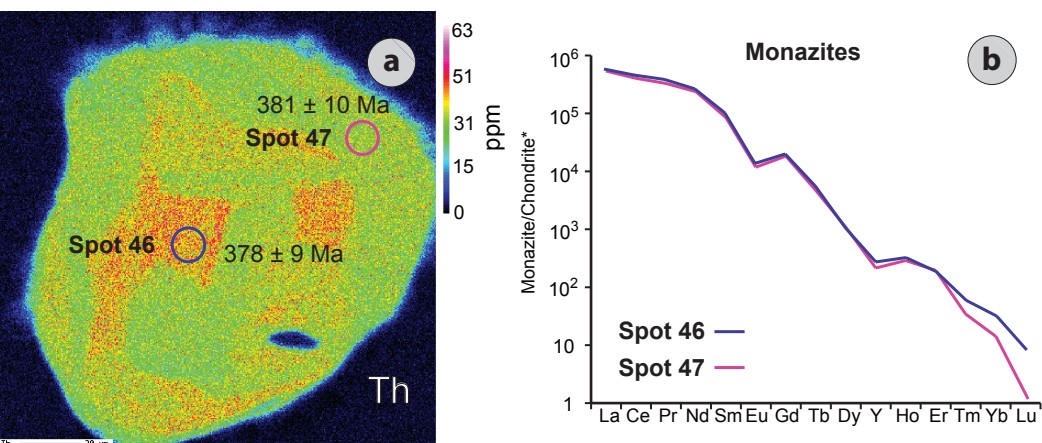

Figure 5. (a) Monazite grain compositional maps in paragneiss with a 30% thorium variation. Location and spot numbers (46 and 47) are indicated, as well as the [206]Pb/[238]U age and error (±2σ). (b) Chondrite-normalized rare earth element (REE) patterns for the same monazites in (a).

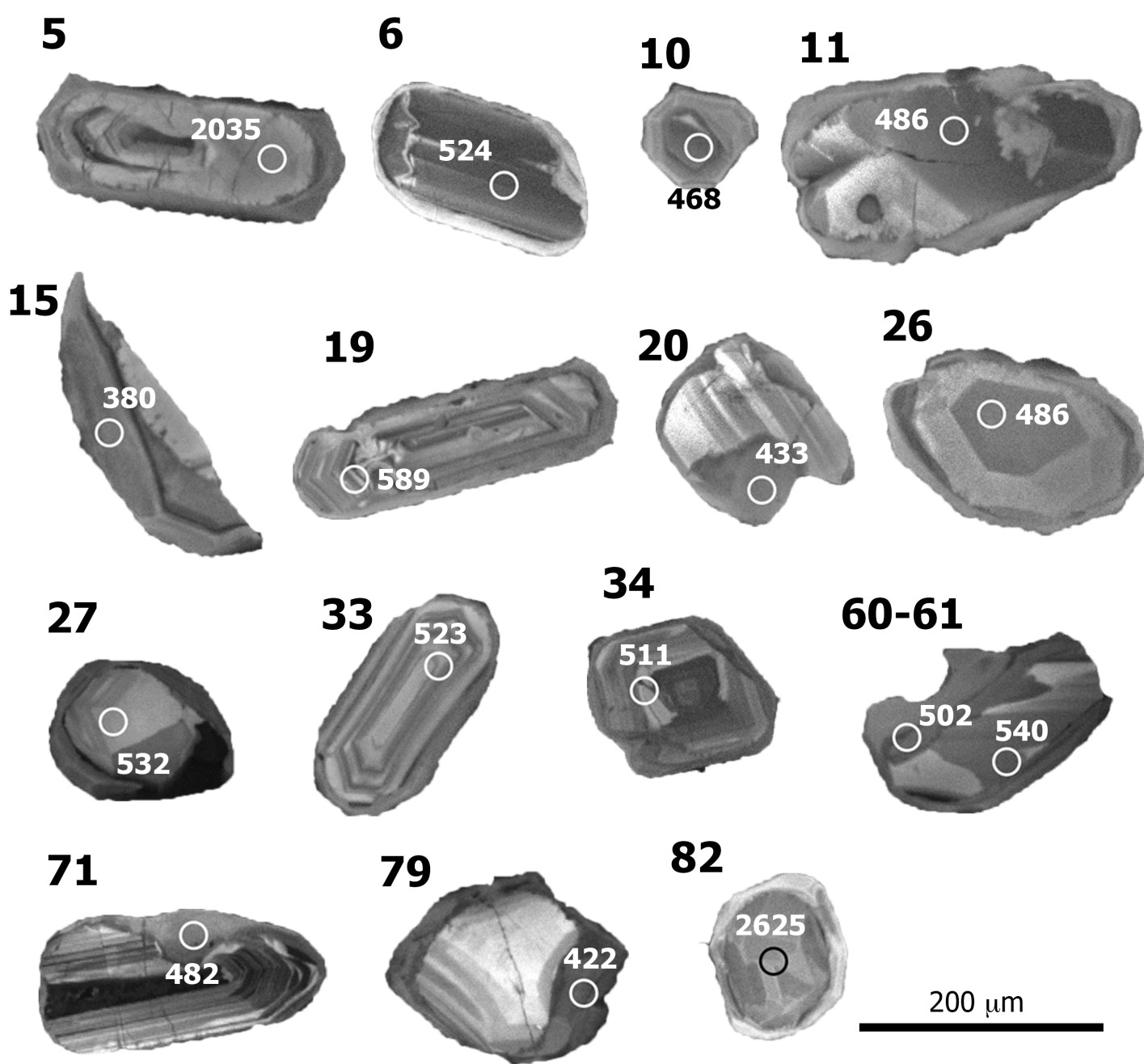

Figure 6: Cathodoluminescence (CL) images with the location of the analyzed spots for selected zircon grains.

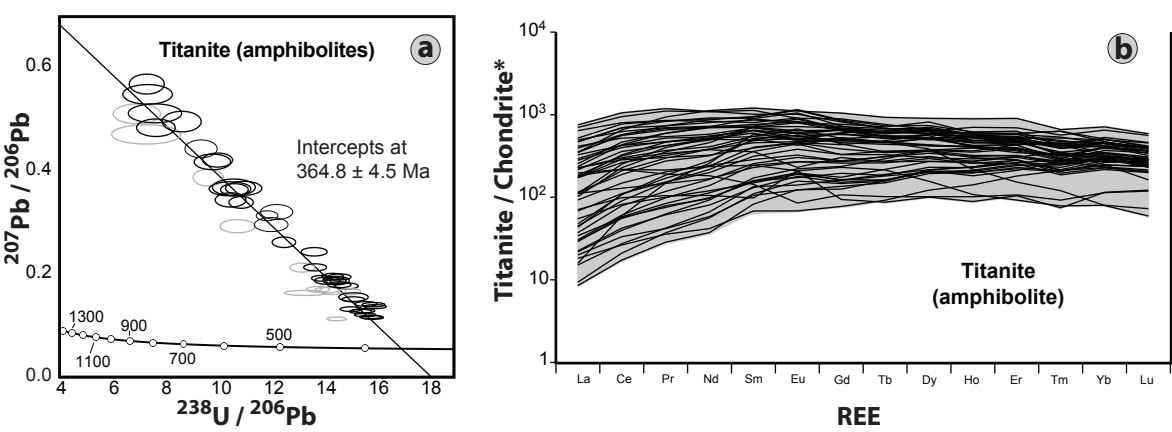

Figure 7. (a) Tera-Wasserburg diagram showing distribution of analyzed titanites (n = 51) from Sobrado amphibolite (JBP-71-21). The rejected analyses are represented by gray ellipses. The ellipses represent the $^{207}$Pb/$^{206}$Pb and $^{238}$U/$^{206}$Pb errors (±2σ). (b) Chondrite-normalized rare earth element (REE) patterns for the same titanites.

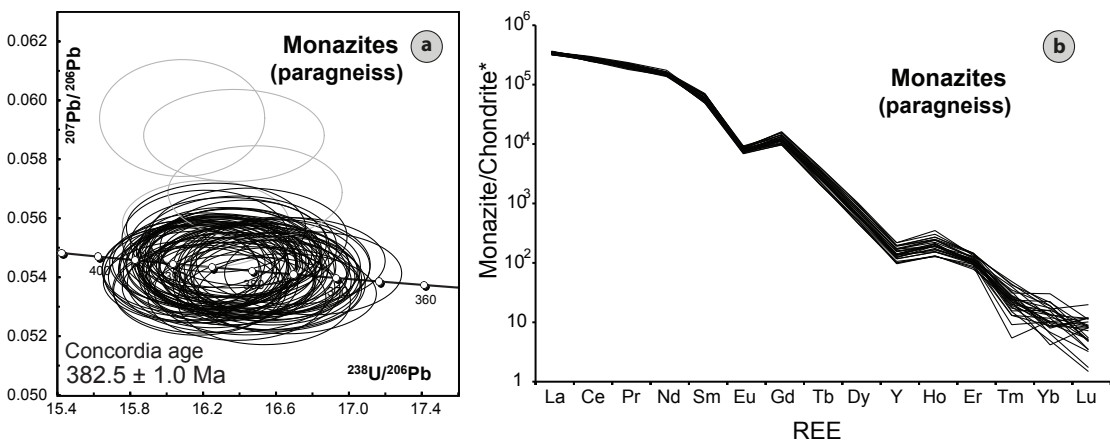

Figure 8. (a) Tera-Wasserburg diagram showing distribution of analyzed monazites (n = 76) from Sobrado paragneiss (*JBP-71-15*). The rejected analyses are represented by gray ellipses. The ellipses represent the $^{207}Pb/^{206}Pb$ and $^{238}U/^{206}Pb$ errors (±2σ). (b) Chondrite-normalized rare earth element (REE) patterns for the same monazites.

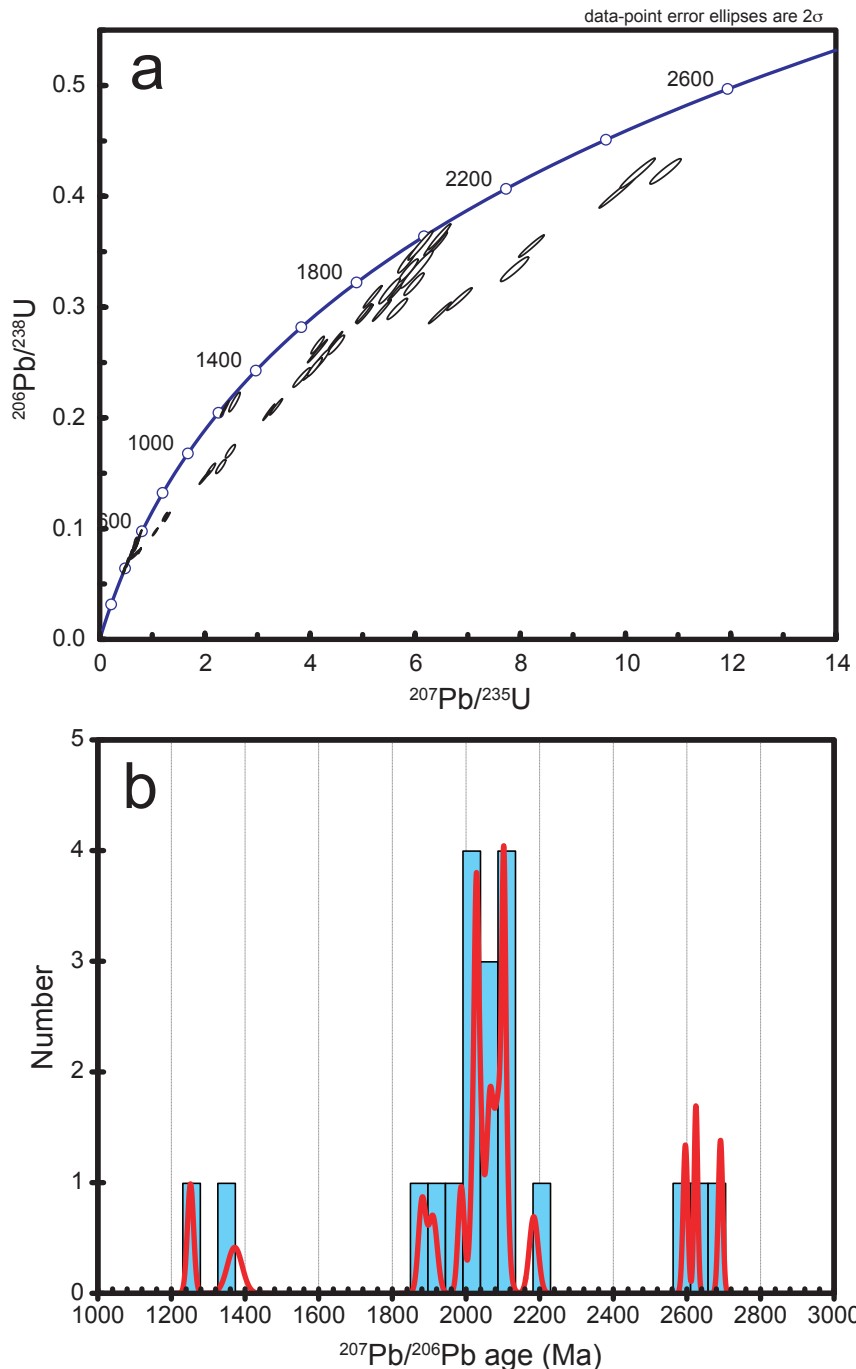

Figure 9: Concordia plot (a) including all zircon with concordance >90% from sample JBP-71-15 (Sobrado migmatitic paragneiss), and (b) age histogram and probability density plot for ages older than 1000 Ma.

Benítez-Pérez et al.

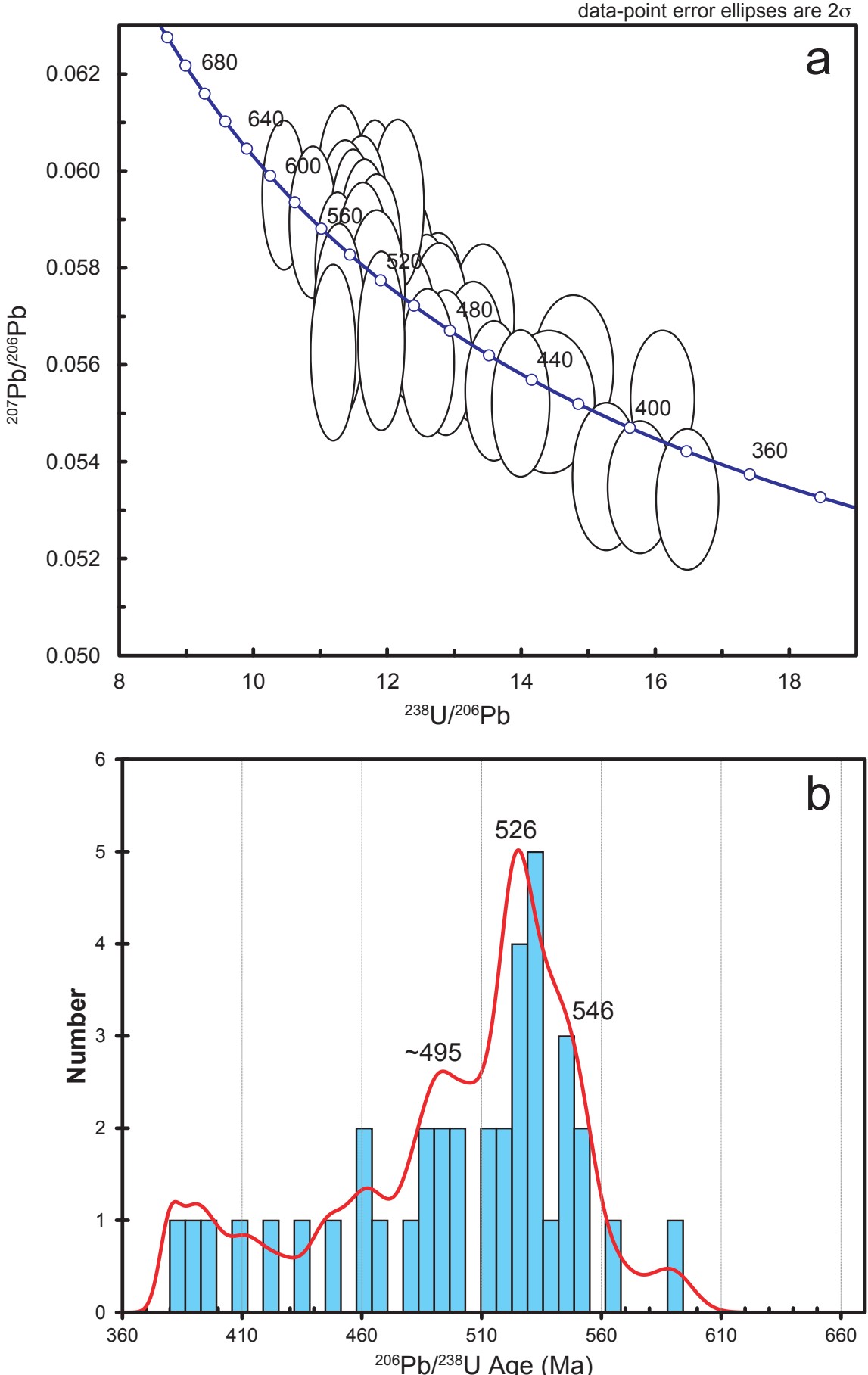

Figure 10: Tera-Wasserburg diagram (a) for the analyses between 589 and 380 Ma, and, (b) age histogram and probability density plot for the same ages.

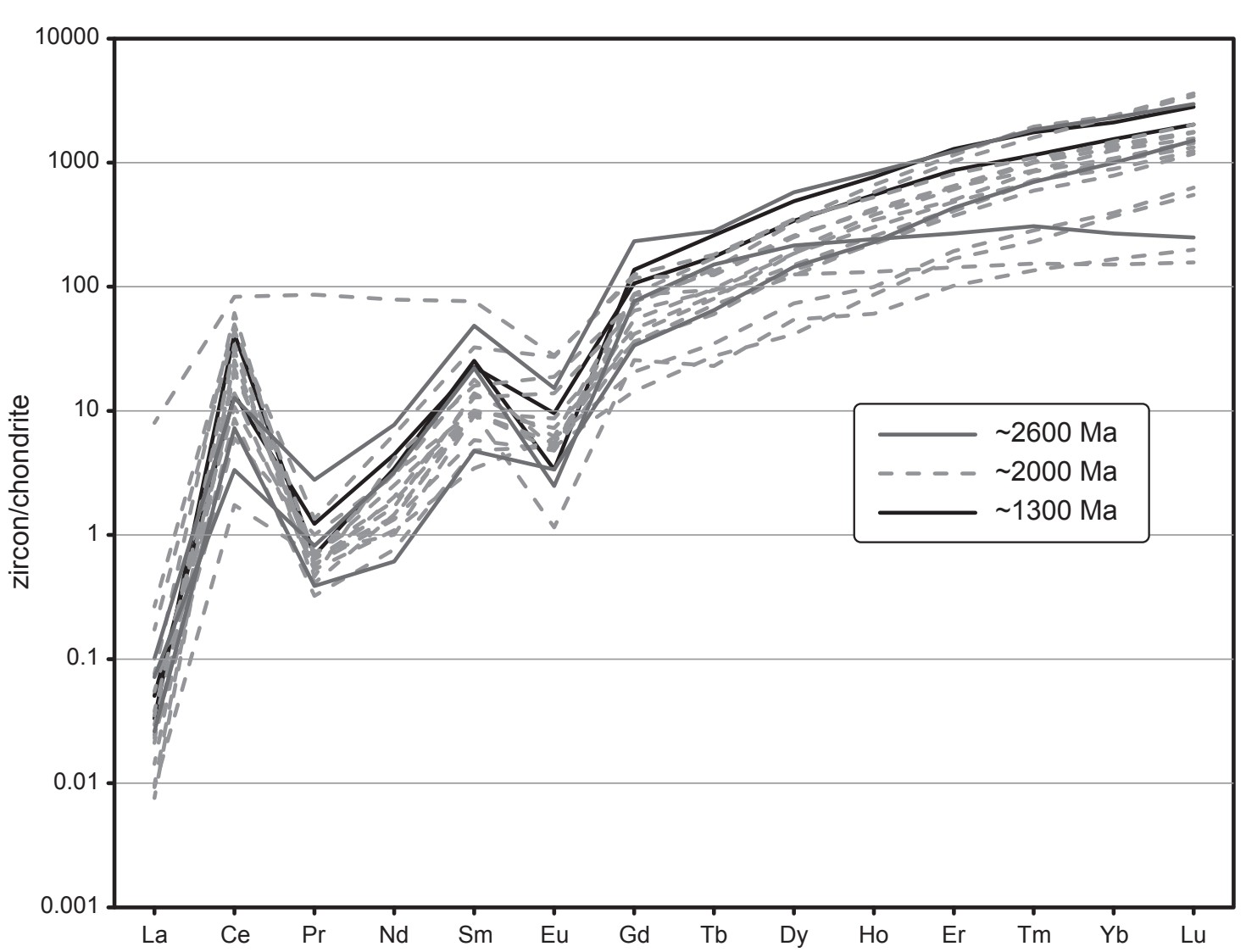

Figure 11: Chondrite-normalized plots for inherited zircon older than 1000 Ma.

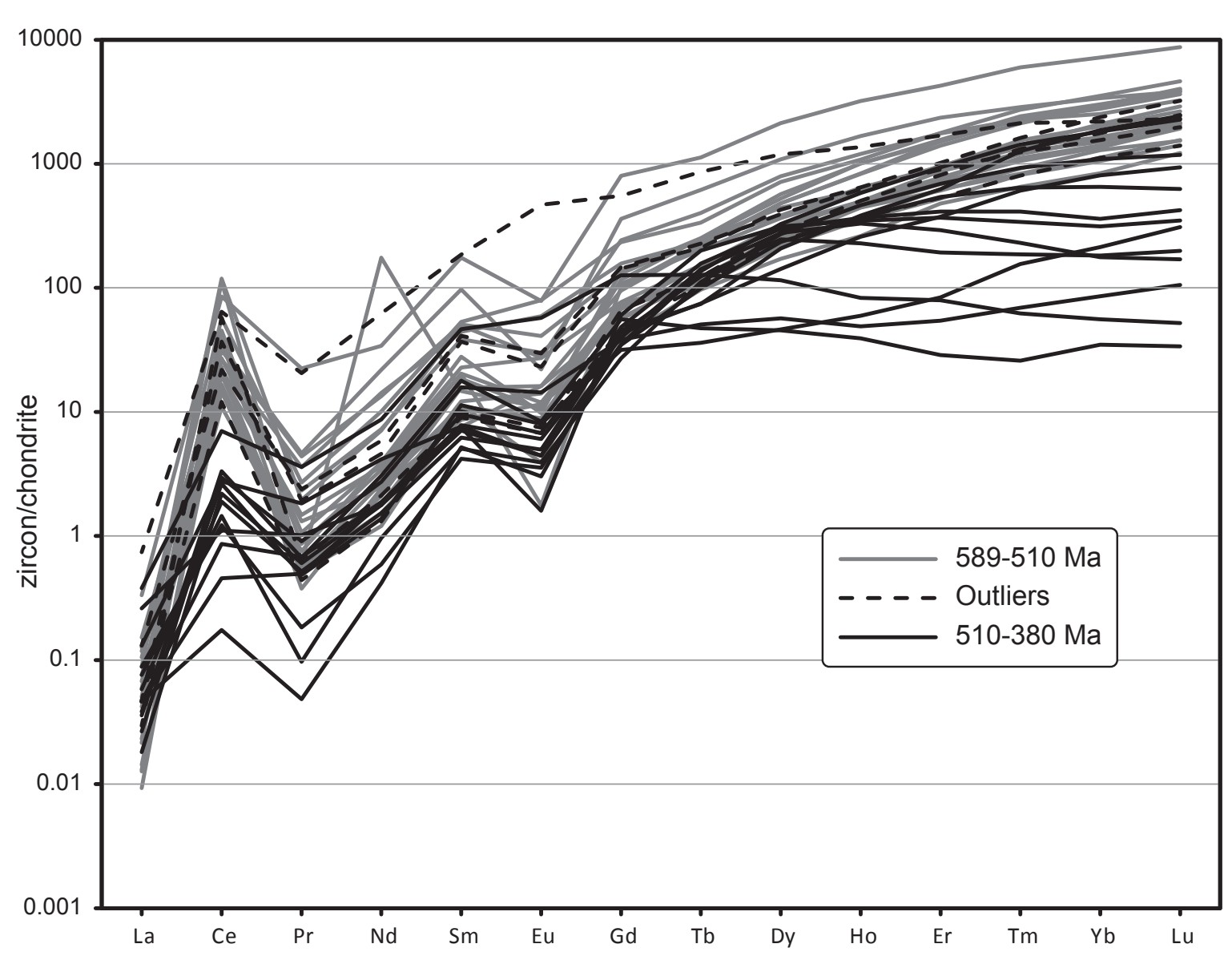

Figure 12: Chondrite-normalized plots for zircon between 589 and 380 Ma.

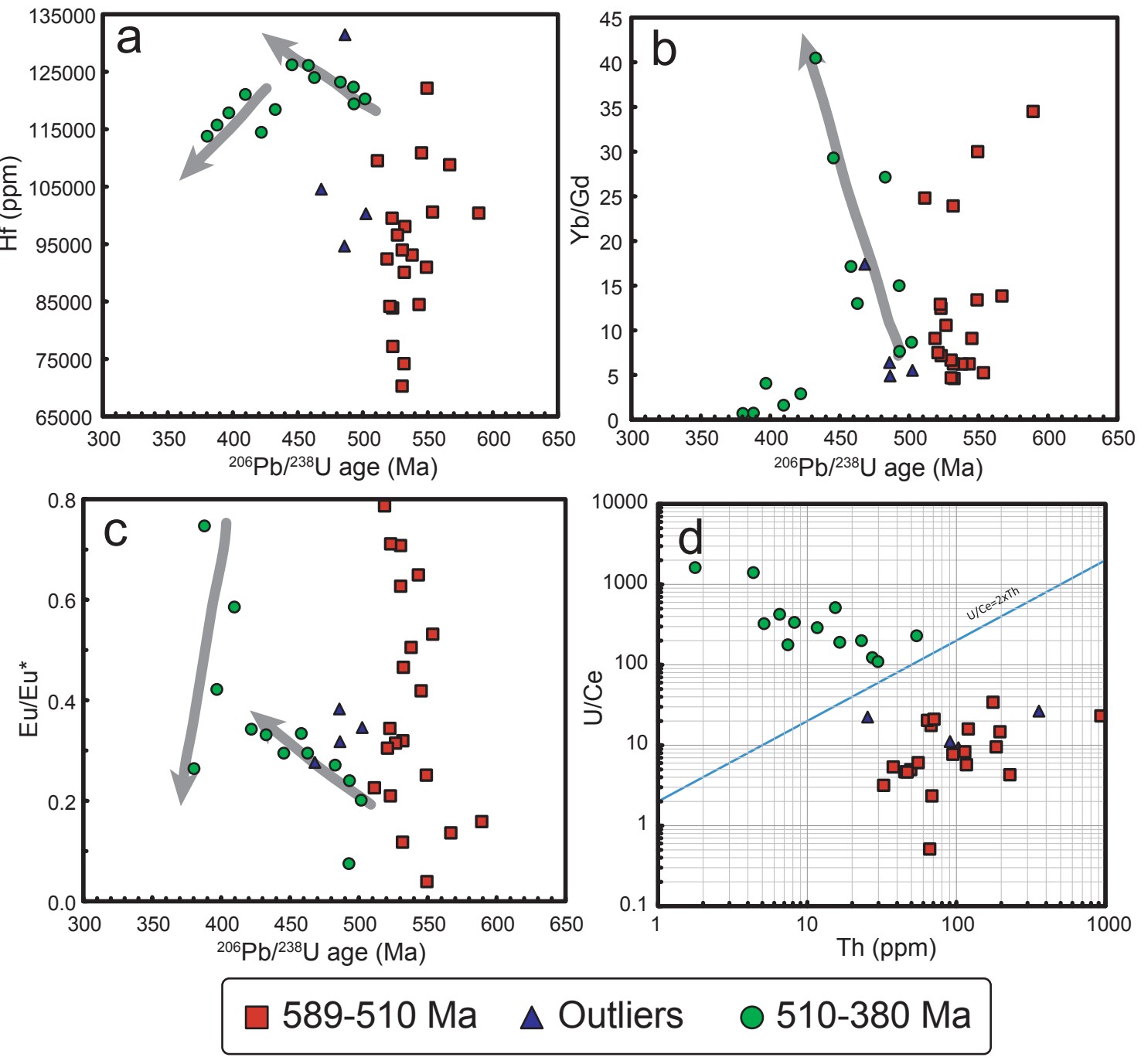

Figure 13: (a) Hafnium versus age, (b) Yb/Gd versus age, (c) Eu/Eu* versus age, and (d) U/Ce versus Th for zircon analyses between 589 and 380 Ma.

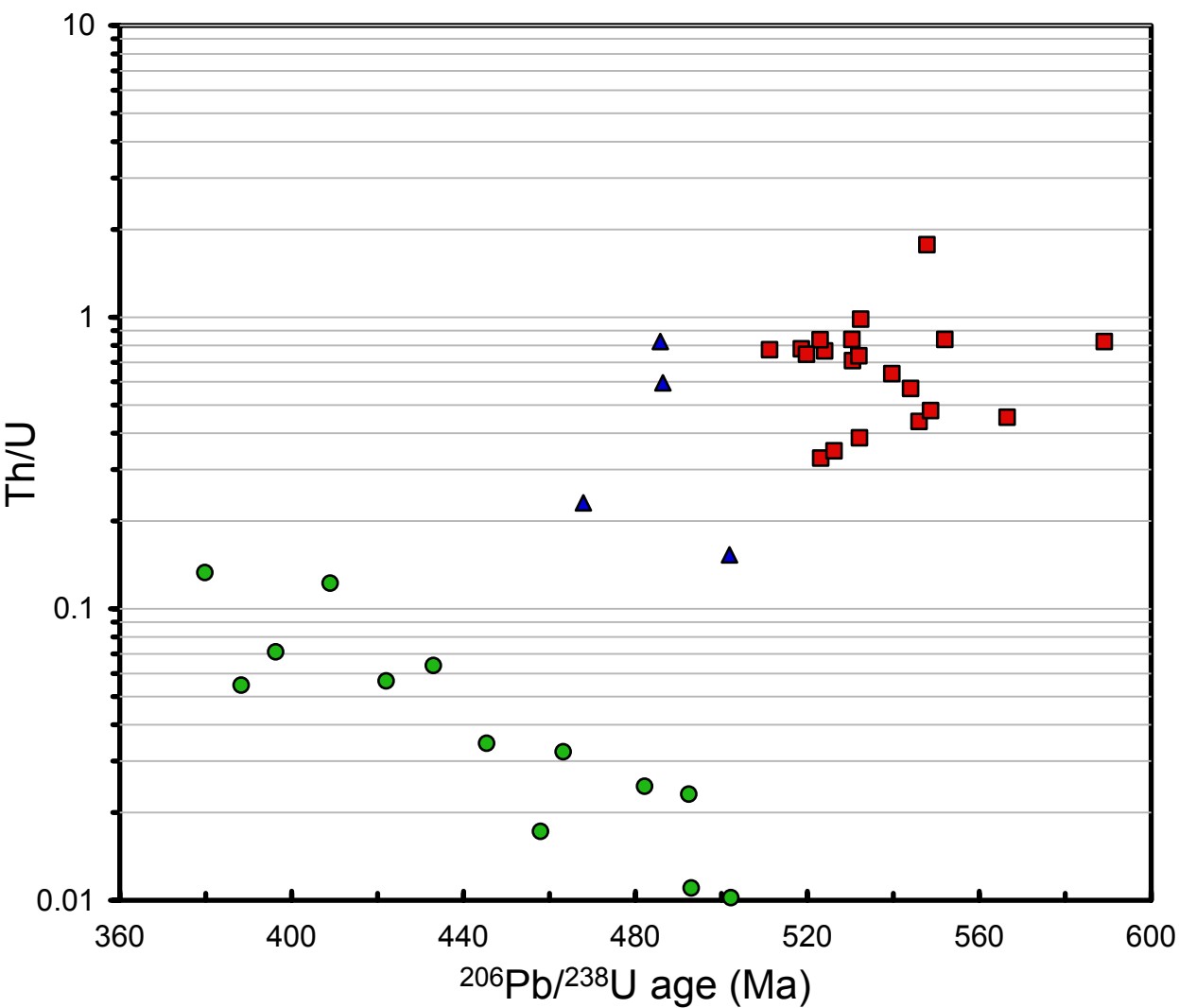

Figure 14:Th/U ratio versus $^{206}$Pb/$^{238}$U ages for the zircon analyses from 589 to 380 Ma. Analysis 63 (510 Ma) is not represented as it has an anomalous value (6.59).

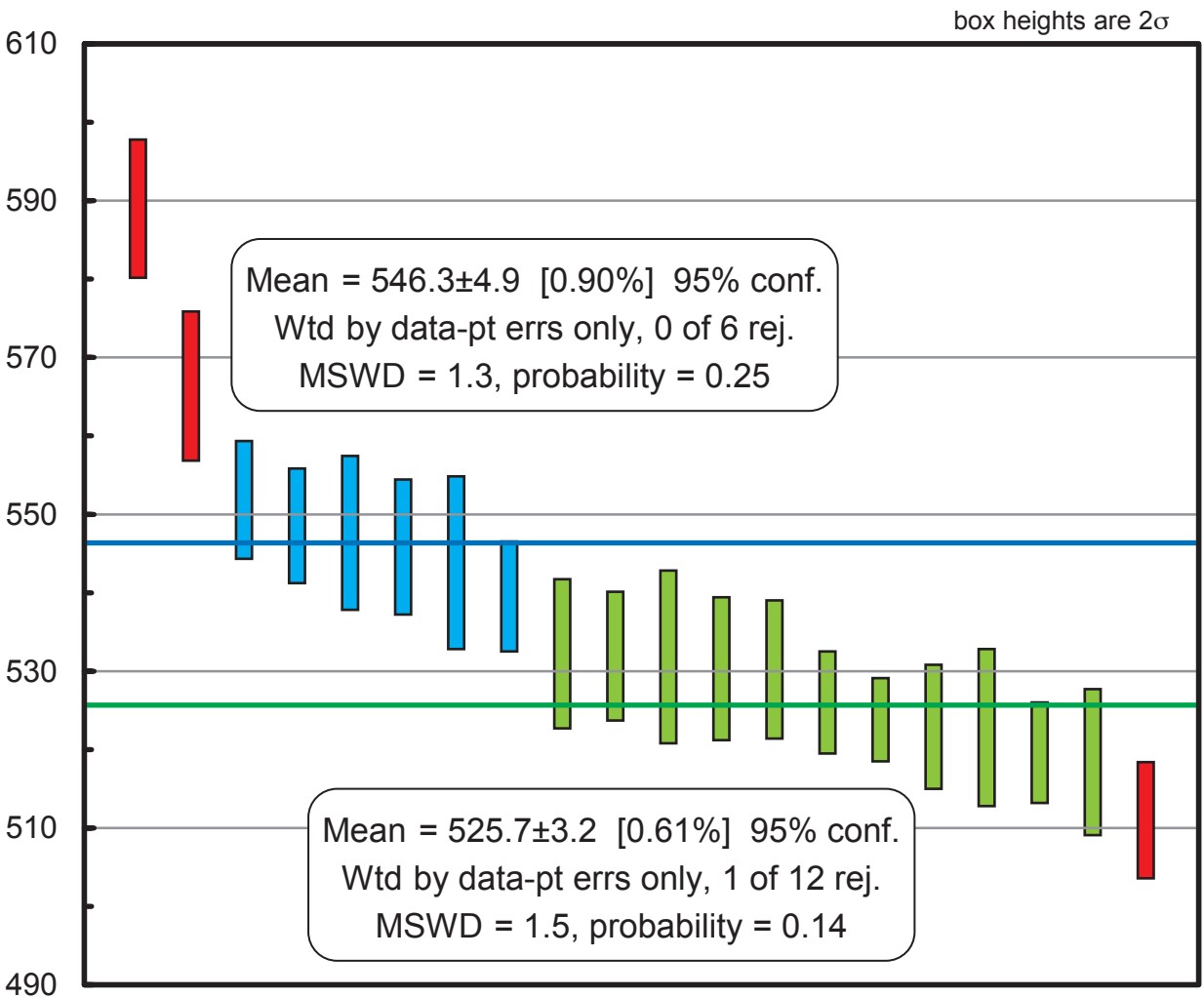

box heights are 2σ

Mean = 546.3±4.9 [0.90%] 95% conf.
Wtd by data-pt errs only, 0 of 6 rej.
MSWD = 1.3, probability = 0.25

Mean = 525.7±3.2 [0.61%] 95% conf.
Wtd by data-pt errs only, 1 of 12 rej.
MSWD = 1.5, probability = 0.14

Figure 15:Weigthed average obtained from magmatic ages distributed between 589 and 510 Ma.

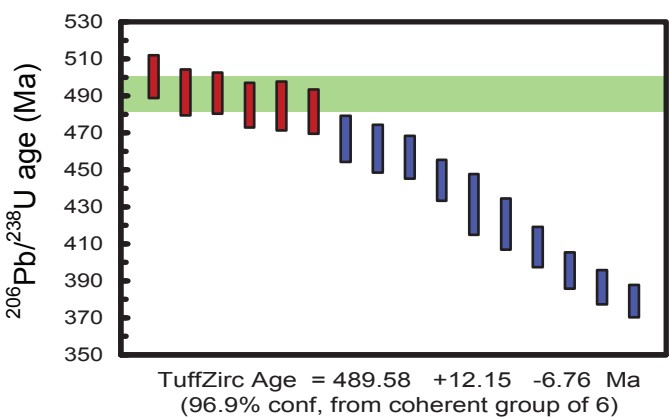

Figure 16: Age of the onset of the oldest HP-HT metamorphic
event obtained using the TuffZirc algorythm.

## Table 1_U-Th_Pb+REE Titatine_McD_s

| Spot | 238U/206Pb | ± | 207Pb/206Pb | ± | 208Pb/232Th | ± | U (ppm) | Th (ppm) | La (ppm) | Ce (ppm) | Pr (ppm) | Nd (ppm) | Sm (ppm) | Eu (ppm) | Gd (ppm) | Tb (ppm) | Dy (ppm) | Ho (ppm) | Er (ppm) | Tm (ppm) | Yb (ppm) | Lu (ppm) |
|---|---|---|---|---|---|---|---|---|---|---|---|---|---|---|---|---|---|---|---|---|---|---|
| 1 | 15.59 | 0.35 | 0.1279 | 0.0032 | 0.02292 | 0.00053 | 7.42 | 16.57 | 751 | 1052 | 1182 | 1096 | 1012 | 1012 | 818 | 681 | 635 | 590 | 521 | 423.1 | 405 | 363.4 |
| 2 | 13.37 | 0.66 | 0.1625 | 0.0040 | 0.02580 | 0.00074 | 8.22 | 16.73 | 701 | 946 | 1107 | 1085 | 1128 | 904 | 889 | 762 | 661 | 584 | 483 | 333.6 | 324 | 232.9 |
| 3 | 13.74 | 0.36 | 0.2123 | 0.0056 | 0.03513 | 0.00101 | 4.16 | 7.44 | 327 | 424 | 520 | 538 | 650 | 588 | 574 | 559 | 522 | 538 | 532 | 411.7 | 465 | 392.7 |
| 4 | 13.85 | 0.36 | 0.1694 | 0.0045 | 0.05450 | 0.00298 | 4.30 | 2.47 | 116 | 188 | 275 | 318 | 484 | 519 | 503 | 471 | 419 | 408 | 365 | 278.9 | 299 | 230.5 |
| 5 | 13.70 | 0.41 | 0.2426 | 0.0067 | 0.05450 | 0.00376 | 2.71 | 2.35 | 109 | 209 | 297 | 383 | 511 | 561 | 458 | 437 | 422 | 416 | 386 | 287.9 | 315 | 261.4 |
| 6 | 14.20 | 0.40 | 0.1864 | 0.0058 | 0.03180 | 0.00119 | 3.16 | 4.74 | 225 | 300 | 369 | 409 | 547 | 671 | 515 | 539 | 495 | 526 | 511 | 393.5 | 448 | 383.7 |
| 7 | 12.06 | 0.52 | 0.2955 | 0.0111 | 0.15000 | 0.18002 | 2.00 | 1.46 | 68 | 111 | 153 | 199 | 314 | 382 | 329 | 361 | 373 | 394 | 399 | 318.6 | 340 | 290.7 |
| 8 | 14.52 | 0.33 | 0.1863 | 0.0050 | 0.03051 | 0.00091 | 4.86 | 7.93 | 394 | 574 | 681 | 705 | 762 | 813 | 685 | 648 | 589 | 608 | 579 | 436.8 | 504 | 417.1 |
| 9 | 15.11 | 0.35 | 0.1310 | 0.0037 | 0.02719 | 0.00074 | 6.55 | 10.39 | 471 | 692 | 733 | 803 | 861 | 828 | 714 | 679 | 636 | 703 | 693 | 565.6 | 650 | 561.0 |
| 10 | 12.55 | 0.37 | 0.2611 | 0.0083 | 0.14400 | 0.04010 | 2.35 | 0.96 | 45 | 90 | 152 | 217 | 359 | 442 | 328 | 329 | 300 | 291 | 304 | 252.6 | 290 | 245.5 |
| 11 | 15.48 | 0.35 | 0.1264 | 0.0034 | 0.03340 | 0.00137 | 8.13 | 6.22 | 350 | 654 | 938 | 1120 | 1201 | 1107 | 1045 | 928 | 902 | 890 | 901 | 659.9 | 711 | 586.6 |
| 12 | 7.30 | 1.08 | 0.4710 | 0.0153 | 0.07200 | 0.09601 | 0.61 | 0.32 | 15 | 27 | 42 | 67 | 134 | 140 | 162 | 189 | 215 | 233 | 264 | 248.2 | 287 | 232.1 |
| 13 | 10.02 | 0.47 | 0.4211 | 0.0111 | 0.11200 | 0.09503 | 1.20 | 0.69 | 31 | 61 | 98 | 139 | 244 | 209 | 230 | 214 | 156 | 108 | 93 | 78.5 | 80 | 72.8 |
| 14 | 9.66 | 0.47 | 0.3870 | 0.0135 | 0.22000 | 0.17006 | 1.09 | 0.56 | 29 | 55 | 79 | 96 | 159 | 197 | 160 | 181 | 200 | 213 | 249 | 230.4 | 296 | 242.7 |
| 15 | 13.99 | 0.32 | 0.1910 | 0.0046 | 0.03377 | 0.00107 | 4.58 | 6.10 | 184 | 317 | 421 | 512 | 599 | 485 | 518 | 467 | 433 | 401 | 374 | 319.8 | 306 | 236.2 |
| 16 | 14.41 | 0.41 | 0.1672 | 0.0063 | 0.03150 | 0.00144 | 4.02 | 5.44 | 216 | 359 | 421 | 492 | 642 | 501 | 609 | 609 | 590 | 549 | 545 | 471.3 | 464 | 395.9 |
| 17 | 14.94 | 0.36 | 0.1761 | 0.0048 | 0.13000 | 0.03609 | 4.38 | 0.71 | 43 | 108 | 170 | 239 | 404 | 460 | 433 | 485 | 572 | 540 | 536 | 488.3 | 420 | 339.0 |
| 18 | 7.35 | 0.77 | 0.5490 | 0.0155 | -0.08000 | -0.14001 | 0.55 | 0.30 | 18 | 26 | 36 | 42 | 68 | 68 | 77 | 92 | 122 | 141 | 181 | 210.9 | 230 | 204.5 |
| 19 | 15.96 | 0.36 | 0.1397 | 0.0034 | 0.02710 | 0.00071 | 8.10 | 9.22 | 414 | 520 | 541 | 580 | 692 | 609 | 652 | 624 | 708 | 579 | 591 | 630.8 | 507 | 405.7 |
| 20 | 15.04 | 0.36 | 0.1649 | 0.0049 | 0.03031 | 0.00096 | 6.39 | 8.79 | 293 | 426 | 468 | 522 | 543 | 499 | 482 | 455 | 514 | 419 | 438 | 446.6 | 412 | 321.1 |
| 21 | 11.90 | 0.34 | 0.3127 | 0.0079 | 0.37000 | 0.22012 | 2.91 | 0.40 | 48 | 108 | 185 | 276 | 507 | 426 | 527 | 465 | 385 | 218 | 153 | 120.2 | 80 | 59.3 |
| 22 | 14.41 | 0.34 | 0.1940 | 0.0054 | 0.05140 | 0.00216 | 5.29 | 3.73 | 255 | 465 | 598 | 716 | 889 | 794 | 766 | 709 | 817 | 603 | 595 | 629.1 | 472 | 382.1 |
| 23 | 14.18 | 0.37 | 0.1682 | 0.0046 | 0.09800 | 0.01612 | 4.77 | 1.43 | 102 | 228 | 355 | 458 | 710 | 718 | 687 | 604 | 671 | 488 | 476 | 449.4 | 344 | 258.5 |
| 24 | 15.92 | 0.35 | 0.1147 | 0.0030 | 0.02388 | 0.00057 | 10.78 | 13.20 | 468 | 695 | 829 | 915 | 901 | 1052 | 731 | 604 | 620 | 458 | 442 | 428.3 | 323 | 278.0 |
| 25 | 7.63 | 0.83 | 0.5120 | 0.0150 | -0.03000 | -0.49000 | 0.68 | 0.14 | 9 | 21 | 37 | 56 | 110 | 157 | 147 | 157 | 209 | 182 | 211 | 270.4 | 234 | 214.2 |
| 26 | 12.27 | 0.50 | 0.3200 | 0.0119 | 0.05770 | 0.00542 | 1.68 | 2.17 | 93 | 148 | 203 | 243 | 286 | 325 | 275 | 274 | 313 | 271 | 307 | 349.0 | 312 | 285.4 |
| 27 | 14.67 | 0.35 | 0.1937 | 0.0048 | 0.03279 | 0.00098 | 4.67 | 6.77 | 354 | 560 | 653 | 700 | 703 | 639 | 652 | 551 | 557 | 401 | 347 | 332.8 | 248 | 162.6 |
| 28 | 15.20 | 0.46 | 0.1543 | 0.0069 | 0.03350 | 0.00183 | 4.88 | 4.43 | 174 | 326 | 463 | 554 | 642 | 542 | 593 | 512 | 581 | 487 | 521 | 521.1 | 496 | 453.7 |
| 29 | 14.56 | 0.31 | 0.1120 | 0.0029 | 0.02158 | 0.00049 | 11.71 | 22.45 | 495 | 625 | 696 | 722 | 728 | 886 | 653 | 584 | 642 | 562 | 575 | 536.0 | 444 | 363.0 |
| 30 | 10.92 | 0.38 | 0.3387 | 0.0094 | 0.05840 | 0.00378 | 1.54 | 2.51 | 111 | 129 | 147 | 168 | 229 | 267 | 270 | 312 | 375 | 364 | 431 | 455.9 | 451 | 411.0 |
| 31 | 10.64 | 0.67 | 0.3680 | 0.0132 | 0.12200 | 0.03409 | 1.07 | 0.81 | 29 | 58 | 84 | 114 | 158 | 173 | 195 | 184 | 230 | 228 | 262 | 283.0 | 270 | 259.3 |
| 32 | 9.39 | 0.50 | 0.4430 | 0.0141 | -0.02000 | -0.12000 | 0.90 | 0.58 | 22 | 42 | 61 | 81 | 130 | 179 | 152 | 162 | 207 | 205 | 242 | 246.2 | 289 | 248.8 |
| 33 | 10.49 | 0.65 | 0.3651 | 0.0111 | 0.09500 | 0.00969 | 1.13 | 1.22 | 55 | 82 | 100 | 120 | 149 | 189 | 163 | 162 | 203 | 196 | 229 | 210.5 | 247 | 236.6 |
| 34 | 15.19 | 0.35 | 0.1501 | 0.0040 | 0.04200 | 0.00163 | 5.56 | 3.48 | 235 | 434 | 593 | 696 | 705 | 801 | 639 | 537 | 517 | 454 | 488 | 390.3 | 419 | 399.2 |
| 35 | 14.53 | 0.41 | 0.1866 | 0.0067 | 0.13800 | 0.00129 | 3.69 | 4.89 | 172 | 263 | 322 | 383 | 457 | 471 | 483 | 465 | 516 | 482 | 494 | 401.6 | 464 | 401.2 |
| 36 | 14.66 | 0.36 | 0.1785 | 0.0047 | 0.02576 | 0.00378 | 4.36 | 2.21 | 111 | 252 | 399 | 514 | 703 | 588 | 649 | 544 | 524 | 463 | 461 | 351.8 | 391 | 368.3 |
| 37 | 16.03 | 0.34 | 0.1359 | 0.0032 | 0.04720 | 0.00230 | 8.41 | 3.38 | 281 | 400 | 473 | 503 | 493 | 611 | 443 | 355 | 366 | 321 | 320 | 264.8 | 297 | 287.8 |
| 38 | 10.52 | 0.48 | 0.3435 | 0.0113 | 0.14400 | 0.06906 | 1.28 | 0.55 | 22 | 40 | 63 | 93 | 172 | 221 | 236 | 293 | 370 | 352 | 385 | 304.0 | 348 | 308.1 |
| 39 | 8.65 | 0.61 | 0.4960 | 0.0164 | 0.00000 | 0.00000 | 0.66 | 0.24 | 9 | 17 | 29 | 37 | 83 | 123 | 126 | 158 | 215 | 232 | 284 | 215.8 | 278 | 267.9 |
| 40 | 15.85 | 0.34 | 0.1160 | 0.0028 | 0.02681 | 0.00080 | 7.59 | 7.20 | 348 | 566 | 735 | 869 | 950 | 1151 | 862 | 731 | 739 | 619 | 632 | 420.2 | 530 | 460.6 |
| 41 | 11.89 | 0.31 | 0.1654 | 0.0046 | 0.06000 | 0.14001 | 2.54 | 0.09 | 16 | 60 | 148 | 259 | 580 | 512 | 585 | 518 | 511 | 425 | 428 | 291.9 | 411 | 362.2 |
| 42 | 9.82 | 0.57 | 0.4175 | 0.0124 | 0.13800 | 0.08205 | 0.95 | 0.79 | 71 | 99 | 105 | 105 | 98 | 210 | 94 | 87 | 100 | 89 | 104 | 74.5 | 115 | 122.0 |
| 43 | 15.67 | 0.34 | 0.1411 | 0.0037 | 0.02576 | 0.00068 | 6.21 | 10.38 | 640 | 799 | 852 | 832 | 803 | 913 | 671 | 607 | 625 | 520 | 493 | 319.8 | 397 | 362.2 |
| 44 | 11.01 | 0.57 | 0.3661 | 0.0111 | 0.11000 | 0.14002 | 1.20 | 0.86 | 34 | 57 | 75 | 99 | 154 | 196 | 174 | 218 | 270 | 274 | 316 | 272.5 | 369 | 345.9 |
| 45 | 7.67 | 0.60 | 0.4840 | 0.0139 | 0.04000 | 0.25000 | 0.57 | 0.33 | 20 | 33 | 42 | 58 | 105 | 123 | 136 | 150 | 193 | 200 | 221 | 204.0 | 263 | 204.9 |
| 46 | 13.25 | 0.39 | 0.2117 | 0.0071 | 0.03790 | 0.00242 | 2.61 | 4.41 | 228 | 284 | 304 | 324 | 395 | 455 | 384 | 348 | 358 | 302 | 284 | 195.5 | 227 | 205.9 |
| 47 | 10.78 | 0.54 | 0.2924 | 0.0105 | 0.13300 | 0.07405 | 1.66 | 0.79 | 37 | 79 | 118 | 165 | 255 | 304 | 277 | 309 | 340 | 313 | 321 | 253.8 | 305 | 288.2 |
| 48 | 6.94 | 0.74 | 0.5110 | 0.0158 | 0.03214 | 0.00100 | 0.64 | 1.89 | 173 | 290 | 339 | 376 | 356 | 230 | 303 | 269 | 273 | 260 | 255 | 186.6 | 217 | 199.2 |
| 49 | 15.64 | 0.33 | 0.1197 | 0.0028 | 0.06970 | 0.00481 | 10.32 | 7.50 | 521 | 772 | 845 | 886 | 937 | 1130 | 839 | 676 | 683 | 551 | 478 | 336.0 | 339 | 299.2 |
| 50 | 10.66 | 0.43 | 0.3635 | 0.0094 | 0.06970 | 0.00481 | 1.77 | 2.45 | 197 | 218 | 213 | 235 | 245 | 269 | 259 | 261 | 294 | 258 | 279 | 210.9 | 246 | 236.6 |
| 51 | 7.32 | 0.55 | 0.5690 | 0.0158 | 0.08940 | 0.00820 | 0.70 | 1.76 | 183 | 217 | 191 | 182 | 143 | 85 | 107 | 102 | 102 | 101 | 107 | 92.3 | 114 | 118.7 |

# Table 2A_U-Th_Pb+REE Monazite _McD_S

| Spot | 238U/206Pb | ± | 207Pb/206Pb | ± | 208Pb/232Th | ± | U (ppm) | Th (ppm) | La (ppm) | Ce (ppm) | Pr (ppm) | Nd (ppm) | Sm (ppm) | Eu (ppm) | Gd (ppm) | Tb (ppm) | Dy (ppm) | Y (ppm) | Ho (ppm) | Er (ppm) | Tm (ppm) | Yb (ppm) | Lu (ppm) |
|---|---|---|---|---|---|---|---|---|---|---|---|---|---|---|---|---|---|---|---|---|---|---|---|
| 1 | 16.29 | 0.35 | 0.0543 | 0.0013 | 0.01930 | 0.00041 | 1640 | 26200 | 554430 | 433931 | 365302 | 243982 | 98243 | 12131 | 19899 | 4429 | 992 | 243 | 273 | 170 | 32.4 | 20 | 3.3 |
| 2 | 16.28 | 0.35 | 0.0545 | 0.0013 | 0.01916 | 0.00042 | 1850 | 29300 | 527004 | 437194 | 352371 | 242451 | 94324 | 13286 | 19749 | 4490 | 1081 | 272 | 322 | 191 | 27.5 | 25 | 0.0 |
| 3 | 16.20 | 0.38 | 0.0544 | 0.0012 | 0.01949 | 0.00044 | 2300 | 25500 | 519409 | 420881 | 346983 | 250109 | 107905 | 12451 | 23869 | 5319 | 1163 | 282 | 374 | 190 | 71.7 | 8 | 24.8 |
| 4 | 16.30 | 0.36 | 0.0546 | 0.0012 | 0.01930 | 0.00047 | 1454 | 34200 | 563713 | 425775 | 323276 | 223632 | 85270 | 11474 | 18543 | 4266 | 898 | 192 | 251 | 118 | 46.6 | 6 | 12.2 |
| 5 | 16.18 | 0.36 | 0.0543 | 0.0012 | 0.01926 | 0.00043 | 1805 | 32400 | 547257 | 420881 | 353448 | 231947 | 92568 | 11385 | 19899 | 4197 | 959 | 179 | 260 | 143 | 38.1 | 14 | 0.0 |
| 6 | 16.09 | 0.35 | 0.0541 | 0.0013 | 0.01946 | 0.00044 | 1660 | 25900 | 524051 | 409462 | 338362 | 238074 | 96622 | 11794 | 19045 | 4634 | 980 | 227 | 291 | 184 | 44.5 | 19 | 5.7 |
| 7 | 16.27 | 0.36 | 0.0544 | 0.0012 | 0.01938 | 0.00044 | 1790 | 29000 | 554430 | 442088 | 369612 | 230853 | 95676 | 12647 | 19598 | 4072 | 1045 | 248 | 310 | 172 | 46.6 | 20 | 30.5 |
| 8 | 16.15 | 0.36 | 0.0545 | 0.0012 | 0.01920 | 0.00042 | 2170 | 44100 | 579325 | 437194 | 359914 | 248796 | 97838 | 13179 | 18693 | 3875 | 927 | 203 | 306 | 182 | 42.1 | 12 | 8.1 |
| 9 | 16.22 | 0.37 | 0.0558 | 0.0012 | 0.01900 | 0.00045 | 2150 | 47100 | 567511 | 422512 | 340517 | 245295 | 90608 | 12451 | 17789 | 3958 | 902 | 232 | 286 | 179 | 52.2 | 6 | 25.6 |
| 10 | 16.38 | 0.36 | 0.0549 | 0.0012 | 0.01940 | 0.00045 | 1680 | 27900 | 526160 | 435563 | 357759 | 242451 | 93919 | 12664 | 19799 | 4078 | 947 | 210 | 264 | 143 | 46.2 | 11 | 8.9 |
| 11 | 16.20 | 0.37 | 0.0547 | 0.0012 | 0.01930 | 0.00044 | 1950 | 35700 | 555274 | 433931 | 383621 | 254923 | 95270 | 12060 | 19296 | 4432 | 996 | 221 | 321 | 181 | 44.1 | 17 | 5.7 |
| 12 | 16.29 | 0.37 | 0.0546 | 0.0012 | 0.01915 | 0.00044 | 2280 | 32800 | 555274 | 415987 | 336207 | 252298 | 103176 | 13215 | 20201 | 4510 | 976 | 236 | 280 | 195 | 39.7 | 7 | 0.0 |
| 13 | 16.35 | 0.38 | 0.0548 | 0.0012 | 0.01897 | 0.00044 | 2230 | 43200 | 540928 | 411093 | 349138 | 246827 | 94662 | 12202 | 18643 | 4551 | 976 | 218 | 282 | 138 | 36.0 | 13 | 11.8 |
| 14 | 16.28 | 0.36 | 0.0545 | 0.0012 | 0.01908 | 0.00043 | 2060 | 41000 | 564135 | 438825 | 362069 | 230197 | 95608 | 12575 | 19296 | 4269 | 866 | 209 | 251 | 162 | 16.2 | 13 | 8.1 |
| 15 | 16.31 | 0.39 | 0.0545 | 0.0012 | 0.01920 | 0.00044 | 2020 | 29600 | 535865 | 450245 | 378233 | 242888 | 109932 | 13268 | 22412 | 5125 | 1085 | 264 | 335 | 179 | 56.7 | 16 | 12.2 |
| 16 | 16.23 | 0.36 | 0.0544 | 0.0012 | 0.01953 | 0.00044 | 1714 | 31400 | 562447 | 460033 | 391164 | 261050 | 103514 | 13748 | 19196 | 4432 | 988 | 211 | 244 | 182 | 37.2 | 14 | 8.9 |
| 17 | 16.40 | 0.36 | 0.0546 | 0.0012 | 0.01910 | 0.00045 | 2360 | 37500 | 565401 | 429038 | 366379 | 241357 | 96689 | 12398 | 18894 | 4352 | 1004 | 218 | 291 | 177 | 25.5 | 10 | 3.3 |
| 18 | 16.22 | 0.36 | 0.0545 | 0.0012 | 0.01943 | 0.00043 | 1662 | 30800 | 580591 | 464927 | 376078 | 262363 | 100000 | 12256 | 18894 | 4654 | 1081 | 228 | 308 | 168 | 48.2 | 17 | 2.8 |
| 19 | 16.25 | 0.38 | 0.0538 | 0.0013 | 0.01932 | 0.00046 | 1600 | 31200 | 618143 | 442088 | 383621 | 244420 | 101419 | 12593 | 20050 | 4399 | 1024 | 217 | 253 | 161 | 34.0 | 4 | 6.9 |
| 20 | 16.11 | 0.35 | 0.0545 | 0.0012 | 0.01915 | 0.00042 | 2190 | 34200 | 567932 | 427406 | 364224 | 240919 | 101351 | 12611 | 19899 | 4598 | 1012 | 238 | 300 | 201 | 30.8 | 18 | 3.3 |
| 21 | 16.17 | 0.35 | 0.0543 | 0.0012 | 0.01926 | 0.00045 | 1804 | 28900 | 537975 | 443719 | 365302 | 243326 | 100608 | 13428 | 18693 | 4197 | 951 | 208 | 286 | 153 | 39.7 | 17 | 3.3 |
| 22 | 16.08 | 0.35 | 0.0543 | 0.0012 | 0.01964 | 0.00044 | 2380 | 38200 | 560338 | 448613 | 383621 | 249672 | 101554 | 12913 | 19497 | 5125 | 1065 | 222 | 308 | 167 | 21.1 | 16 | 9.3 |
| 23 | 16.08 | 0.37 | 0.0594 | 0.0016 | 0.01886 | 0.00044 | 1704 | 84000 | 562447 | 391517 | 327586 | 231947 | 87635 | 12487 | 16281 | 4019 | 951 | 246 | 264 | 215 | 68.0 | 30 | 40.7 |
| 24 | 16.18 | 0.39 | 0.0536 | 0.0012 | 0.01938 | 0.00044 | 1650 | 29500 | 541772 | 417618 | 359914 | 238731 | 97230 | 12647 | 18693 | 4529 | 1110 | 246 | 315 | 188 | 31.2 | 19 | 11.0 |
| 25 | 16.37 | 0.39 | 0.0546 | 0.0012 | 0.01876 | 0.00045 | 1900 | 57800 | 564557 | 417618 | 337284 | 232604 | 94797 | 11634 | 16935 | 4030 | 894 | 217 | 295 | 148 | 79.4 | 12 | 3.7 |
| 26 | 16.43 | 0.38 | 0.0550 | 0.0013 | 0.01922 | 0.00044 | 1663 | 27800 | 566667 | 425512 | 360991 | 243545 | 97635 | 12700 | 17236 | 4488 | 963 | 249 | 288 | 184 | 56.3 | 17 | 14.6 |
| 27 | 16.32 | 0.38 | 0.0549 | 0.0012 | 0.01934 | 0.00046 | 2000 | 32900 | 575949 | 432300 | 354526 | 249453 | 100608 | 13250 | 19347 | 4958 | 1167 | 255 | 306 | 177 | 47.8 | 25 | 7.3 |
| 28 | 16.35 | 0.38 | 0.0538 | 0.0011 | 0.01948 | 0.00046 | 3680 | 28300 | 582700 | 445351 | 363147 | 264770 | 118243 | 15062 | 25025 | 6648 | 1553 | 383 | 443 | 213 | 45.7 | 16 | 16.7 |
| 29 | 16.58 | 0.35 | 0.0539 | 0.0011 | 0.01908 | 0.00044 | 3050 | 32100 | 551899 | 432300 | 385776 | 240044 | 104054 | 14512 | 24171 | 6427 | 1451 | 352 | 399 | 221 | 43.3 | 24 | 7.7 |
| 30 | 16.27 | 0.38 | 0.0546 | 0.0012 | 0.01941 | 0.00043 | 1822 | 27600 | 562447 | 440457 | 375000 | 260394 | 104054 | 13694 | 21156 | 5235 | 1102 | 240 | 332 | 191 | 30.4 | 16 | 16.7 |
| 31 | 16.63 | 0.40 | 0.0544 | 0.0012 | 0.01874 | 0.00045 | 1926 | 28200 | 570042 | 437194 | 350216 | 232604 | 99324 | 11261 | 18693 | 4432 | 992 | 211 | 288 | 196 | 51.8 | 18 | 3.3 |
| 32 | 16.68 | 0.42 | 0.0546 | 0.0013 | 0.01915 | 0.00047 | 2130 | 35500 | 542194 | 419250 | 371767 | 248359 | 102703 | 13144 | 18191 | 4834 | 996 | 243 | 291 | 168 | 47.8 | 21 | 0.0 |
| 33 | 16.36 | 0.39 | 0.0541 | 0.0012 | 0.01936 | 0.00044 | 2220 | 29500 | 566540 | 414356 | 369612 | 257987 | 100338 | 13197 | 20402 | 4709 | 1053 | 248 | 300 | 174 | 39.3 | 8 | 13.4 |
| 34 | 16.42 | 0.39 | 0.0544 | 0.0012 | 0.01944 | 0.00047 | 1950 | 30300 | 587342 | 407830 | 355603 | 255580 | 100000 | 12824 | 19648 | 4598 | 1041 | 253 | 289 | 166 | 21.9 | 16 | 18.3 |
| 35 | 16.39 | 0.41 | 0.0544 | 0.0012 | 0.01944 | 0.00050 | 2480 | 32400 | 562025 | 412724 | 367457 | 245514 | 113446 | 14725 | 24372 | 6427 | 1439 | 341 | 452 | 231 | 64.4 | 16 | 24.8 |
| 36 | 16.31 | 0.36 | 0.0544 | 0.0013 | 0.01912 | 0.00041 | 2010 | 43900 | 580591 | 402936 | 355603 | 258425 | 90405 | 10533 | 16935 | 4460 | 850 | 208 | 242 | 168 | 49.4 | 3 | 3.3 |
| 37 | 16.34 | 0.44 | 0.0552 | 0.0013 | 0.01948 | 0.00050 | 2200 | 42200 | 582700 | 435563 | 365302 | 247046 | 100811 | 13002 | 21809 | 5512 | 1150 | 239 | 295 | 179 | 33.2 | 35 | 9.3 |
| 38 | 16.54 | 0.38 | 0.0543 | 0.0012 | 0.01913 | 0.00048 | 2148 | 44200 | 571308 | 414356 | 363147 | 241794 | 92770 | 11829 | 18543 | 4645 | 1008 | 214 | 286 | 162 | 49.4 | 24 | 8.9 |
| 39 | 16.39 | 0.44 | 0.0550 | 0.0014 | 0.01960 | 0.00053 | 1419 | 32800 | 565823 | 443719 | 365302 | 249234 | 91014 | 11901 | 16935 | 4598 | 907 | 222 | 267 | 157 | 51.8 | 22 | 6.1 |
| 40 | 16.44 | 0.39 | 0.0542 | 0.0013 | 0.01949 | 0.00047 | 1930 | 31600 | 556540 | 424144 | 382543 | 257112 | 96757 | 12504 | 19598 | 4903 | 1175 | 256 | 326 | 203 | 45.3 | 17 | 20.7 |
| 41 | 16.37 | 0.38 | 0.0548 | 0.0012 | 0.01928 | 0.00048 | 1690 | 40600 | 562447 | 435563 | 387931 | 262363 | 85405 | 11332 | 18342 | 4571 | 1045 | 239 | 333 | 178 | 41.3 | 28 | 13.4 |
| 42 | 16.56 | 0.41 | 0.0544 | 0.0013 | 0.01950 | 0.00044 | 1655 | 34400 | 573418 | 420881 | 350216 | 243107 | 92095 | 12131 | 19246 | 4654 | 1037 | 231 | 321 | 164 | 27.9 | 12 | 0.0 |
| 43 | 16.26 | 0.37 | 0.0545 | 0.0012 | 0.01927 | 0.00045 | 2490 | 37200 | 569198 | 432300 | 383621 | 245952 | 88108 | 10728 | 18291 | 4211 | 915 | 204 | 286 | 164 | 21.5 | 16 | 13.8 |
| 44 | 16.33 | 0.40 | 0.0540 | 0.0012 | 0.01942 | 0.00047 | 2260 | 30300 | 585654 | 437194 | 376078 | 267396 | 95338 | 14369 | 19950 | 4737 | 1089 | 232 | 300 | 160 | 49.4 | 9 | 8.5 |
| 45 | 16.22 | 0.37 | 0.0542 | 0.0013 | 0.01942 | 0.00045 | 1600 | 44700 | 528692 | 414356 | 329741 | 239168 | 87365 | 10906 | 18945 | 4820 | 1057 | 245 | 326 | 183 | 11.7 | 23 | 0.0 |
| 46 | 16.57 | 0.41 | 0.0539 | 0.0013 | 0.01907 | 0.00049 | 1674 | 39800 | 538397 | 402936 | 328664 | 237199 | 85135 | 11812 | 18342 | 4681 | 1089 | 215 | 289 | 195 | 34.4 | 14 | 9.3 |
| 47 | 16.44 | 0.42 | 0.0535 | 0.0012 | 0.01927 | 0.00048 | 1920 | 30300 | 528692 | 417618 | 351293 | 243326 | 94054 | 12771 | 18593 | 5208 | 1000 | 262 | 311 | 180 | 57.5 | 31 | 8.1 |
| 48 | 16.39 | 0.40 | 0.0533 | 0.0012 | 0.01936 | 0.00049 | 1789 | 31400 | 540928 | 446982 | 350216 | 240263 | 89189 | 12220 | 17538 | 4349 | 898 | 202 | 317 | 151 | 21.5 | 19 | 5.7 |
| 49 | 16.21 | 0.37 | 0.0553 | 0.0013 | 0.01947 | 0.00046 | 1810 | 35800 | 570464 | 430669 | 375000 | 262582 | 99459 | 12824 | 21859 | 5125 | 1264 | 283 | 308 | 190 | 24.7 | 8 | 2.4 |

# Table 2B_U-Th_Pb+REE Monazite _McD_S

| Spot | 238U/206Pb | ± | 207Pb/206Pb | ± | 208Pb/232Th | ± | U (ppm) | Th (ppm) | La (ppm) | Ce (ppm) | Pr (ppm) | Nd (ppm) | Sm (ppm) | Eu (ppm) | Gd (ppm) | Tb (ppm) | Dy (ppm) | Y (ppm) | Ho (ppm) | Er (ppm) | Tm (ppm) | Yb (ppm) | Lu (ppm) |
|---|---|---|---|---|---|---|---|---|---|---|---|---|---|---|---|---|---|---|---|---|---|---|---|
| 50 | 16.40 | 0.37 | 0.0543 | 0.0012 | 0.01944 | 0.00045 | 2020 | 31600 | 561603 | 411093 | 340517 | 233479 | 92838 | 12131 | 19196 | 4598 | 1000 | 219 | 337 | 168 | 21.5 | 14 | 16.7 |
| 51 | 16.66 | 0.40 | 0.0534 | 0.0011 | 0.01944 | 0.00047 | 4730 | 33200 | 523207 | 432300 | 303879 | 243545 | 110135 | 14512 | 25879 | 6759 | 1650 | 358 | 575 | 217 | 79.8 | 31 | 7.7 |
| 52 | 16.54 | 0.40 | 0.0540 | 0.00122 | 0.01953 | 0.000469 | 1900 | 36500 | 585654 | 474715 | 359914 | 258425 | 111283.8 | 14547 | 21859 | 5983 | 1341 | 318 | 410 | 239 | 69.6 | 35.4 | 13.4 |
| 53 | 16.61 | 0.42 | 0.0544 | 0.00132 | 0.01891 | 0.000502 | 1596 | 66200 | 545992 | 412398 | 295259 | 232260 | 84932.43 | 11758 | 18593 | 4391 | 1085 | 255 | 308 | 212 | 30.0 | 11.2 | 5.3 |
| 54 | 16.49 | 0.39 | 0.0569 | 0.00129 | 0.01916 | 0.000475 | 1611 | 57800 | 542616 | 437194 | 352371 | 244201 | 100202.7 | 13499 | 20804 | 4931 | 1081 | 232 | 350 | 159 | 36.4 | 21.7 | 8.1 |
| 55 | 16.34 | 0.48 | 0.0537 | 0.00125 | 0.01950 | 0.000545 | 1760 | 30200 | 581013 | 476346 | 371767 | 246827 | 94729.73 | 13357 | 19899 | 4377 | 1024 | 193 | 251 | 169 | 32.8 | 25.5 | 5.7 |
| 56 | 16.80 | 0.40 | 0.0541 | 0.00115 | 0.01912 | 0.000446 | 2240 | 37600 | 536709 | 438825 | 335129 | 235449 | 84797.3 | 11758 | 16985 | 4177 | 996 | 204 | 288 | 164 | 40.9 | 28.6 | 8.1 |
| 57 | 16.49 | 0.41 | 0.0540 | 0.00113 | 0.01979 | 0.000522 | 3610 | 30800 | 526160 | 461664 | 355603 | 247484 | 116621.6 | 13979 | 26533 | 6648 | 1602 | 318 | 447 | 240 | 51.0 | 49.7 | 18.3 |
| 58 | 16.33 | 0.40 | 0.0544 | 0.00121 | 0.01965 | 0.000471 | 2110 | 39100 | 568354 | 464927 | 329741 | 240700 | 80000 | 11279 | 16030 | 3767 | 748 | 165 | 216 | 128 | 38.9 | 16.8 | 0.0 |
| 59 | 16.59 | 0.38 | 0.0542 | 0.0012 | 0.01946 | 0.000442 | 3130 | 33400 | 556118 | 461664 | 344828 | 250985 | 93648.65 | 11883 | 19698 | 4873 | 976 | 222 | 271 | 173 | 42.9 | 13.0 | 19.5 |
| 60 | 16.59 | 0.38 | 0.0540 | 0.0012 | 0.01944 | 0.000447 | 3270 | 30600 | 567511 | 450245 | 322198 | 245733 | 111283.8 | 13854 | 23819 | 5956 | 1524 | 323 | 445 | 236 | 36.4 | 16.8 | 19.5 |
| 61 | 16.55 | 0.42 | 0.0537 | 0.00124 | 0.01905 | 0.000517 | 1860 | 69500 | 540928 | 433931 | 309267 | 224508 | 87702.7 | 12256 | 16683 | 3518 | 752 | 174 | 209 | 156 | 36.4 | 6.8 | 13.4 |
| 62 | 16.14 | 0.40 | 0.0538 | 0.00114 | 0.01945 | 0.00045 | 3700 | 34000 | 579325 | 481240 | 346983 | 233260 | 114189.2 | 14760 | 25226 | 6288 | 1467 | 359 | 498 | 232 | 72.1 | 27.3 | 32.5 |
| 63 | 16.35 | 0.38 | 0.0539 | 0.00121 | 0.01971 | 0.000462 | 2030 | 32300 | 582700 | 477977 | 342672 | 245733 | 94256.76 | 13464 | 18894 | 4343 | 967 | 227 | 255 | 166 | 38.9 | 36.6 | 8.9 |
| 64 | 16.45 | 0.44 | 0.0554 | 0.00135 | 0.01954 | 0.000525 | 1700 | 32600 | 584388 | 461664 | 345905 | 228665 | 100000 | 12398 | 22010 | 5457 | 1248 | 287 | 397 | 194 | 44.9 | 22.4 | 14.2 |
| 65 | 16.59 | 0.40 | 0.0538 | 0.00132 | 0.01921 | 0.00047 | 1476 | 33600 | 602532 | 453507 | 372845 | 246827 | 88648.65 | 12007 | 18794 | 4609 | 988 | 210 | 277 | 182 | 65.2 | 19.9 | 13.8 |
| 66 | 16.37 | 0.41 | 0.0588 | 0.00127 | 0.01931 | 0.000483 | 1900 | 42200 | 518987 | 451876 | 314655 | 231072 | 82297.3 | 12575 | 16432 | 3449 | 833 | 164 | 212 | 144 | 8.9 | 17.4 | 0.0 |
| 67 | 16.36 | 0.40 | 0.0541 | 0.00114 | 0.01942 | 0.000473 | 2520 | 30760 | 554008 | 451876 | 355603 | 263020 | 112702.7 | 15293 | 23266 | 5789 | 1362 | 313 | 401 | 203 | 46.6 | 21.7 | 19.1 |
| 68 | 16.58 | 0.40 | 0.0536 | 0.00114 | 0.01947 | 0.000486 | 4360 | 30500 | 551055 | 445351 | 365302 | 253611 | 106891.9 | 12895 | 21859 | 5180 | 1167 | 253 | 342 | 175 | 54.3 | 10.6 | 2.8 |
| 69 | 16.48 | 0.36 | 0.0544 | 0.00126 | 0.01949 | 0.000443 | 2458 | 38700 | 540506 | 474715 | 376078 | 270460 | 98175.68 | 12860 | 19598 | 4571 | 1020 | 230 | 319 | 166 | 40.5 | 14.3 | 13.0 |
| 70 | 16.53 | 0.39 | 0.0532 | 0.00121 | 0.01933 | 0.0046 | 2109 | 32100 | 576371 | 450408 | 367457 | 266521 | 106081.1 | 13606 | 20251 | 4535 | 1053 | 232 | 317 | 178 | 28.3 | 21.7 | 0.0 |
| 71 | 16.45 | 0.44 | 0.0542 | 0.0013 | 0.01924 | 0.00052 | 1890 | 47000 | 562447 | 438825 | 335129 | 252079 | 98310.81 | 12877 | 19246 | 4321 | 959 | 221 | 321 | 186 | 34.0 | 13.7 | 19.1 |
| 72 | 16.48 | 0.42 | 0.0544 | 0.00118 | 0.01953 | 0.000499 | 1660 | 34700 | 583966 | 491028 | 385776 | 286433 | 96013.51 | 13819 | 19749 | 4488 | 1179 | 283 | 364 | 203 | 44.5 | 15.5 | 8.5 |
| 73 | 16.27 | 0.40 | 0.0557 | 0.00123 | 0.01927 | 0.000501 | 1970 | 86900 | 597046 | 405057 | 315733 | 243107 | 80135.14 | 11545 | 15829 | 3399 | 813 | 159 | 214 | 136 | 31.6 | 16.1 | 11.8 |
| 74 | 16.18 | 0.41 | 0.0540 | 0.00127 | 0.01937 | 0.000484 | 2270 | 37100 | 594937 | 442088 | 334052 | 248578 | 89054.05 | 12078 | 18492 | 3853 | 833 | 192 | 251 | 157 | 15.0 | 16.8 | 13.4 |
| 75 | 16.17 | 0.38 | 0.0535 | 0.00122 | 0.01985 | 0.000459 | 2460 | 31100 | 580591 | 451876 | 365302 | 266740 | 116216.2 | 14174 | 23317 | 5568 | 1276 | 276 | 394 | 232 | 25.5 | 26.1 | 14.6 |
| 76 | 16.11 | 0.38 | 0.0542 | 0.00126 | 0.01946 | 0.000452 | 1563 | 33400 | 582278 | 446982 | 348060 | 269147 | 104729.7 | 14121 | 20251 | 4479 | 984 | 219 | 271 | 175 | 22.3 | 9.3 | 8.9 |

Table 3_U-Th_Pb Zircon sorted by age

| Spot | Location | U (ppm) | Th (ppm) | Th/U | $^{207}Pb/^{235}U$ | 2σ | $^{206}Pb/^{238}U$ | 2σ | $^{207}Pb/^{206}Pb$ | 2σ | ρ | $^{207}Pb/^{235}U$ Age (Ma) | 2σ | $^{206}Pb/^{238}U$ Age (Ma) | 2σ | $^{207}Pb/^{206}Pb$ Age (Ma) | 2σ | Probability of concordance |
|---|---|---|---|---|---|---|---|---|---|---|---|---|---|---|---|---|---|---|
| 3 | Core | 690 | 223 | 0.32 | 10.760 | 0.242 | 0.423 | 0.0093 | 0.184 | 0.004 | -0.05 | 2502 | 9 | 2272 | 18 | 2691 | 6 | 0.91 |
| 72 | | 495 | 22 | 0.04 | 10.230 | 0.273 | 0.421 | 0.0111 | 0.174 | 0.004 | 0.06 | 2455 | 16 | 2264 | 33 | 2596 | 6 | 0.92 |
| 82 | | 280 | 136 | 0.48 | 9.840 | 0.281 | 0.403 | 0.0112 | 0.177 | 0.004 | -0.36 | 2418 | 19 | 2183 | 36 | 2625 | 5 | 0.90 |
| 80 | | 46 | 34 | 0.74 | 6.460 | 0.176 | 0.364 | 0.0095 | 0.128 | 0.003 | -0.06 | 2039 | 17 | 1998 | 29 | 2065 | 8 | 0.98 |
| 40 | | 62 | 40 | 0.64 | 6.390 | 0.182 | 0.358 | 0.0097 | 0.130 | 0.003 | -0.31 | 2030 | 18 | 1974 | 31 | 2095 | 9 | 0.97 |
| 37 | | 151 | 53 | 0.35 | 6.426 | 0.151 | 0.358 | 0.0082 | 0.131 | 0.003 | -0.19 | 2035 | 11 | 1972 | 19 | 2107 | 7 | 0.97 |
| 44 | | 100 | 109 | 1.09 | 6.100 | 0.193 | 0.356 | 0.0108 | 0.124 | 0.003 | -0.14 | 1989 | 21 | 1960 | 38 | 2020 | 8 | 0.99 |
| 2 | Rim | 444 | 141 | 0.32 | 8.210 | 0.198 | 0.355 | 0.0082 | 0.168 | 0.003 | -0.31 | 2256 | 12 | 1958 | 19 | 2542 | 4 | 0.87 |
| 5 | | 70 | 102 | 1.47 | 5.854 | 0.150 | 0.342 | 0.0090 | 0.125 | 0.003 | 0.01 | 1957 | 15 | 1895 | 28 | 2035 | 9 | 0.97 |
| 28 | | 878 | 74 | 0.08 | 7.890 | 0.218 | 0.334 | 0.0090 | 0.173 | 0.004 | -0.07 | 2218 | 18 | 1858 | 30 | 2584 | 10 | 0.84 |
| 48 | | 28 | 15 | 0.53 | 6.030 | 0.251 | 0.334 | 0.0137 | 0.130 | 0.003 | -0.12 | 1976 | 32 | 1856 | 58 | 2098 | 15 | 0.94 |
| 32 | | 101 | 49 | 0.49 | 5.867 | 0.153 | 0.333 | 0.0085 | 0.129 | 0.003 | 0.18 | 1956 | 14 | 1853 | 26 | 2080 | 10 | 0.95 |
| 18 | | 49 | 19 | 0.38 | 5.977 | 0.155 | 0.321 | 0.0083 | 0.137 | 0.003 | 0.03 | 1975 | 16 | 1799 | 23 | 2184 | 12 | 0.91 |
| 4 | | 677 | 299 | 0.44 | 5.659 | 0.146 | 0.316 | 0.0080 | 0.130 | 0.003 | -0.30 | 1924 | 14 | 1767 | 24 | 2103 | 5 | 0.92 |
| 83 | | 48 | 22 | 0.47 | 5.510 | 0.163 | 0.314 | 0.0093 | 0.127 | 0.003 | 0.10 | 1906 | 20 | 1762 | 34 | 2056 | 15 | 0.93 |
| 39 | | 125 | 113 | 0.90 | 5.190 | 0.144 | 0.309 | 0.0082 | 0.122 | 0.003 | -0.39 | 1849 | 17 | 1737 | 27 | 1987 | 9 | 0.94 |
| 49 | | 403 | 115 | 0.29 | 6.840 | 0.196 | 0.307 | 0.0083 | 0.161 | 0.003 | -0.10 | 2090 | 18 | 1732 | 29 | 2466 | 9 | 0.83 |
| 35 | | 182 | 124 | 0.68 | 5.660 | 0.158 | 0.298 | 0.0078 | 0.138 | 0.003 | -0.22 | 1924 | 17 | 1683 | 25 | 2204 | 13 | 0.87 |
| 51 | | 304 | 97 | 0.32 | 5.366 | 0.145 | 0.297 | 0.0079 | 0.131 | 0.003 | -0.13 | 1878 | 16 | 1676 | 26 | 2111 | 5 | 0.89 |
| 66 | | 693 | 223 | 0.32 | 6.470 | 0.176 | 0.295 | 0.0079 | 0.158 | 0.003 | -0.11 | 2041 | 17 | 1664 | 26 | 2433 | 5 | 0.82 |
| 21 | | 587 | 251 | 0.43 | 5.019 | 0.124 | 0.294 | 0.0070 | 0.125 | 0.003 | -0.25 | 1824 | 12 | 1663 | 19 | 2028 | 5 | 0.91 |
| 16 | | 118 | 107 | 0.91 | 5.043 | 0.126 | 0.294 | 0.0071 | 0.125 | 0.003 | -0.23 | 1826 | 13 | 1660 | 20 | 2032 | 9 | 0.91 |
| 65 | | 407 | 6 | 0.02 | 4.481 | 0.111 | 0.270 | 0.0067 | 0.120 | 0.002 | 0.15 | 1730 | 11 | 1539 | 20 | 1956 | 5 | 0.89 |
| 1 | | 52 | 36 | 0.70 | 4.496 | 0.129 | 0.266 | 0.0072 | 0.122 | 0.003 | -0.26 | 1729 | 17 | 1524 | 24 | 1995 | 14 | 0.88 |
| 22 | | 128 | 32 | 0.25 | 4.144 | 0.103 | 0.266 | 0.0065 | 0.115 | 0.002 | -0.13 | 1663 | 12 | 1519 | 19 | 1881 | 10 | 0.91 |
| 24 | | 166 | 71 | 0.43 | 4.140 | 0.154 | 0.260 | 0.0086 | 0.117 | 0.002 | -0.72 | 1660 | 25 | 1490 | 35 | 1910 | 12 | 0.90 |
| 54 | | 66 | 43 | 0.64 | 4.218 | 0.118 | 0.253 | 0.0070 | 0.121 | 0.003 | -0.09 | 1677 | 16 | 1452 | 25 | 1975 | 13 | 0.87 |
| 81 | | 80 | 47 | 0.59 | 4.091 | 0.115 | 0.246 | 0.0068 | 0.120 | 0.003 | 0.00 | 1652 | 16 | 1417 | 24 | 1960 | 12 | 0.86 |
| 23 | | 71 | 51 | 0.71 | 4.060 | 0.162 | 0.244 | 0.0080 | 0.121 | 0.003 | -0.50 | 1650 | 29 | 1406 | 33 | 1979 | 15 | 0.85 |
| 31 | | 41 | 17 | 0.41 | 3.840 | 0.134 | 0.236 | 0.0072 | 0.118 | 0.003 | -0.29 | 1600 | 23 | 1367 | 28 | 1931 | 16 | 0.85 |
| 38 | | 48 | 27 | 0.56 | 2.564 | 0.088 | 0.214 | 0.0072 | 0.087 | 0.002 | 0.00 | 1289 | 20 | 1251 | 31 | 1372 | 20 | 0.97 |
| 58 | | 463 | 39 | 0.08 | 3.354 | 0.099 | 0.210 | 0.0060 | 0.116 | 0.002 | -0.39 | 1496 | 18 | 1228 | 23 | 1903 | 10 | 0.82 |
| 46 | | 275 | 126 | 0.46 | 2.377 | 0.068 | 0.208 | 0.0059 | 0.082 | 0.002 | -0.19 | 1235 | 15 | 1219 | 22 | 1252 | 8 | 0.99 |
| 50 | | 321 | 114 | 0.35 | 3.216 | 0.091 | 0.205 | 0.0059 | 0.115 | 0.002 | -0.13 | 1463 | 16 | 1202 | 23 | 1876 | 8 | 0.82 |
| 9 | | 53 | 45 | 0.84 | 2.485 | 0.076 | 0.170 | 0.0049 | 0.106 | 0.002 | 0.04 | 1267 | 17 | 1011 | 20 | 1724 | 17 | 0.80 |
| 41 | | 36 | 16 | 0.45 | 2.302 | 0.077 | 0.156 | 0.0052 | 0.107 | 0.003 | 0.07 | 1211 | 19 | 935 | 23 | 1754 | 23 | 0.77 |
| 45 | | 65 | 29 | 0.45 | 2.113 | 0.072 | 0.153 | 0.0047 | 0.100 | 0.002 | -0.17 | 1152 | 19 | 919 | 20 | 1620 | 18 | 0.80 |
| 30 | | 270 | 29 | 0.11 | 1.994 | 0.083 | 0.146 | 0.0049 | 0.099 | 0.002 | -0.73 | 1111 | 25 | 878 | 22 | 1607 | 20 | 0.79 |
| 78 | | 157 | 40 | 0.26 | 1.241 | 0.040 | 0.111 | 0.0031 | 0.081 | 0.002 | -0.30 | 818 | 14 | 677 | 13 | 1215 | 19 | 0.83 |
| 77 | | 397 | 18 | 0.05 | 1.284 | 0.042 | 0.111 | 0.0033 | 0.084 | 0.002 | -0.40 | 838 | 15 | 675 | 14 | 1287 | 10 | 0.81 |
| 73 | | 238 | 34 | 0.14 | 1.054 | 0.042 | 0.097 | 0.0028 | 0.077 | 0.002 | -0.57 | 730 | 17 | 597 | 12 | 1121 | 24 | 0.82 |
| 19 | | 224 | 185 | 0.83 | 0.775 | 0.020 | 0.096 | 0.0024 | 0.060 | 0.001 | 0.00 | 582 | 8 | 589 | 9 | 585 | 15 | 1.01 |
| 14 | | 264 | 120 | 0.45 | 0.744 | 0.020 | 0.092 | 0.0024 | 0.059 | 0.001 | 0.06 | 564 | 8 | 567 | 10 | 564 | 18 | 1.00 |
| 69 | | 39 | 33 | 0.84 | 0.687 | 0.022 | 0.089 | 0.0022 | 0.056 | 0.001 | 0.07 | 531 | 10 | 552 | 8 | 466 | 40 | 1.04 |
| 64 | | 409 | 196 | 0.48 | 0.708 | 0.019 | 0.089 | 0.0021 | 0.058 | 0.001 | 0.02 | 544 | 8 | 549 | 7 | 531 | 15 | 1.01 |
| 56 | | 37 | 66 | 1.77 | 0.693 | 0.026 | 0.089 | 0.0025 | 0.057 | 0.002 | -0.24 | 534 | 13 | 548 | 10 | 494 | 49 | 1.02 |
| 52 | | 113 | 49 | 0.44 | 0.720 | 0.022 | 0.088 | 0.0023 | 0.060 | 0.001 | 0.08 | 550 | 9 | 546 | 9 | 591 | 23 | 0.99 |
| 43 | | 119 | 68 | 0.57 | 0.716 | 0.022 | 0.088 | 0.0025 | 0.059 | 0.001 | -0.04 | 548 | 10 | 544 | 11 | 567 | 22 | 0.99 |
| 60 | Core | 148 | 95 | 0.64 | 0.708 | 0.019 | 0.087 | 0.0021 | 0.059 | 0.001 | 0.17 | 545 | 8 | 540 | 7 | 559 | 24 | 0.99 |
| 57 | | 119 | 117 | 0.99 | 0.709 | 0.019 | 0.086 | 0.0024 | 0.059 | 0.001 | 0.20 | 544 | 7 | 532 | 10 | 574 | 24 | 0.98 |
| 62 | | 456 | 176 | 0.39 | 0.691 | 0.018 | 0.086 | 0.0022 | 0.058 | 0.001 | -0.01 | 535 | 8 | 532 | 8 | 539 | 12 | 1.00 |
| 27 | | 61 | 45 | 0.74 | 0.681 | 0.023 | 0.086 | 0.0025 | 0.058 | 0.002 | 0.21 | 527 | 11 | 532 | 11 | 522 | 39 | 1.01 |
| 68 | | 66 | 47 | 0.71 | 0.695 | 0.023 | 0.086 | 0.0023 | 0.058 | 0.001 | 0.04 | 535 | 11 | 531 | 9 | 558 | 33 | 0.99 |
| 42 | | 136 | 114 | 0.84 | 0.692 | 0.020 | 0.086 | 0.0023 | 0.059 | 0.001 | -0.12 | 534 | 9 | 530 | 9 | 551 | 24 | 0.99 |
| 13 | | 184 | 64 | 0.35 | 0.674 | 0.017 | 0.085 | 0.0020 | 0.058 | 0.001 | 0.29 | 523 | 7 | 526 | 7 | 523 | 24 | 1.01 |
| 6 | | 1215 | 931 | 0.77 | 0.694 | 0.016 | 0.085 | 0.0019 | 0.060 | 0.001 | 0.09 | 535 | 5 | 524 | 5 | 588 | 9 | 0.98 |
| 55 | | 215 | 71 | 0.33 | 0.681 | 0.020 | 0.085 | 0.0021 | 0.058 | 0.001 | -0.21 | 527 | 9 | 523 | 8 | 549 | 16 | 0.99 |
| 33 | | 82 | 69 | 0.84 | 0.667 | 0.019 | 0.085 | 0.0025 | 0.058 | 0.001 | 0.10 | 519 | 9 | 523 | 10 | 512 | 26 | 1.01 |
| 12 | | 74 | 55 | 0.75 | 0.656 | 0.018 | 0.084 | 0.0020 | 0.057 | 0.002 | 0.12 | 512 | 8 | 520 | 6 | 468 | 40 | 1.02 |
| 74 | Core | 48 | 38 | 0.78 | 0.675 | 0.021 | 0.084 | 0.0023 | 0.058 | 0.002 | -0.04 | 524 | 10 | 519 | 9 | 520 | 42 | 0.99 |
| 34 | | 296 | 229 | 0.77 | 0.655 | 0.017 | 0.083 | 0.0020 | 0.057 | 0.001 | 0.04 | 512 | 7 | 511 | 7 | 498 | 16 | 1.00 |
| 63 | | 122 | 804 | 6.59 | 0.674 | 0.021 | 0.082 | 0.0022 | 0.059 | 0.001 | -0.03 | 523 | 10 | 510 | 8 | 577 | 29 | 0.97 |
| 47 | | 173 | 2 | 0.01 | 0.637 | 0.018 | 0.081 | 0.0019 | 0.058 | 0.001 | -0.03 | 500 | 8 | 502 | 6 | 522 | 27 | 1.00 |
| 61 | Rim | 166 | 25 | 0.15 | 0.636 | 0.016 | 0.081 | 0.0020 | 0.057 | 0.001 | 0.25 | 500 | 6 | 502 | 7 | 481 | 18 | 1.00 |
| 7 | | 763 | 40 | 0.05 | 0.761 | 0.021 | 0.080 | 0.0021 | 0.069 | 0.001 | -0.49 | 575 | 8 | 497 | 8 | 903 | 13 | 0.86 |
| 67 | | 394 | 4 | 0.01 | 0.635 | 0.015 | 0.079 | 0.0018 | 0.057 | 0.001 | -0.15 | 499 | 5 | 493 | 6 | 500 | 14 | 0.99 |
| 75 | Rim | 222 | 5 | 0.02 | 0.625 | 0.017 | 0.079 | 0.0021 | 0.056 | 0.001 | 0.14 | 493 | 8 | 492 | 8 | 453 | 22 | 1.00 |
| 11 | | 601 | 358 | 0.60 | 0.615 | 0.016 | 0.078 | 0.0020 | 0.057 | 0.001 | 0.02 | 487 | 6 | 486 | 7 | 501 | 13 | 1.00 |
| 26 | | 125 | 103 | 0.82 | 0.604 | 0.015 | 0.078 | 0.0022 | 0.057 | 0.001 | 0.46 | 480 | 5 | 486 | 9 | 488 | 24 | 1.01 |
| 71 | | 301 | 7 | 0.02 | 0.602 | 0.016 | 0.078 | 0.0020 | 0.056 | 0.001 | 0.18 | 478 | 6 | 482 | 7 | 453 | 19 | 1.01 |
| 29 | | 261 | 11 | 0.04 | 0.685 | 0.082 | 0.077 | 0.0038 | 0.065 | 0.005 | -0.90 | 524 | 46 | 479 | 21 | 740 | 140 | 0.90 |
| 10 | | 393 | 91 | 0.23 | 0.582 | 0.016 | 0.075 | 0.0021 | 0.056 | 0.001 | -0.13 | 465 | 7 | 468 | 8 | 463 | 14 | 1.00 |
| 53 | | 253 | 8 | 0.03 | 0.581 | 0.017 | 0.075 | 0.0021 | 0.057 | 0.001 | 0.06 | 465 | 9 | 463 | 9 | 489 | 20 | 1.00 |
| 70 | | 378 | 7 | 0.02 | 0.572 | 0.015 | 0.074 | 0.0019 | 0.055 | 0.001 | 0.11 | 459 | 7 | 458 | 8 | 434 | 17 | 1.00 |
| 17 | | 337 | 12 | 0.03 | 0.535 | 0.013 | 0.072 | 0.0018 | 0.055 | 0.001 | 0.29 | 436 | 6 | 445 | 7 | 419 | 23 | 1.02 |
| 20 | | 258 | 17 | 0.06 | 0.522 | 0.021 | 0.069 | 0.0027 | 0.055 | 0.001 | -0.08 | 426 | 12 | 433 | 14 | 421 | 19 | 1.01 |
| 79 | | 272 | 15 | 0.06 | 0.524 | 0.019 | 0.068 | 0.0023 | 0.056 | 0.001 | -0.08 | 427 | 11 | 422 | 11 | 448 | 23 | 0.99 |
| 76 | | 223 | 27 | 0.12 | 0.487 | 0.015 | 0.066 | 0.0018 | 0.054 | 0.001 | -0.13 | 403 | 9 | 409 | 7 | 357 | 26 | 1.02 |
| 25 | | 325 | 23 | 0.07 | 0.464 | 0.012 | 0.063 | 0.0016 | 0.053 | 0.001 | 0.10 | 387 | 5 | 396 | 6 | 352 | 14 | 1.02 |
| 59 | | 989 | 54 | 0.05 | 0.474 | 0.012 | 0.062 | 0.0015 | 0.055 | 0.001 | 0.17 | 394 | 5 | 388 | 5 | 424 | 13 | 0.99 |
| 15 | | 224 | 30 | 0.13 | 0.446 | 0.011 | 0.061 | 0.0014 | 0.053 | 0.001 | 0.06 | 375 | 5 | 380 | 4 | 337 | 22 | 1.01 |
| 36 | | 1837 | 213 | 0.12 | -0.001 | 0.000 | 0.000 | 0.0000 | 0.617 | 0.013 | -0.46 | -1 | 0 | 0 | 0 | 4547 | 6 | 0.08 |
| 8 | | 937 | 352 | 0.38 | -0.001 | 0.000 | 0.000 | 0.0000 | 0.576 | 0.012 | -0.12 | -1 | 0 | 0 | 0 | 4449 | 0 | 0.08 |

# Table 4A_REE Zircon _McD_S sorted by age

| Spot | Description | U (ppm) | Th (ppm) | Hf (ppm) | La (ppm) | Ce (ppm) | Pr (ppm) | Nd (ppm) | Sm (ppm) | Eu (ppm) | Gd (ppm) | Tb (ppm) | Dy (ppm) | Ho (ppm) | Er (ppm) | Tm (ppm) | Yb (ppm) | Lu (ppm) | Th/U | Yb/Gd | Eu/Eu* | Ce/Sm | Lu/Dy | U/Ce |
|---|---|---|---|---|---|---|---|---|---|---|---|---|---|---|---|---|---|---|---|---|---|---|---|---|
| 8 | c (h) | 937 | 352 | 118738 | 0.13 | 29.04 | 3.49 | 9.41 | 45.27 | 22.38 | 337 | 609 | 1451 | 2381 | 3575 | 4818 | 5839 | 6870 | 0.38 | 29.63 | 0.18 | 2.66 | 0.47 | 53 |
| 36 | c (h) | 1837 | 213 | 191262 | 97.47 | 234.91 | 360.99 | 678.34 | 1304.05 | 710.48 | 980 | 590 | 691 | 934 | 1188 | 1866 | 2845 | 4106 | 0.12 | 0.61 | 0.63 | 0.75 | 0.59 | 13 |
| 15 | r | 224 | 30 | 113786 | 0.04 | 3.33 | 0.67 | 3.06 | 17.91 | 8.35 | 56 | 47 | 46 | 39 | 29 | 26 | 35 | 34 | 0.13 | 0.71 | 0.26 | 0.77 | 0.07 | 110 |
| 59 | c (h) | 989 | 54 | 115728 | 0.38 | 7.01 | 3.60 | 8.71 | 46.49 | 57.19 | 126 | 128 | 115 | 83 | 79 | 62 | 56 | 52 | 0.05 | 0.74 | 0.75 | 0.63 | 0.05 | 230 |
| 25 | r | 325 | 23 | 117864 | 0.04 | 2.64 | 0.48 | 1.38 | 11.35 | 7.99 | 32 | 36 | 36 | 60 | 84 | 155 | 213 | 309 | 0.07 | 4.06 | 0.42 | 0.96 | 0.67 | 200 |
| 76 | r | 223 | 27 | 121068 | 0.03 | 2.95 | 0.91 | 2.74 | 15.81 | 14.39 | 38 | 51 | 57 | 49 | 54 | 69 | 86 | 106 | 0.12 | 1.63 | 0.59 | 0.77 | 0.19 | 123 |
| 79 | r | 272 | 15 | 114466 | 0.05 | 0.86 | 0.68 | 1.88 | 7.84 | 6.04 | 40 | 142 | 277 | 330 | 292 | 231 | 177 | 170 | 0.06 | 2.89 | 0.34 | 0.46 | 0.06 | 514 |
| 20 | r | 258 | 17 | 118447 | 0.02 | 2.20 | 0.60 | 1.77 | 9.59 | 6.75 | 43 | 75 | 142 | 258 | 375 | 607 | 807 | 935 | 0.06 | 40.48 | 0.33 | 0.95 | 0.66 | 191 |
| 17 | c (s) | 337 | 12 | 126214 | 0.06 | 1.89 | 0.52 | 1.53 | 6.22 | 4.97 | 46 | 74 | 208 | 383 | 625 | 1263 | 1870 | 2439 | 0.03 | 29.29 | 0.29 | 1.26 | 1.17 | 291 |
| 70 | r | 378 | 7 | 126117 | 0.08 | 1.45 | 0.10 | 0.96 | 5.07 | 3.91 | 27 | 111 | 271 | 463 | 689 | 927 | 1093 | 1179 | 0.02 | 17.14 | 0.33 | 1.19 | 0.44 | 425 |
| 53 | m | 253 | 8 | 123981 | 0.09 | 1.22 | 0.18 | 0.59 | 4.19 | 3.55 | 35 | 99 | 235 | 372 | 541 | 644 | 652 | 626 | 0.03 | 13.00 | 0.29 | 1.21 | 0.27 | 338 |
| 10 | c (h) | 393 | 91 | 104563 | 0.03 | 57.59 | 0.64 | 2.12 | 9.05 | 6.75 | 66 | 122 | 283 | 504 | 813 | 1316 | 1764 | 2480 | 0.23 | 17.39 | 0.28 | 26.34 | 0.88 | 11 |
| 29 | m | 261 | 11 | 116505 | 0.05 | 3.13 | 0.27 | 0.81 | 3.38 | 3.20 | 32 | 72 | 179 | 295 | 488 | 785 | 1075 | 1488 | 0.04 | 40.00 | 0.31 | 3.84 | 0.83 | 136 |
| 71 | r | 301 | 7 | 123204 | 0.13 | 2.76 | 1.83 | 4.16 | 7.64 | 4.44 | 35 | 136 | 326 | 582 | 938 | 1433 | 1820 | 2276 | 0.02 | 27.14 | 0.27 | 1.50 | 0.70 | 178 |
| 26 | c (s) | 125 | 103 | 94660 | 0.13 | 21.86 | 2.38 | 5.86 | 41.08 | 29.66 | 146 | 227 | 393 | 625 | 901 | 1239 | 1553 | 1976 | 0.82 | 6.39 | 0.38 | 2.20 | 0.50 | 9 |
| 11 | c (h) | 601 | 358 | 131456 | 0.05 | 36.87 | 1.93 | 4.60 | 36.89 | 23.09 | 143 | 211 | 427 | 647 | 1019 | 1615 | 2373 | 3215 | 0.60 | 4.89 | 0.32 | 4.14 | 0.75 | 27 |
| 7 | c (b) | 763 | 40 | 131553 | 0.87 | 9.15 | 10.13 | 11.36 | 28.78 | 28.60 | 106 | 165 | 404 | 661 | 1075 | 1822 | 2658 | 3374 | 0.05 | 9.80 | 0.52 | 1.32 | 0.83 | 136 |
| 67 | c (h) | 394 | 4 | 122330 | 0.05 | 0.46 | 0.50 | 1.86 | 7.50 | 1.60 | 60 | 199 | 313 | 348 | 369 | 340 | 314 | 350 | 0.01 | 15.00 | 0.08 | 0.25 | 0.11 | 1407 |
| 75 | c (o) | 222 | 5 | 119417 | 0.26 | 1.11 | 1.01 | 1.77 | 7.09 | 4.44 | 48 | 157 | 299 | 366 | 412 | 413 | 360 | 423 | 0.02 | 7.65 | 0.24 | 0.65 | 0.14 | 326 |
| 47 | m | 173 | 2 | 120291 | 0.05 | 0.17 | 0.05 | 0.42 | 5.20 | 3.02 | 43 | 124 | 247 | 229 | 193 | 186 | 182 | 199 | 0.01 | 8.67 | 0.20 | 0.14 | 0.08 | 1619 |
| 61 | c (h) | 166 | 25 | 100291 | 0.05 | 12.07 | 0.44 | 1.31 | 10.07 | 7.46 | 46 | 106 | 241 | 366 | 534 | 806 | 1118 | 1398 | 0.15 | 5.52 | 0.35 | 4.97 | 0.58 | 22 |
| 63 | c (h) | 122 | 804 | 22524 | 0.74 | 64.11 | 20.58 | 62.36 | 185.81 | 467.14 | 553 | 861 | 1191 | 1363 | 1688 | 2126 | 2193 | 2309 | 6.59 | 1.11 | 1.46 | 1.43 | 0.19 | 3 |
| 34 | o | 296 | 229 | 109515 | 0.06 | 112.07 | 0.55 | 2.19 | 18.51 | 10.48 | 116 | 249 | 524 | 1007 | 1506 | 2324 | 2969 | 3943 | 0.77 | 24.81 | 0.23 | 25.07 | 0.75 | 4 |
| 74 | m | 48 | 38 | 92427 | 0.08 | 14.71 | 0.56 | 1.53 | 7.57 | 16.16 | 56 | 121 | 267 | 445 | 749 | 1255 | 1503 | 2114 | 0.78 | 9.09 | 0.79 | 8.05 | 0.79 | 5 |
| 12 | o | 74 | 55 | 84175 | 0.05 | 19.90 | 1.48 | 3.63 | 20.34 | 11.90 | 75 | 112 | 241 | 443 | 608 | 915 | 1286 | 1537 | 0.75 | 7.50 | 0.30 | 4.05 | 0.64 | 6 |
| 55 | c (o) | 215 | 71 | 99515 | 0.04 | 16.66 | 0.60 | 175.05 | 14.73 | 10.83 | 67 | 152 | 303 | 496 | 875 | 1215 | 1553 | 2130 | 0.33 | 12.92 | 0.34 | 4.68 | 0.70 | 21 |
| 6 | c (o) | 1215 | 931 | 83883 | 0.33 | 85.32 | 22.41 | 34.14 | 175.00 | 78.51 | 799 | 1127 | 2130 | 3205 | 4256 | 5980 | 7205 | 8699 | 0.77 | 12.47 | 0.21 | 2.02 | 0.41 | 23 |
| 33 | o | 82 | 69 | 77184 | 0.13 | 57.10 | 2.70 | 10.09 | 53.31 | 79.40 | 234 | 335 | 715 | 1081 | 1769 | 2725 | 3534 | 4634 | 0.84 | 7.16 | 0.71 | 4.44 | 0.65 | 2 |
| 13 | c (o) | 184 | 64 | 96602 | 0.01 | 14.68 | 0.51 | 1.20 | 11.08 | 8.88 | 72 | 122 | 262 | 467 | 719 | 1053 | 1360 | 1923 | 0.35 | 10.56 | 0.31 | 5.49 | 0.73 | 20 |
| 42 | c (o) | 136 | 114 | 70291 | 0.15 | 26.59 | 3.59 | 14.22 | 44.26 | 58.97 | 157 | 235 | 382 | 641 | 950 | 1547 | 2000 | 2642 | 0.84 | 6.67 | 0.71 | 2.49 | 0.69 | 8 |
| 68 | c (o) | 66 | 47 | 93981 | 0.07 | 23.00 | 0.38 | 2.41 | 8.65 | 11.55 | 39 | 97 | 172 | 262 | 478 | 652 | 842 | 1224 | 0.71 | 4.69 | 0.63 | 11.02 | 0.71 | 5 |
| 57 | c (o) | 119 | 117 | 74175 | 0.11 | 33.77 | 4.64 | 21.88 | 97.30 | 22.02 | 360 | 615 | 1085 | 1676 | 2350 | 2862 | 3385 | 3817 | 0.99 | 23.91 | 0.12 | 1.44 | 0.35 | 6 |
| 27 | c (s) | 61 | 45 | 90097 | 0.01 | 21.40 | 0.59 | 2.67 | 10.47 | 8.70 | 71 | 103 | 213 | 352 | 523 | 814 | 1056 | 1411 | 0.74 | 6.25 | 0.32 | 8.46 | 0.66 | 5 |
| 62 | c (o) | 456 | 176 | 98058 | 0.05 | 21.70 | 2.33 | 7.26 | 38.45 | 30.02 | 108 | 202 | 365 | 588 | 962 | 1421 | 2087 | 2907 | 0.39 | 4.62 | 0.47 | 2.34 | 0.80 | 34 |
| 60 | m | 148 | 95 | 93107 | 0.12 | 31.32 | 4.38 | 13.37 | 50.68 | 41.03 | 130 | 248 | 442 | 617 | 906 | 1360 | 1646 | 2150 | 0.64 | 6.22 | 0.51 | 2.56 | 0.49 | 8 |
| 43 | c (o) | 119 | 68 | 84466 | 0.05 | 11.00 | 0.75 | 3.94 | 22.77 | 27.18 | 77 | 152 | 326 | 496 | 773 | 1202 | 1559 | 2053 | 0.57 | 6.25 | 0.65 | 2.00 | 0.63 | 18 |
| 52 | c (o) | 113 | 49 | 110874 | 0.02 | 36.87 | 0.55 | 2.10 | 15.68 | 16.16 | 95 | 217 | 480 | 830 | 1400 | 2117 | 2944 | 4024 | 0.44 | 9.09 | 0.42 | 9.74 | 0.84 | 5 |
| 56 | c (s) | 37 | 66 | 90971 | 0.09 | 118.60 | 1.95 | 7.07 | 49.66 | 27.53 | 242 | 402 | 793 | 1201 | 1756 | 2275 | 2522 | 3256 | 1.77 | 13.41 | 0.25 | 9.89 | 0.41 | 1 |
| 64 | c (o) | 409 | 196 | 122136 | 0.05 | 45.35 | 0.86 | 3.41 | 20.27 | 1.78 | 101 | 224 | 480 | 835 | 1513 | 2275 | 2783 | 3638 | 0.48 | 30.00 | 0.04 | 9.27 | 0.76 | 15 |
| 69 | c (h) | 39 | 33 | 100583 | 0.01 | 19.90 | 0.66 | 1.82 | 12.16 | 14.03 | 57 | 121 | 226 | 374 | 623 | 834 | 1118 | 1553 | 0.84 | 5.25 | 0.53 | 6.78 | 0.69 | 3 |
| 14 | c (h) | 264 | 120 | 108835 | 0.01 | 27.08 | 1.31 | 2.47 | 11.35 | 3.91 | 72 | 116 | 253 | 449 | 688 | 1113 | 1404 | 1955 | 0.45 | 13.85 | 0.14 | 9.88 | 0.77 | 16 |
| 73 | r | 238 | 34 | 114369 | 0.09 | 10.28 | 1.86 | 4.97 | 21.62 | 15.81 | 75 | 189 | 289 | 447 | 575 | 810 | 901 | 1150 | 0.14 | 6.36 | 0.39 | 1.97 | 0.40 | 38 |

# Table 4B_REE Zircon _McD_S sorted by age

| Spot | Description | U (ppm) | Th (ppm) | Hf (ppm) | La (ppm) | Ce (ppm) | Pr (ppm) | Nd (ppm) | Sm (ppm) | Eu (ppm) | Gd (ppm) | Tb (ppm) | Dy (ppm) | Ho (ppm) | Er (ppm) | Tm (ppm) | Yb (ppm) | Lu (ppm) | Th/U | Yb/Gd | Eu/Eu* | Ce/Sm | Lu/Dy | U/Ce |
|---|---|---|---|---|---|---|---|---|---|---|---|---|---|---|---|---|---|---|---|---|---|---|---|---|
| 19 | o | 224 | 185 | 100388 | 0.02 | 38.34 | 1.07 | 3.89 | 27.91 | 9.77 | 136 | 252 | 569 | 1004 | 1563 | 2413 | 3019 | 3813 | 0.83 | 34.50 | 0.16 | 5.69 | 0.67 | 10 |
| 77 | m | 397 | 18 | 123301 | 0.38 | 3.07 | 1.47 | 3.00 | 19.12 | 12.43 | 74 | 158 | 205 | 203 | 248 | 331 | 400 | 423 | 0.05 | 2.48 | 0.33 | 0.66 | 0.21 | 211 |
| 78 | r | 157 | 40 | 85631 | 0.18 | 11.75 | 1.50 | 1.95 | 6.01 | 11.19 | 28 | 53 | 89 | 121 | 238 | 416 | 573 | 907 | 0.26 | 3.62 | 0.86 | 8.09 | 1.02 | 22 |
| 30 | r (o) | 270 | 29 | 111068 | 0.04 | 4.26 | 0.24 | 0.94 | 6.15 | 6.22 | 55 | 120 | 256 | 410 | 628 | 955 | 1137 | 1402 | 0.11 | 13.53 | 0.34 | 2.87 | 0.55 | 103 |
| 45 | c (s) | 65 | 29 | 100971 | 0.05 | 7.94 | 0.17 | 1.05 | 9.32 | 8.35 | 26 | 66 | 132 | 194 | 333 | 474 | 609 | 793 | 0.45 | 6.19 | 0.54 | 3.53 | 0.60 | 13 |
| 41 | c (h) | 36 | 16 | 94660 | 0.03 | 15.01 | 0.44 | 1.55 | 7.09 | 9.77 | 47 | 71 | 138 | 225 | 406 | 619 | 814 | 1061 | 0.45 | 8.64 | 0.53 | 8.76 | 0.77 | 4 |
| 9 | o | 53 | 45 | 82718 | 0.03 | 35.73 | 1.76 | 4.99 | 21.35 | 13.68 | 101 | 129 | 269 | 434 | 559 | 789 | 1106 | 1358 | 0.84 | 7.62 | 0.30 | 6.93 | 0.50 | 2 |
| 50 | c (o) | 321 | 114 | 97087 | 0.38 | 18.76 | 0.80 | 3.74 | 14.46 | 18.29 | 59 | 108 | 226 | 379 | 621 | 976 | 1205 | 1565 | 0.35 | 8.21 | 0.62 | 5.37 | 0.69 | 28 |
| 58 | c (o) | 463 | 39 | 113010 | 0.30 | 4.32 | 1.37 | 2.47 | 15.54 | 14.92 | 59 | 124 | 200 | 273 | 375 | 502 | 632 | 850 | 0.08 | 2.94 | 0.49 | 1.15 | 0.43 | 175 |
| 46 | c (h) | 275 | 126 | 105825 | 0.03 | 41.60 | 0.69 | 3.44 | 25.34 | 3.37 | 137 | 260 | 488 | 766 | 1288 | 1753 | 2112 | 2817 | 0.46 | 23.13 | 0.06 | 6.80 | 0.58 | 11 |
| 38 | c (o) | 48 | 27 | 84078 | 0.05 | 13.51 | 1.23 | 4.46 | 22.50 | 9.59 | 107 | 174 | 341 | 549 | 869 | 1150 | 1547 | 2033 | 0.56 | 10.87 | 0.20 | 2.49 | 0.60 | 6 |
| 31 | m | 41 | 17 | 108350 | 0.04 | 15.50 | 0.22 | 0.83 | 6.08 | 4.97 | 13 | 29 | 51 | 104 | 181 | 307 | 492 | 720 | 0.41 | 6.40 | 0.57 | 10.56 | 1.40 | 4 |
| 22 | c (h) | 128 | 32 | 95728 | 0.01 | 1.75 | 0.68 | 2.52 | 9.59 | 1.15 | 26 | 23 | 54 | 61 | 102 | 136 | 167 | 199 | 0.25 | 8.14 | 0.07 | 0.75 | 0.37 | 119 |
| 24 | c (h) | 166 | 71 | 117573 | 0.02 | 6.04 | 1.00 | 3.04 | 17.91 | 4.97 | 86 | 94 | 126 | 132 | 143 | 153 | 151 | 157 | 0.43 | 2.56 | 0.13 | 1.40 | 0.12 | 45 |
| 65 | c (s) | 407 | 6 | 124078 | 0.02 | 0.44 | 0.29 | 0.88 | 13.58 | 2.13 | 89 | 216 | 261 | 218 | 201 | 213 | 239 | 239 | 0.02 | 3.55 | 0.06 | 0.13 | 0.09 | 1508 |
| 81 | c (o) | 80 | 47 | 79709 | 0.08 | 24.96 | 0.89 | 2.54 | 12.97 | 32.68 | 65 | 158 | 296 | 542 | 1025 | 1615 | 2348 | 3687 | 0.59 | 5.96 | 1.12 | 7.97 | 1.25 | 5 |
| 23 | c (o) | 71 | 51 | 90291 | 0.03 | 28.71 | 1.33 | 3.92 | 19.26 | 15.45 | 84 | 120 | 221 | 368 | 566 | 818 | 1236 | 1602 | 0.71 | 6.92 | 0.38 | 6.18 | 0.72 | 4 |
| 54 | m | 66 | 43 | 97379 | 0.31 | 22.51 | 0.82 | 3.79 | 16.28 | 10.48 | 76 | 125 | 205 | 344 | 533 | 794 | 981 | 1329 | 0.64 | 5.45 | 0.30 | 5.73 | 0.65 | 5 |
| 39 | c (o) | 125 | 113 | 92039 | 0.08 | 61.34 | 0.46 | 4.16 | 15.95 | 18.83 | 75 | 134 | 259 | 408 | 631 | 866 | 1261 | 1565 | 0.90 | 6.67 | 0.54 | 15.93 | 0.60 | 3 |
| 1 | c (s) | 52 | 36 | 89126 | 22.83 | 74.39 | 75.75 | 54.05 | 50.00 | 16.87 | 84 | 97 | 213 | 346 | 534 | 798 | 1186 | 1602 | 0.70 | 6.25 | 0.26 | 6.16 | 0.75 | 1 |
| 44 | c (o) | 100 | 109 | 83786 | 0.17 | 44.05 | 1.35 | 6.46 | 32.43 | 27.18 | 125 | 182 | 351 | 526 | 813 | 1097 | 1385 | 1772 | 1.09 | 5.43 | 0.43 | 5.63 | 0.50 | 4 |
| 21 | o | 587 | 251 | 110194 | 0.04 | 20.39 | 0.71 | 1.88 | 9.59 | 5.51 | 54 | 96 | 198 | 379 | 609 | 1000 | 1311 | 1748 | 0.43 | 10.40 | 0.24 | 8.80 | 0.88 | 47 |
| 16 | c (o) | 118 | 107 | 98835 | 8.02 | 83.20 | 86.21 | 78.77 | 76.35 | 27.89 | 118 | 124 | 251 | 425 | 650 | 1016 | 1478 | 2028 | 0.91 | 4.09 | 0.29 | 4.51 | 0.81 | 2 |
| 5 | m | 70 | 102 | 97767 | 0.01 | 25.61 | 0.58 | 1.33 | 5.81 | 4.80 | 36 | 60 | 136 | 222 | 373 | 595 | 786 | 1183 | 1.47 | 9.33 | 0.33 | 18.26 | 0.87 | 4 |
| 83 | c (s) | 48 | 22 | 81942 | 0.02 | 8.65 | 0.32 | 0.77 | 4.80 | 5.15 | 14 | 28 | 41 | 86 | 168 | 232 | 370 | 549 | 0.47 | 4.41 | 0.62 | 7.46 | 1.32 | 9 |
| 80 | c (s) | 46 | 34 | 98738 | 0.04 | 30.67 | 0.41 | 1.47 | 9.66 | 8.70 | 46 | 85 | 150 | 256 | 424 | 725 | 888 | 1240 | 0.74 | 7.08 | 0.41 | 13.15 | 0.83 | 2 |
| 32 | o | 101 | 49 | 103010 | 0.27 | 49.92 | 0.69 | 3.15 | 10.07 | 7.28 | 36 | 71 | 123 | 229 | 406 | 704 | 1075 | 1488 | 0.49 | 13.75 | 0.38 | 20.54 | 1.21 | 3 |
| 40 | m (s) | 62 | 40 | 89709 | 0.02 | 34.26 | 0.56 | 2.04 | 12.84 | 13.85 | 64 | 94 | 185 | 302 | 483 | 704 | 1025 | 1329 | 0.64 | 3.87 | 0.48 | 11.05 | 0.72 | 3 |
| 48 | c (s) | 28 | 15 | 91553 | 0.03 | 13.88 | 0.53 | 1.09 | 3.45 | 6.04 | 21 | 35 | 74 | 100 | 194 | 285 | 395 | 630 | 0.53 | 3.15 | 0.72 | 16.69 | 0.86 | 3 |
| 4 | c (h) | 677 | 299 | 110194 | 0.01 | 21.04 | 0.66 | 1.40 | 9.59 | 4.80 | 72 | 140 | 327 | 579 | 1025 | 1595 | 2329 | 3443 | 0.44 | 32.31 | 0.18 | 9.08 | 1.05 | 52 |
| 37 | c (s) | 151 | 53 | 94951 | 0.01 | 22.02 | 0.64 | 1.68 | 13.85 | 6.22 | 87 | 170 | 343 | 661 | 1150 | 1947 | 2391 | 3610 | 0.35 | 13.85 | 0.18 | 6.59 | 1.05 | 11 |
| 51 | c (o) | 304 | 97 | 107767 | 0.21 | 6.82 | 0.72 | 2.76 | 16.96 | 3.91 | 71 | 145 | 273 | 480 | 800 | 1089 | 1509 | 2167 | 0.32 | 17.65 | 0.11 | 1.67 | 0.79 | 73 |
| 18 | c (o) | 49 | 19 | 91553 | 0.05 | 10.77 | 0.73 | 1.01 | 9.12 | 5.51 | 42 | 80 | 186 | 344 | 500 | 846 | 1081 | 1439 | 0.38 | 17.33 | 0.28 | 4.89 | 0.77 | 7 |
| 35 | o | 182 | 124 | 99515 | 0.08 | 3.38 | 0.53 | 2.54 | 13.31 | 4.09 | 68 | 98 | 170 | 260 | 328 | 478 | 522 | 679 | 0.68 | 10.00 | 0.14 | 1.05 | 0.40 | 88 |
| 66 | c (o) | 693 | 223 | 106699 | 0.23 | 16.97 | 1.29 | 4.75 | 17.16 | 44.23 | 59 | 116 | 204 | 352 | 580 | 781 | 1081 | 1411 | 0.32 | 2.17 | 1.39 | 4.09 | 0.69 | 67 |
| 49 | c (o) | 403 | 115 | 125922 | 0.11 | 2.28 | 0.54 | 2.01 | 20.07 | 1.74 | 93 | 161 | 232 | 251 | 306 | 348 | 337 | 386 | 0.29 | 11.58 | 0.04 | 0.47 | 0.17 | 288 |
| 2 | r | 444 | 141 | 105728 | 0.09 | 8.40 | 1.38 | 2.54 | 15.68 | 7.82 | 90 | 129 | 267 | 425 | 611 | 899 | 1118 | 1610 | 0.32 | 8.18 | 0.21 | 2.22 | 0.60 | 86 |
| 28 | c (h) | 878 | 74 | 98350 | 0.48 | 11.97 | 4.04 | 6.48 | 16.82 | 8.88 | 35 | 61 | 97 | 171 | 259 | 445 | 646 | 878 | 0.08 | 7.78 | 0.37 | 2.95 | 0.90 | 120 |
| 72 | r | 495 | 22 | 110194 | 0.03 | 7.26 | 0.39 | 0.61 | 4.73 | 3.37 | 34 | 65 | 145 | 229 | 432 | 700 | 994 | 1504 | 0.04 | 16.92 | 0.27 | 6.36 | 1.04 | 111 |
| 82 | c (b) | 280 | 136 | 129029 | 0.07 | 3.34 | 0.82 | 3.22 | 22.16 | 2.49 | 76 | 151 | 215 | 242 | 269 | 308 | 269 | 250 | 0.48 | 9.17 | 0.06 | 0.63 | 0.12 | 137 |
| 3 | c (h) | 690 | 223 | 96505 | 0.10 | 12.72 | 2.78 | 7.72 | 48.58 | 15.28 | 233 | 280 | 577 | 837 | 1231 | 1846 | 2304 | 2959 | 0.32 | 10.91 | 0.14 | 1.08 | 0.51 | 88 |