# Peer review of "Unravelling the origins and P-T-t evolution of the allochthonous Sobrado unit (Órdenes complex, NW Iberia) using combined U-Pb titanite, monazite and zircon geochronology and REE geochemistry José Manuel Benítez-Pérez1,3, Pedro Castiñeiras2, Juan Gómez-Barreiro3 (\*), José R. Martínez Catalán3, Andrew Kylander-Clark4, Robert Holdsworth5"

_Solid Earth, 2020_

## Referee Comment (RC1) · Anonymous Referee #1 · 12 May 2020

Review of se-2020-38: "Unravelling the origins and P-T-t evolution of the allochthonous Sobrado unit (Órdenes Complex, NW Iberia) using combined U-Pb titanite, monazite and zircon geochronology and REE geochemistry" by Benítez-Pérez et al.

General comments The ms provides with new geochronological data that contribute to the knowledge of the tectonometamorphic evolution of a very complex Paleozoic orogenic area. The combination with REE analyses of the dated minerals is relevant to

constrain the interpretation of the obtained ages. This approach is particularly useful for the zircon ages, which show a great dispersal. The ms is concise and well written, and figures are appropriate (but some improvements can be made). I recommend publication after corrections, suggestions and clarifications are made. Section 2 (geological background) is too concise for a non local reader, it should be partially rewritten and extended (see specific comments below). A final discussion/conclusion might underline how the new data strengthen the understanding of the studied region. Large tables can be provided as supplement to the ms.

Specific comments P3 L3: Upper Allochthon detached from Gondwana during Cambro-Ordovician rifting; but both zircon aliquots (mean ages 490 and 530 Ma) are interpreted in a magmatic arc setting (see abstract, discussion and conclusions). L5: what does mean "oceanic supracrustal sequences"? something that overlies the oceanic crust? L24: these ophiolitic rocks belong to the Upper Allochthon? The referred ages are protolith or metamorphic? Give details. L25: these ages correspond to the Upper Allochthon? Are they prograde metamorphic ages? Give details. L26: thrust wedge collapse was coeval to continental subduction in the Upper Allochthon? Clarify. L27: which internal zones? L30: regional oroclinal bending in Iberia is under discussion (see also Pastor Galan...); leave it aside. L33: the study focuses on two units (Sobrado and?). L34: reference to this HP/HT event has been made in the previous paragraph. The older (Ordovician?) granulite facies event is not mentioned in this section, and it has relevance to the paper. Rewrite these paragraphs to be more comprehensive: give first a detailed description of previous geochronological data of the different units, then the preferred tectonometamorphic evolution. L37: lithological description of Sobrado horses is too succint, more rocks appear in legend of Fig 1B. L38: lower slice: are these rocks ophiolitic? Could them belong to the underlying Middle Allochthon? P4 L11: Fornas in Fig 1? P4 L32-36: transfer to next section (mineral description) and rearrange. P6 L43: are there discordant analyses? (those with >10% have to be rejected, and display them with a different color in Fig 6), also in Fig 6 add an age histogram with probablity density including all concordant ages. P6 L46: Fig 7A

does not show the 380-500 Ma aliquot (why?), as Fig 7B do (see fig caption); use the same colors than in the following figures for the two aliquots, idem in Fig 11. P7 L1: you suggests "inheritance" (likely for <600 and possible for >600 Ma, why?). According to the interpretation, zircons older than 500 Ma (MDA) must be inherited. P7 L14: age results for the third aliquot (>600 Ma) have to be described here. P7 L17: REE patterns of zircons older than 600 Ma are not shown anywhere; for comparison, include them in Fig 9 (or at least make reference to a Table). P9 L16: slope from 486 to 380 Ma (Fig 11A). L16-19: you mean such a protracted evolution caused U/Pb to open in the zircons formed during the 490 Ma granulite event. But older zircons were not affected? if yes, the inherited ages (including the 530 Ma median age) are misleading. L21: 502 Ma? This age is used in Fig 11A (380-500 Ma aliquot) to obtain a metamorphic 490 Ma median age. Grain n° 61 belongs to the inherited igneous 500-600 Ma aliquot (Fig 2). Is it grain 61 the 510 Ma zircon in Fig 11B? If yes, MDA is 510 instead of 502 Ma. L39: inherited zircons older than 600 Ma. L40: I would not name "population" to a set of only 2-3 data. L46: WAC is an unlike source for Mesoproterozoic zircons. Gutierrez Alonso et al. (2003) sourced zircons of this period far from Amazonia, not from NE Africa cratons as in more recent interpretations (update reference). P10 L18: evolving to amphibolite-facies.

Technical corrections Do not use plural in the studied samples (a paragneiss, an amphibolite). P2 L16: facies. P2 L21, 32: luminescence P2 L22: petrologenetic P2 L47: . . P3 L8, 11: Autochthon P3 L14: associed P4 L11-12: detachtment P4 L19: parts of P4 L16: xenocrystic P5 L32, 33: $\mu$m P5 L47: 2015b P8 L20: negative negative Arrow in Fig 1C: Corredoiras detachment? Fig 6: correct position of 1800 in the corcordia line.

---

## Referee Comment (RC2) · Michele Zucali (Referee) · 13 May 2020

»Overall quality of the discussion paper ("general comments"):

The manuscript by title: Unravelling the origins and P-T-t evolution of the allochthonous Sobrado unit (Órdenes Complex, NW Iberia) using combined U-Pb titanite, monazite and zircon geochronology and REE geochemistry by José Manuel Benítez-Pérez et al. introduces interesting age data and discussion regarding the Paleozoic evolution of the

[Figure]

Sobrado unit.

The new data are of paramount importance in the understanding of this local tectonic setting as well in the general (global) framework of the Paleozoic geology.

Since this paper addresses such a general question, it somehow needs to better introduce the general geology and associated data, as age and PT conditions. It is partly attempted in the background chapter but still needs a clearer (simpler?) presentation. As a non-expert in the regional geology, I've found myself lost through the text when trying to put the data at the right place in space (map), time (age), and former position (P-T conditions). Check out and simplify the use of different terms and subdivisions (e.g., units, horses, slices, etc...).

»Individual scientific questions/issues ("specific comments")

1) Age Interpretations chapter needs profound changes (see notes on the pdf file) 2) Figures need some work; here a few details as well other on the pdf. Figures, in general, should be re-think and make them better fitting in the manuscript. Here some notes about the figures.

2.1) I really miss images of the rock, thin sections images where the mineral assemblage is shown, microstructural relations are discussed, and analyzed mineral are located in the microstructural frame. Forging the base of the interpretation of the ages. 2.2) I also think that the geological background data, in terms of P-T conditions and available ages, it may be conveniently shown in a synoptic diagram, which will turn useful for successive inclusion of your novel data and general discussion.

2.3) Figure1. I love the details on the map, BUT probably they are too much for this contribution. Don't you think? ==> within the three horses of the Sobrado, the map details different lithologic types (hard to distinguish on the map, by the way) and tectonic contacts with cinematic and so on. Those are not further used in the manuscript, either in the geological background or the discussion. It would probably be more useful and

handy a simplified map. > check the consistency between FIGURE 1 and Geological background.

2.4) FIGURE 2 It is used in two steps: first, at the mineral description paragraph, describing morphologies and zoning patterns; second, when discussing ages and their relations with the zircon patterns. The figure mirrors this double use (age groups and morphologies) but not the captions, too poor. Besides, depending on the size of the image in the final manuscript, several grains might result in small. So resize accordingly to your ideal print size.

3) some suggested references for general background and discussion: Manzotti et al. 2012 - Lithos 146–147 (2012) 276–292 Roda et al 2018 - Lithos 310-311 (2018) 31–49 Manzotti et al 2017 - Swiss J Geosci - DOI.10.1007/s00015-017-0284-1 Jouffray et al. 2020 -> International Journal of Earth Sciences - https://doi.org/10.1007/s00531-020-01848-2

» compact listing of purely technical corrections at the very end ("technical corrections": typing errors, etc.) The noted pdf file reports several technical corrections along with scientific questions.

Please also note the supplement to this comment:
https://www.solid-earth-discuss.net/se-2020-38/se-2020-38-RC2-supplement.pdf

**Supplement:**

[revised manuscript text omitted]
 | 0.10 | 12.72 | 2.78 | 7.72 | 48.58 | 15.28 | 233 | 280 | 577 | 837 | 1231 | 1846 | 2304 | 2959 | 0.32 | 10.91 | 0.14 | 1.08 | 0.51 | 88 |

---

## Author Comment (AC1) · 31 May 2020

We want to thank the constructive suggestions and work made by Anonymous Referee #1.

We will reply here the general questions. We consider most of the suggestions are reasonable and will be accepted in a corrected version of the paper.

[Figure]

The geological background section has been enlarged to provide non local reader with a better perspective of the context. It has now two sub-sections, one devoted to the NW Allochthonous complexes and the other to the structure and rock association of the Sobrado Unit. The number of significant references has been increased to cover al aspects of structural and metamorphic evolution. In a similar way discussion will be reinforced to show the importance of the results.

Specific comments

P3L3: Upper Allochthon detached from Gondwana during Cambro-Ordovician rifting; but both zircon aliquots (mean ages 490 and 530 Ma) are interpreted in a magmatic arc setting (see abstract, discussion and conclusions).

The dual interpretation (rift and arc environments) has been explained in previous contributions and the explanation is included in the new version of the Geological setting: The arc in front of the Iapetus Ocean created a back-arc which evolved to open the Rheic Ocean. The two settings correspond to the two margins of the detached future Upper Allochthon: convergent in the Iapetus side, divergent in the Rheic side.

P3L5: what does mean "oceanic supracrustal sequences"? something that overlies the oceanic crust?

No: metabasites interlayered with metasediments probably representing upper crustal oceanic or transitional oceanic/continental sequences. The term was confusing and has been suppressed and replaced by the description of the previous sentence. P3L24: these ophiolitic rocks belong to the Upper Allochthon? The referred ages are protolith or metamorphic? Give details

The ophiolitic rocks are part of the Middle allochthon. A more detailed description is provided in the corrected version.

P3L25: these ages correspond to the Upper Allochthon? Are they prograde metamorphic ages? Give details

No, these are ages from the basal allochthon, related to HP-LT metamorphism. Made clear in the new version.

P3L26: thrust wedge collapse was coeval to continental subduction in the Upper Allochthon? Clarify.

No, this age refers to middle and basal allochthon. Extension and partial collapse in the upper parts of the accretionary wedge occurred during the incorporation of new units at its lower part. Included in the new version.

P3L27: which internal zones?

Variscan orogen internal zones in the Autochthon. Deleted and better explained in the new version.

P3L30: regional oroclinal bending in Iberia is under discussion (see also Pastor Galan...); leave it aside. Done.

P3L33: the study focuses on two units (Sobrado and?).

Only Sobrado unit. It is an error and has been corrected.

P3L34: reference to this HP/HT event has been made in the previous paragraph. The older(Ordovician?) granulite facies event is not mentioned in this section, and it has relevance to the paper. Rewrite these paragraphs to be more comprehensive: give first a detailed description of previous geochronological data of the different units, then the preferred tectonometamorphic evolution.

It has been rewritten and clarified.

P3L37: lithological description of Sobrado horses is too succint, more rocks appear in legend of Fig 1B.

In principle, only those types relevant for the research had been described. The new text briefly describe other lithologies, mainly those of the overlying Corredoiras unit.

P3L38: lower slice: are these rocks ophiolitic? Could them belong to the underlying Middle Allochthon?

They could be ophiolitic, but this has not been established. Probably represent a transitional crust or at least implies some kind of oceanization. hBut in any case, it is the lower horse of the Sobrado unit, belongs to the Upper Allochthon and is not part of the Middle Allochthon.

P4L11: Fornas in Fig 1?

Not included because it occurs some 30 km to the W, which is now explained in the text. The references of Gómez Barreiro and Álvarez Valero describe this structure.

P4L32-36: transfer to next section (mineral description) and rearrange.

P6 L43: are there discordant analyses? (those with >10% have to be rejected, and display them with a different color in Fig 6), also in Fig 6 add an age histogram with probablity density including all concordant ages.

Firstly, we reject analysis based on high common Pb content or clear analytical errors. However, we like to report the discordant analyses (even if we don't use them in the age calculation). However, the reviewer is right, they should be displayed with a different color. We will modify this in the corrected version of the manuscript.

P6 L46: Fig 7A does not show the 380-500 Ma aliquot (why?), as Fig 7B do (see fig caption); use the same colors than in the following figures for the two aliquots, idem in Fig 11. We agree, it will be solved in a new version of the figure.

P7 L1: you suggests "inheritance" (likely for <600 and possible for >600 Ma, why?). According to the interpretation, zircons older than 500 Ma (MDA) must be inherited.

This is a mistake.

P7 L14: age results for the third aliquot (>600 Ma) have to be described here. Yes, they will be described

P7 L17: REE patterns of zircons older than 600 Ma are not shown anywhere; for comparison, include them in Fig 9 (or at least make reference to a Table). We agree and they will be included in the new version of the manuscript.

P9 L16: slope from 486 to 380 Ma (Fig 11A). That's true, another end is needed for defining the slope

P9 L16-19: you mean such a protracted evolution caused U/Pb to open in the zircons formed during the 490 Ma granulite event. But older zircons were not affected? if yes, the inherited ages (including the 530 Ma median age) are misleading.

It seems like the lead loss only affected to zircons being formed at that time. Older zircon was unaffected or shielded as inclusions in rock-forming minerals

P9 L21: 502 Ma? This age is used in Fig 11A (380-500 Ma aliquot) to obtain a meta-morphic 490 Ma median age. Grain n_ 61 belongs to the inherited igneous 500-600 Ma aliquot (Fig 2). Is it grain 61 the 510 Ma zircon in Fig 11B? If yes, MDA is 510 instead of 502 Ma.

This part has been corrected and rewritten in the new version of the manuscript.

P9 L39: inherited zircons older than 600 Ma. Ok

L40: I would not name "population" to a set of only 2-3 data. Ok

P9 L46: WAC is an unlike source for Mesoproterozoic zircons. Gutierrez Alonso et al. (2003) sourced zircons of this period far from Amazonia, not from NE Africa cratons as in more recent interpretations (update reference).

Reference updated

P10 L18: evolving to amphibolite-facies.

Done

Technical corrections

Do not use plural in the studied samples (a paragneiss, an amphibolite).

P2 L16: facies. Done P2 L21, 32: luminescence Done P2 L22: petrologenetic Done P2 L47: . . Done P3 L8, 11: Autochthon Done P3 L14: associted Done P4 L11-12: detachtment Done P4 L19: parts of P4 L16: xenocrystic. It is correct here

P5 L32, 33: _m Done P5 L47: 2015b Done P8 L20: negative negative Done Arrow in Fig 1C: Corredoiras detachment? Yes, included

Fig 6: correct position of 1800 in the corcordia line. Done

---

## Author Comment (AC2) · 31 May 2020

Reply to the comments by M- Zucali, Referee #2 (13 May 2020):

The new data are of paramount importance in the understanding of this local tectonic setting as well in the general (global) framework of the Paleozoic geology. Since this paper addresses such a general question, it somehow needs to better introduce the general geology and associated data, as age and PT conditions. It is partly attempted

in the background chapter but still needs a clearer (simpler?) presentation. As a non-expert in the regional geology, I've found myself lost through the text when trying to put the data at the right place in space (map), time (age), and former position (P-T conditions). Check out and simplify the use of different terms and subdivisions (e.g., units, horses, slices, etc...).

We kindly appreciate the work and constructive suggestions made by Prof. Zucali. Most of then will be incorporated in the corrected version of the manuscript. Here we will try to reply some general comments. We agree with the referee, geological background needs to be improved to reach a wider audience and a better understanding of the context. He wants it made more clear and shorter, but resuming a complex history is not easy and one has to choose between understanding and length. In our original manuscript, we made a rather simple text for the setting, but it was enough to arouse curiosity of Reviewer #1, who found things poorly explained. The new geological setting is now clearer and richer, including more age data and references, as Prof. Zucali too demands. But, sorry, it is somewhat longer. It seems that some readers will be interested in more regional information than others. A short Geological setting would not satisfy the first group, while a wider one can always be skipped by the latter.

ÂżIndividual scientific questions/issues ("specific comments") 1) Age Interpretations chapter needs profound changes (see notes on the pdf file)

Some problems have been detected in some figures that will be corrected and explained in the corrected version.

2) Figures need some work; here a few details as well other on the pdf. Figures, in general, should be re-think and make them better fitting in the manuscript. Here some notes about the figures.

2.1) I really miss images of the rock, thin sections images where the mineral assemblage is shown, microstructural relations are discussed, and analyzed mineral are located in the microstructural frame. Forging the base of the interpretation of the ages.

The minerals that we have used are not very informative in terms of textural relationships. The idea of REE-U-Pb combination is an attempt to solve this lack of information. Most of mineral assemblages have been previously described in previous references and the interested reader is referred to them.

2.2) I also think that the geological background data, in terms of P-T conditions and available ages, it may be conveniently shown in a synoptic diagram, which will turn useful for successive inclusion of your novel data and general discussion. A reference to Martínez Catalán et al. (2020) has been added in the Geological setting. There, a figure offers a synoptic resume of the NW Iberian Allochthon, Parautochthon and Autochthon. Furthermore, more ages, references and explanations have been included in the Geological setting after recommendations made by Reviewer #1 and Prof. Zucali on its annotated manuscript.

2.3) Figure1. I love the details on the map, BUT probably they are too much for this contribution. Don't you think? ==> within the three horses of the Sobrado, the map details different lithologic types (hard to distinguish on the map, by the way) and tectonic contacts with cinematic and so on. Those are not further used in the manuscript, either in the geological background or the discussion. It would probably be more useful and handy a simplified map. > check the consistency between FIGURE 1 and Geological background.

The geological background has been improved and enlarged in relation with comments and suggestions posed by Reviewer #1, which seemed interested in a better explanation of the whole history of the NW Iberian Allochthon and of Figure 1. One does not need to enter in the complexity of the figure if not feeling like, but may be others do.

2.4) FIGURE 2 It is used in two steps: first, at the mineral description paragraph, describing morphologies and zoning patterns; second, when discussing ages and their relations with the zircon patterns. The figure mirrors this double use (age groups and morphologies) but not the captions, too poor. Besides, depending on the size of the

image in the final manuscript, several grains might result in small. So resize accordingly to your ideal print size.

Captions will be improved and figure resolution will be adapted for a better visualization in an on-line journal if needed.

3) some suggested references for general background and discussion: Manzotti et al. 2012 - Lithos 146–147 (2012) 276–292 Roda et al 2018 - Lithos 310-311 (2018) 31–49 Manzotti et al 2017 - Swiss J Geosci - DOI.10.1007/s00015-017-0284-1 Jouffray et al. 2020 -> International Journal of Earth Sciences - https://doi.org/10.1007/s00531-020-01848-2

Thanks for the suggestions. We will consider including some of the most relevant along the discussion.

Answers to comments made on the PDF: P1 L37: No, TuffZirc will be deleted P1 L42: Same as above P2 L24-25: references incorporated into the new version P5 L1-9: We will elaborate a new figure to clarify this point. P6 L10: We consider it's good practice explaining the process that we have followed in interpreting the analysis. Maybe we can specify that the analyses were considered no further for the calculation of the age. P7 L8: We thank prof. Zucali to point it out this problem. A complete table will be included in the corrected manuscript. P7 L12-13: Agree. An explanation will be included and that zircon removed. P7 L22: It will be changed by "This anomalous Ce content is related to the presence of water in the moment of zircon growth". P7 L25: No. P7 L34: We have substituted "magmatic signature" by "strongly fractionated pattern usually interpreted as magmatic". P7 L42: Yes, there is. Barth and Wooden (2010). We will change the reference. P7 L46: We will explain in a clearer way the evolution of the zircons. P8 L28: done P8 L30: Ok. P8 L38: 1. We will clarify this point. 2. You are right, TuffZirc is not a method. It's an algorythm. P9 L29: We will make a new figure to clarify the evolution and regional interpretation of ages.

---

## Editor Comment (EC1) · Puy Ayarza (Editor) · 4 Jun 2020

Dear authors,

after considering your response to the reviewers, I think you have addressed most issues properly. If something, I agree with reviewer 2 in how helpful it would be to add a figure with a thin section where structural relatioships can be studied. I also agree with reviewer 1 in the need to add details to the complicated Geological Setting of the

area, although it might turn out to be a bit overwhelming for some readers. Accordingly, I'd like you to upload the revised versión of your MS at your earliest convinence

---

## Author Comment (AC3) · 4 Jun 2020

Those changes will be incorporated in the revised version

---

## Editor Decision (ED1)

[revised manuscript text omitted]
 | 0.10 | 12.72 | 2.78 | 7.72 | 48.58 | 15.28 | 233 | 280 | 577 | 837 | 1231 | 1846 | 2304 | 2959 | 0.32 | 10.91 | 0.14 | 1.08 | 0.51 | 88 |

---

## Author Response (AR2)

**Final Response**

We have produced a new version of the manuscript incorporating the suggestions of both editors and referees. We kindly acknowledge their constructive comments to make our results clearer and improve the quality of the manuscript.

We have structured the final reply presenting in blue the comments and in red the author response. Changes in the manuscript have been marked as MS Word changes showing the modifications. Three new figures have been incorporated (1, 3 and 4) and the order of several ones has been modified to ensure a proper presentation of the information.

F. Rossetti's comments:

(i) It is necessary to better introduce the geological problem and significance of the Sobrado unit in the regional context; (ii) It is necessary to document the significance of the studied samples and their representativeness at the regional scale. Which the criteria for sample selection?

We have included a new figure 1 to deal with the regional geological scenario, and rewritten part of the introduction to solve those problems following your comments in the annotated PDF. It is better explained the part of the unit that has been sampled. Besides, both rocks represent the typical metamorphic product of the exhumation, not only in the Sobrado Unit, but in most of the Allochthon. Besides, we do not agree to include the analytical details for each published age, since we include temporal ranges and not specific ages with the errors. The references are there to give the interested reader those details.

(iii) It is compulsory to link the dated accessory minerals to the microtextural fabrics of the studied samples. This request was already posed by Rev#2, acknowledged in the Author's response letter but not accomplished in the revised version. Providing a better assessment of the microtextural fabrics and the reaction textures involving the main and the accessory mineral phases would add strength to the proposed reconstruction (despite feasible). This is in particular relevant for the growth stages of the Ti-phases (Rt, Ilm, Ttn) and monazite.

We have included two new figures (3 and 4) as part of the microtextural analysis. We have focused on Ttn, Mnz and Zrn and their textural microtextural setting in the fabric. We understand that a deeper analysis is far from the scope of this work and would require a completely different strategy (e.g. *in-situ* dating techniques etc). In our case, microstructural data provide a reasonable context to combine with REE and U-Pb data strengthening the interpretation.

Which the evidence for monazite and titanite re-crystallisation? Why excluding a syn-tectonic growth?

This was confusion. We have change the term by crystallization. Microstructural analysis completes, to some extend, deformation-blastesis relationships and it is clear that most of the Ttn is synkinematic, and Mnz is probably coeval with the main foliation.

(iv) Data are mixed with inferences and this should be avoided.

Where detected we have rewritten and reorganized the information.

**Puy Ayarza's comments:**

I am uploading an annottated MS with just a few issues concerning the order in which the authors cite figures,

Corrected and reorganized or the lack of coherence regarging the random use of upper/lower case for certain names.

Corrected

[revised manuscript text omitted]
. (e) Titanite mineral fish showing a top-to-the NNE shear sense. Small ilmenite inclusions are visible in titanite. (f) Retrogression of plagioclase into zoisite and ilmenite inclusions inside titanite grains parallel to the main foliation. All micrographs in in plane-polarized light (PPL). (Mineral abbreviations after Whitney and Evans, 2010; Hbl: hornblende, Ttn: titanite, Ilm: ilmenite, Rt: rutile, Pl: plagioclase, Zo: zoisite, Grt: garnet*)*.

Figure 4. Granulite facies migmatitic paragneiss from the Sobrado Middle horse (Fig. 2). (a) General microstructure where leucosome bands and preferred orientation of elongated garnets, biotite, rutile and kyanite define the foliation. (b) Zircon elongated grain in recrystallized leucosome domain. Main inclusions in garnet include rutile, ilmenite and zircon (PPL) and (c) Cross-polarized ligth (CPL). (d) Monazite and zircon subrounded grain in the leucosome domain. (e) Subrounded elongated grain of zircon included in a garnet. (f) Plastically deformed garnet (sigmoidal grain) in a leucosome. Zircon and monazite are found in the quartz-rich area around the garnet. (g) Sigmoidal garnet in a leucosome. Inclusions of prismatic and bipiramidal zircons and rutile are observed. (h) Monazite grain in a garnet pressure shadow with biotite. Ky: kyanite, Bt: biotite, Qtz: quartz, Kfs: K-felspard, Pl: plagioclase, Rt: rutile, Zrn: zircon, Mnz: monazite, Grt: garnet.

Figure 5. (a) Monazite grain compositional maps in paragneiss with a 30% thorium variation. Location and spot numbers (46 and 47) are indicated, as well as the $^{206}Pb/^{238}U$ age and error ($\pm 2\sigma$). (b) Chondrite-normalized rare earth element (REE) patterns for the same monazites in (a).

Figure 6. Cathodoluminescence (CL) images with the location of the analyzed spots for selected zircon grains.

Figure 7. (a) Tera-Wasserburg diagram showing distribution of analysed titanites (n = 51) from Sobrado amphibolite (JBP-71-21). The rejected analyses are represented by gray ellipses. The ellipses represent the $^{207}Pb/^{206}Pb$ and $^{238}U/^{206}Pb$ errors ($\pm 2\sigma$). (b) Chondrite-normalized rare earth element (REE) patterns for the same titanites.

Figure 8. (a) Tera-Wasserburg diagram showing distribution of analysed monazites (n = 76) from Sobrado paragneiss (JBP-71-15). The rejected analyses are represented by gray ellipses. The ellipses represent the $^{207}$Pb/$^{206}$Pb and $^{238}$U/$^{206}$Pb errors ($\pm 2\sigma$). (b) Chondrite-normalized rare earth element (REE) patterns for the same monazites.

Figure 9. Concordia plot (a) including all zircon with concordance >90% from sample JBP-71-15 (Sobrado migmatitic paragneiss), and (b) age histogram and probability density plot for ages older than 1000 Ma.

Figure 10. Tera-Wasserburg diagram (a) for the analyses between 589 and 380 Ma, and, (b) age histogram and probability density plot for the same ages.

Figure 11. Chondrite-normalized plots for inherited zircon older than 1000 Ma.

Figure 12. Chondrite-normalized plots for zircon between 589 and 380 Ma.

Figure 13. (a) Hafnium versus age, (b) Yb/Gd versus age, (c) Eu/Eu* versus age, and (d) U/Ce versus Th for zircon analyses between 589 and 380 Ma.

Figure 14. Th/U ratio versus $^{206}$Pb/$^{238}$U ages for the zircon analyses from 589 to 380 Ma. Analysis 63 (510 Ma) is not represented as it has an anomalous value (6.59).

Figure 15. Weigthed average obtained from magmatic ages distributed between 589 and 510 Ma.

Figure 16. Age of the onset of the oldest HP-HT metamorphic event obtained using the TuffZirc algorythm.

Table 1: U-Th_Pb+REE Titanite_McD_S

Table 2A: U-Th_Pb+REE Monazite_McD_S

Table 3: U-Th_Pb Zircon sorted by age

Table 4A: REE Zircon_McD_S sorted by age

Table 4B: REE Zircon_McD_S sorted by age

---

## Editor Decision (ED2)

[revised manuscript text omitted]
 | 0.10 | 12.72 | 2.78 | 7.72 | 48.58 | 15.28 | 233 | 280 | 577 | 837 | 1231 | 1846 | 2304 | 2959 | 0.32 | 10.91 | 0.14 | 1.08 | 0.51 | 88 |

---

## Author Response (AR3)

FINAL RESPONSE

We have elaborated a corrected version of the manuscript following the indications of the editor. Minor changes/typos included in the commented PDF have been incorporated. Our replies to some comments are included below. We kindly appreciate the constructive comments of the editor to improve the quality of the manuscript.

New figures: Fig. 2 and 3

F. Rossetti's comments:

The manuscript is certainly improved, but further (minor) work is needed to render the manuscript suitable for final publication. In particular: I would propose (i) to reduce the number of micrographs shown in Figs. 3 and 4, since some of them are either redundant or not relevant (see the comments on the Figures). On this regard, I would also suggest to remove infos regarding the ductile deformation in garnet and titanite (including shear senses), since not fully pertinent with the scientific rationale of the study (see the commented pdf file); and (ii) to reorganise the text in some parts.

p4L13: I would prefer using tectonic slice...In  any case, please be consistent throughout the text

Reply: Ok, Modified both in the text and figure 2.

p4L24: tectono-metamorphic stages?  reply: Yes

P5L6: Not convincing.
I would remove Fig. 3e. No kynematic data presented in the manuscript Tth growth is interpreted as syn-tectonic (and I agree based on the microtextural evidence shown i Fig. 3), but a "titanite fish" means a pre-tectonic growth...
please, see also the comments on the Figures

Reply: We have removed this photograph. and Fig 3f. Caption corrected accordingly

P5L12: I would remove

We do not agree here, because it is important to highlight the plasticity of Kfs and Pl as an evidence of HT deformation fabric.

P5L13-16:  same reason here: it is not common to spot such a group of evidences of HT deformation, which clearly support the synkinematic character of HT metamorphism, and hence the microstructural/geochronological interpretation relationship. Besides, different zircon morphologies are depicted on each photo, which is important to support the discussion.

We use "plastic" instead of "ductile" because we want to underline the mechanistic connotation (after Rutter, 1986), which is the important point here to relate microstructure and metamorphism.

P8L5-11: we have modified the final part, which is now in the discussion.
P8L19: "that suggests equilibrium growth with garnet?"

     Reply: this is a possibility that's why we wrote: "can be related to the presence of garnet". REE- analyses are required in major phases to confirm the equilibrium growth hypothesis.

P8L20-23: rephrased
P8L44-48: repetitive with what said in lines 5-11.
     Reply: not really, Here appears the interpretation of the trends!.

P10L42-45: rewritten.

[revised manuscript text omitted]